**REPORT**

# A cortical pool of LIN-5 (NuMA) controls cytokinetic furrow formation and cytokinesis completion

Kuheli Adhikary[1] , Sukriti Kapoor[2] , and Sachin Kotak[1]

In animal cells, cleavage furrow formation is controlled by localized activation of the GTPase RhoA at the equatorial membrane using cues transmitted from the spindle. Here, we explore the function of LIN-5, a well-studied protein known for its role in aster separation and spindle positioning in cleavage furrow formation. We show that the cortical pool of LIN-5, recruited by GPR-1/2 and important for cortical force generation, regulates cleavage furrow formation independently of its roles in aster separation and spindle positioning. Instead, our data suggest that enrichment of LIN-5/GPR-1/2 at the polar cortical region is essential to ensure the timely accumulation of contractile ring components—myosin II and Anillin at the equatorial cortex. We additionally define a late cytokinesis role of cortical LIN-5/GPR-1/2 in midbody stabilization and abscission. These results indicate that the cortical LIN-5/GPR-1/2 complex contributes to multiple aspects of cytokinesis independently of its roles in spindle positioning and elongation.

## Introduction

Due to their relatively large size and susceptibility to errors in cell division, embryos face challenges in executing error-free developmental programs. One key challenge that arises during division concerns the precise timing and position of the cleavage furrow. This precision is essential to ensure that when the furrow ingresses, embryos yield daughters with appropriate cytoplasmic mass in terms of quality and quantity. Cleavage furrow formation and its ingression during anaphase are driven by a membrane-associated actomyosin-based structure known as the contractile ring. The accumulation of small GTPase RhoA at the equatorial membrane regulates the contractile ring assembly and its constriction (reviewed in Basant and Glotzer [2018]; Green et al. [2012]; Pollard and O'Shaughnessy [2019]). RhoA directly controls actin polymerization and indirectly regulates myosin II activation at the equatorial membrane, thus helping in contractile ring assembly. Micromanipulation experiments have established that the position of the mitotic spindle dictates the active RhoA zone at the equatorial cortical region (Rappaport, 1985; von Dassow, 2009). The two critical components of the anaphase spindle that control the position and the size of the active RhoA zone are the spindle midzone microtubules (also known as central spindle) and the astral microtubules (Bringmann and Hyman, 2005; Dechant and Glotzer, 2003; reviewed in Mishima [2016]). The spindle midzone is a stable array of antiparallel microtubules that assemble between the segregating chromosomes and accumulate regulatory proteins essential for cytokinesis (Cao and Wang, 1996; Wheatley and Wang, 1996; reviewed in Glotzer [2009]). One key complex that is enriched at the spindle midzone during anaphase and is responsible for RhoA activation at the equatorial membrane is centralspindlin (Somers and Saint, 2003; Yuce et al., 2005). Centralspindlin is a heterotetramer consisting of a dimer of GTPase-activating protein cytokinesis defective 4 (CYK-4; MgcRacGAP or Cyk4 in mammals) and a dimer of kinesin-6 zygotic epidermal enclosure defective 4 [ZEN-4; mitotic kinesin-like protein 1 in mammals] (Davies et al., 2015; Jantsch-Plunger et al., 2000; Mishima et al., 2002; Mollinari et al., 2002; Pavicic-Kaltenbrunner et al., 2007; Verbrugghe and White, 2004). Centralspindlin interacts with Rho guanine exchange factor (GEF)—epithelial cell transforming sequence 2 (ECT-2; Ect2 in mammals)—during anaphase onset and helps in ECT-2 activation at the equatorial membrane, which in turn activates RhoA (Basant et al., 2015; Dechant and Glotzer, 2003; Gómez-Cavazos et al., 2020; Kotynkova et al., 2016; Prokopenko et al., 1999; Su et al., 2011; Tatsumoto et al., 1999; Yuce et al., 2005). Notably, in larger cells such as one-cell *Caenorhabditis elegans* embryo, the spindle midzone carrying centralspindlin pool is present at a significant distance from the equatorial membrane (Basant et al., 2015; Gómez-Cavazos et al., 2020; Jantsch-Plunger et al., 2000; Schlientz et al., 2024, *Preprint*). Most likely because of this reason, centralspindlin is not necessary for the initial enrichment of the contractile ring proteins at the equatorial membrane and, thus, for cleavage furrow formation (Dechant and Glotzer, 2003; Lewellyn

[1]Department of Microbiology and Cell Biology (MCB), Indian Institute of Science (IISc), Bangalore, India;   [2]Molecular, Cell and Developmental Biology, University of California, Los Angeles (UCLA), Los Angeles, CA, USA.

Correspondence to Sachin Kotak: sachinkotak@iisc.ac.in.



et al., 2010; Lewellyn et al., 2011; Werner et al., 2007; Schlientz et al., 2024, *Preprint*). However, centralspindlin is required for proper myosin II accumulation during furrow ingression and cytokinesis completion (Lewellyn et al., 2011; Loria et al., 2012).

How does a contractile ring initially assemble in such large cells where the centralspindlin is placed significantly away from the equator? An elegant spindle-severing experiment in one-cell *C. elegans* embryos revealed that midzone and aster microtubule–based signals usually overlap to define a single furrow formation axis; when manipulated, more than one furrow is established (Bringmann and Hyman, 2005). However, our molecular understanding of aster-based regulation that temporally suppresses the formation of multiple furrows and timely enriches the contractile ring to the equatorial cortex remains limited. In this context, the cortical force generators comprising of abnormal cell lineage-5 (LIN-5; NuMA in mammals), two redundant G protein regulators (referred to as GPR-1/2; LGN or GPSM2 in mammals), and two partially redundant G protein alpha subunits (GOA-1 and GPA-16: collectively referred to as Gα) (Couwenbergs et al., 2007; Kotak et al., 2012; Nguyen-Ngoc et al., 2007; Park and Rose, 2008; Srinivasan et al., 2003) are implicated in timely cleavage furrow induction (Bringmann et al., 2007; Chapa-Y-Lazo et al., 2020; Dechant and Glotzer, 2003; Lewellyn et al., 2010; Price and Rose, 2017). However, since LIN-5/GPR-1/2/Gα regulates multiple processes, including aster separation, spindle positioning, and myosin II stripping from the polar cortical region during anaphase, whether these complexes directly control furrowing and, thus, cytokinesis remained unclear.

Here, relying on the highly precise cleavage furrow initiation timing of the one-cell *C. elegans* embryo, we investigated the function of LIN-5/GPR-1/2 (also referred to as LIN-5–based complexes) in regulating cleavage furrow timing. We show that the impact of cortical LIN-5–based complexes on furrow timing is independent of their function in aster separation, spindle positioning, and their recently proposed role in removing the heavy chain of nonmuscle myosin II (hereafter referred to as NMY-2) from the polar cortical surface. Instead, our data indicate that the cortical enrichment of LIN-5–based complexes at the polar and lateral region of the cell cortex during anaphase ensures timely accumulation of contractile ring proteins—NMY-2 and Anillin at the equatorial cortical region, thereby promoting timely furrow formation. Unexpectedly, we also uncovered a postmitotic role of cortical LIN-5–based complexes in stabilizing contractile ring components to the midbody and, thus, abscission. In summary, our results suggest that cortical enrichment of LIN-5/GPR-1/2 complexes at the polar and lateral surfaces of the cell cortex has two vital functions. First, these complexes regulate the timing of cleavage furrow formation by bolstering the accumulation of contractile ring components at the equator. Second, they confine the contractile ring components to the midbody membrane to warrant accurate abscission.

## Results and discussion

### LIN-5, independent of its role in aster separation, regulates the timing of cleavage furrow formation

We wanted to study the role of LIN-5 in cleavage furrow formation. For this, we performed time-lapse confocal microscopy in embryos coexpressing GFP-tagged plasma membrane (PLCδ1-PH) and tubulin (TBB-2), and mCherry-tagged chromatin (Histone-H2B) and centrosomal (γ-tubulin) markers (referred to as GFP$^{Mem./Tub.}$;mCherry$^{Chr./Cent.}$) and measured the time interval between nuclear envelope breakdown (NEBD)-to-anaphase onset and furrow involution (Fig. S1 A; the appearance of a furrow consisting of two adhered back-to-back plasma membranes in a side view of the embryo; as described in Lewellyn et al. [2010]) in control and *lin-5(RNAi)*. The average timing of NEBD-to-anaphase onset in LIN-5–depleted embryos was similar to that of the control (Fig. S1 B). However, RNAi-mediated depletion of LIN-5 caused a delay of about 60 s in cleavage furrow formation, which is ~25% delay in the mitotic cycle (from NEBD to furrow involution) in the one-cell embryo (Fig. S1, B–D). An analogous delay in cleavage furrow formation was seen upon depletion of LIN-5 cortical anchoring partner GPR-1/2 (Fig. S1, B and E). In contrast, the depletion of centralspindlin components kinesin-6 motor ZEN-4 had no impact on the timing of the cleavage furrow formation (Fig. S1, B and F). However, these embryos failed cytokinesis because of subsequent furrow regression (Fig. S1 F; Dechant and Glotzer, 2003; Raich et al., 1998). Notably, the simultaneous depletion of LIN-5 and ZEN-4 led to the complete loss of furrow formation (Fig. S1 G).

Cortically anchored LIN-5–based complexes via dynein are responsible for the pulling force generation and, thus, for accurate aster separation during anaphase (Fig. 1, A and B; and Fig. S1, H and I; reviewed in Kotak [2019]; Lechler and Mapelli [2021]). Because defective aster separation is linked with delayed furrowing (Lewellyn et al., 2010), we sought to examine the relevance of aster separation on furrow initiation in the context of LIN-5 loss. To do so, we rescued the spindle shortening phenotype seen upon LIN-5 depletion using two independent genetic perturbations. In one setting, we codepleted LIN-5 with the inner kinetochore component HCP-4 (CENP-C ortholog; Fig. S1, J and K; Desai et al., 2003; Oegema et al., 2001), and in another setting, we depleted LIN-5 in embryos carrying temperature-sensitive mutation in *spd-1* (referred to as *spd-1(ts)*) (Fig. S1, L and M; Verbrugghe and White, 2004). HCP-4 depletion disrupts kinetochore microtubules that connect chromosomes to the spindle pole, leading to premature aster separation (Fig. 1 B; Lewellyn et al., 2010; Oegema et al., 2001). Codepletion of HCP-4 and LIN-5 rescued the shorter pole-to-pole distance seen upon LIN-5 depletion (Fig. 1 B). Similarly, the inactivation of spindle defective 1 (SPD-1) (protein regulating cytokinesis 1 [PRC1] in mammals) that helps in bundling the antiparallel spindle midzone microtubules rescued the shorter spindle phenotype observed in *lin-5(RNAi)* embryos (Fig. 1 B). Next, we examined NEBD-to-furrow involution timings in these genetic backgrounds. Although the furrow involution timings in HCP-4–depleted or *spd-1(ts)* mutant embryos were unchanged when compared to control, there was a significant delay in furrow involution timing in embryos that were codepleted for LIN-5 and HCP-4 or *spd-1(ts)* mutant embryos that were depleted for LIN-5 (Fig. 1, C–J). These findings indicate that LIN-5 involvement in furrow initiation does not stem from delayed mitotic progression or its function in aster separation.

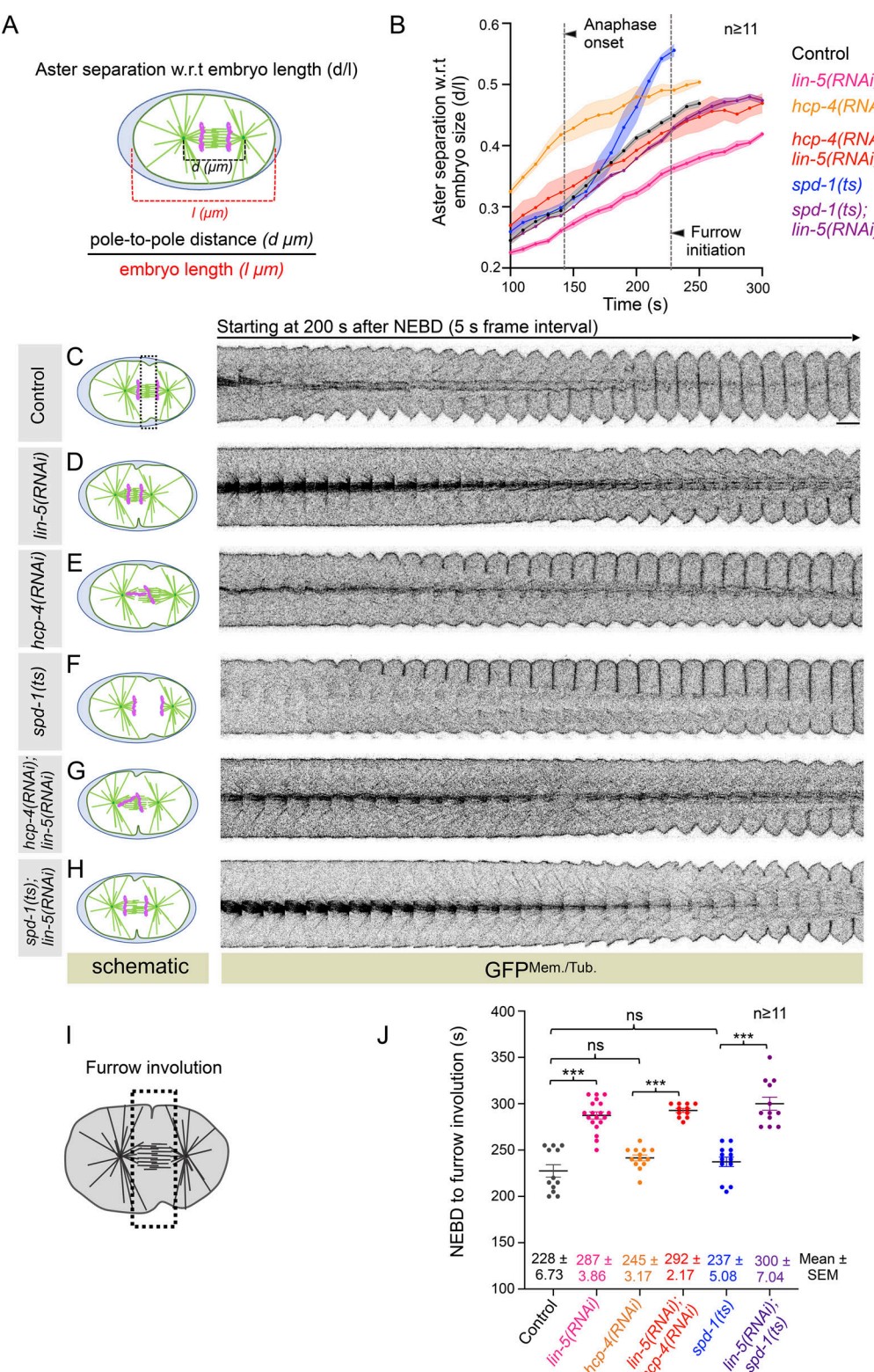

Figure 1. **LIN-5 controls cleavage furrow induction, independent of its role in aster separation. (A and B)** Schematics of the method used for measuring aster separation w.r.t. embryo length (d/l) in embryos coexpressing GFP-tagged PLCδ1-PH (membrane marker), GFP-TBB-2 (microtubules marker), mCherry-H2B (chromatin marker), and mCherry-γ-tubulin (centrosomal marker), referred to as GFP^Mem./Tub.;mCherry^Chr./Cent.. Time is w.r.t. NEBD, which was monitored by the appearance of the GFP-tubulin signal in the pronuclei. In B, dotted straight lines mark the average time for the anaphase onset and furrow initiation for control embryos, as indicated. *n* represents the number of embryos analyzed. The respective shaded color shows error bars in SEM. **(C–H)** Schematics on the left depict the effect of different RNAi conditions for the aster separation (in green) and chromosome behavior (in pink). On the right are the representative pseudokymographs of the division plane acquired by time-lapse confocal microscopy of embryos expressing GFP^Mem./Tub.;mCherry^Chr./Cent., which are either untreated (C), treated with various RNAi, or mutated for *spd-1* (D–H), as indicated. For comparative analysis, only GFP^Mem./Tub. localization is shown in the

pseudokymographs panel. Please note the re-use of the *lin-5(RNAi)* pseudokymograph in Fig. S1 D. Also, see Fig. S1, H–M. Time is w.r.t. NEBD; the scale bar is 5 µm. **(I)** Schematic of a one-cell embryo highlighting the selected equatorial cortical region to refer to furrow involution: the appearance of a furrow composed of two adhered back-to-back plasma membranes in a side view. **(J)** Quantification of the time interval between NEBD and furrow involution in embryos coexpressing GFP$^{Mem./Tub.}$;mCherry$^{Chr./Cent.}$ that are either left untreated, treated with various RNAi, or mutated for *spd-1* (in different colors), as indicated. Each dot represents one embryo, and *n* represents the number of embryos analyzed. In this and other similar graphs used to measure furrow involution timings, the solid black line on the graph represents the mean, whose values and SEM values are also mentioned at the bottom. ns, $P > 0.05$; ***$P < 0.001$ as determined using two-tailed Student's *t* test.

## LIN-5 cortical enrichment, but not its asymmetric distribution at the posterior membrane, is crucial for timely furrow formation

LIN-5–based complexes are asymmetrically localized at the cell cortex; i.e., more complexes are present at the posterior than the anterior cell cortex during anaphase (Fig. 2 A, Fig. 5 A, and Fig. S7 A; Colombo et al., 2003; Gotta et al., 2003; Park and Rose, 2008; Srinivasan et al., 2003). The anisotropic distribution of these complexes at the cell membrane generates unequal pulling forces to position the spindle asymmetrically in the one-cell embryo (Colombo et al., 2003; Gotta et al., 2003; Grill et al., 2001; Kotak et al., 2012; Nguyen-Ngoc et al., 2007; Srinivasan et al., 2003; reviewed in Kotak [2019]). In *lin-5(RNAi)*, because of the absence of cortical pulling forces, the spindle settles either in the embryo center or toward the anterior, leading to a nearly equal division of the one-cell embryo (Fig. S1 I; Srinivasan et al., 2003). Therefore, one possibility for the furrow involution delay seen upon LIN-5 depletion could be the symmetric positioning of the spindle. Thus, we sought to determine the furrow involution timing in the anterior AB cell at the two-cell-stage embryo, where the LIN-5–based complexes are not required for controlling spindle positioning and division axis (Srinivasan et al., 2003; Sugioka and Bowerman, 2018). We performed time-lapse confocal microscopy in embryos expressing GFP$^{Mem./Tub.}$; mCherry$^{Chr./Cent.}$ and investigated the relevance of the LIN-5 in furrow involution timing in the AB cell (Fig. S2, A and B). Notably, LIN-5–depleted AB cells showed a significant delay in furrow involution timing compared with control (Fig. S2, D and E). This delay appears to be independent of the LIN-5 function in preventing proper aster separation, as the delay persisted upon rescuing the aster separation by performing LIN-5 depletion in *spd-1(ts)* mutant embryos (Fig. S2, C–E). These results indicate that LIN-5 regulates furrow timing independent of its role in spindle positioning.

While performing time-lapse confocal microscopy, we noted that a subset of LIN-5–depleted embryos initially set up their spindle in a skewed (tilted) orientation, followed by spindle rotation onto the long A-P axis. These embryos might take additional time to align their spindle at the right axis in anaphase, which could cause a delay in cleavage furrow formation. However, irrespective of whether LIN-5–depleted embryos set up their spindle in a skewed orientation or not, all of them showed a significant delay in cleavage furrow formation (Fig. S2, F–I), suggesting skewed spindle orientation cannot account for the delay in cleavage furrow formation upon LIN-5 depletion.

Next, to examine the relevance of cortical LIN-5 for timely furrow induction in the one-cell embryos, we relied on the depletion of PARtition defective (PAR) proteins. PAR proteins act upstream of LIN-5/GPR-1/2 and regulate their levels at the cell

cortex (Colombo et al., 2003; Gotta et al., 2003; Srinivasan et al., 2003). In the PAR-2–depleted condition, the levels of endogenously tagged LIN-5 with mNeonGreen (mNG) were significantly weaker at the posterior cell cortex (Fig. 2, A, B, D, and E). In contrast, upon PAR-3 depletion, the cortical levels of mNG-LIN-5 were elevated at the anterior cell cortex (Fig. 2, C–E). A similar observation has been made previously for GPR-1/2 (Colombo et al., 2003). These genetic perturbations allowed us to test the relevance of equally weak or equally enriched levels of cortical LIN-5 (and GPR-1/2) on cleavage furrow timing in GFP$^{Mem./Tub.}$; mCherry$^{Chr./Cent.}$-expressing embryos. Notably, PAR-2 depletion, which significantly decreases the cortical levels of LIN-5 at the posterior cell cortex, delays furrow involution (Fig. 2, F, G, K, L, and P). In contrast, having relatively high LIN-5 levels at the anterior cell cortex upon PAR-3–depleted embryos did not impact cleavage furrow formation timings (Fig. 2, H, M, and P). The delay in the cleavage furrow formation that we noticed upon PAR-2 depletion cannot be attributed to the shorter pole-to-pole distance since embryos depleted of PAR-2 in *spd-1(ts)* mutant background, which were rescued for shorter pole-to-pole distance, showed a similar delay in cleavage furrow formation (Fig. 2, I, J, and N–Q). Altogether, these results indicate that (1) LIN-5 cortical enrichment, but not its asymmetric distribution at the cell cortex, is critical for controlling the timings of cleavage furrow formation, and (2) spindle, which is positioned in the embryo center in PAR-2 and PAR-3 depletion condition, is not a critical determinant in controlling the cleavage furrow timing.

## LIN-5 regulates furrow formation independent of its role in removing NMY-2 from the polar cortical region

We established that cortical LIN-5 is critical for timely furrow formation. Cortical LIN-5–based complexes promote dynein accumulation at the cell cortex (reviewed in di Pietro et al. [2016]; Kotak [2019]; Lechler and Mapelli [2021]). Recently, cortical LIN-5/GPR-1/2 and dynein-mediated NMY-2 removal from the polar cortical region toward the spindle poles was hypothesized to be critical for preventing the accumulation of NMY-2 at the polar cortex (Chapa-y-Lazo et al., 2020). This mechanism of active expulsion of NMY-2 from the polar cortical region to promote NMY-2 anisotropy at the equatorial cortex, mediated by LIN-5, was proposed to control cleavage furrow timing (Chapa-y-Lazo et al., 2020). As reported by Chapa-y-Lazo et al., we noticed that the GFP-NMY-2 levels were markedly increased at the posterior cortical region in *lin-5(RNAi)*; *lin-5(ts)* embryos (referred to as LIN-5–inhibited embryos) compared with the control embryos at the time of cleavage furrow formation (compare GFP-NMY-2 localization at the posterior cortical region at 200 s after NEBD between control and LIN-

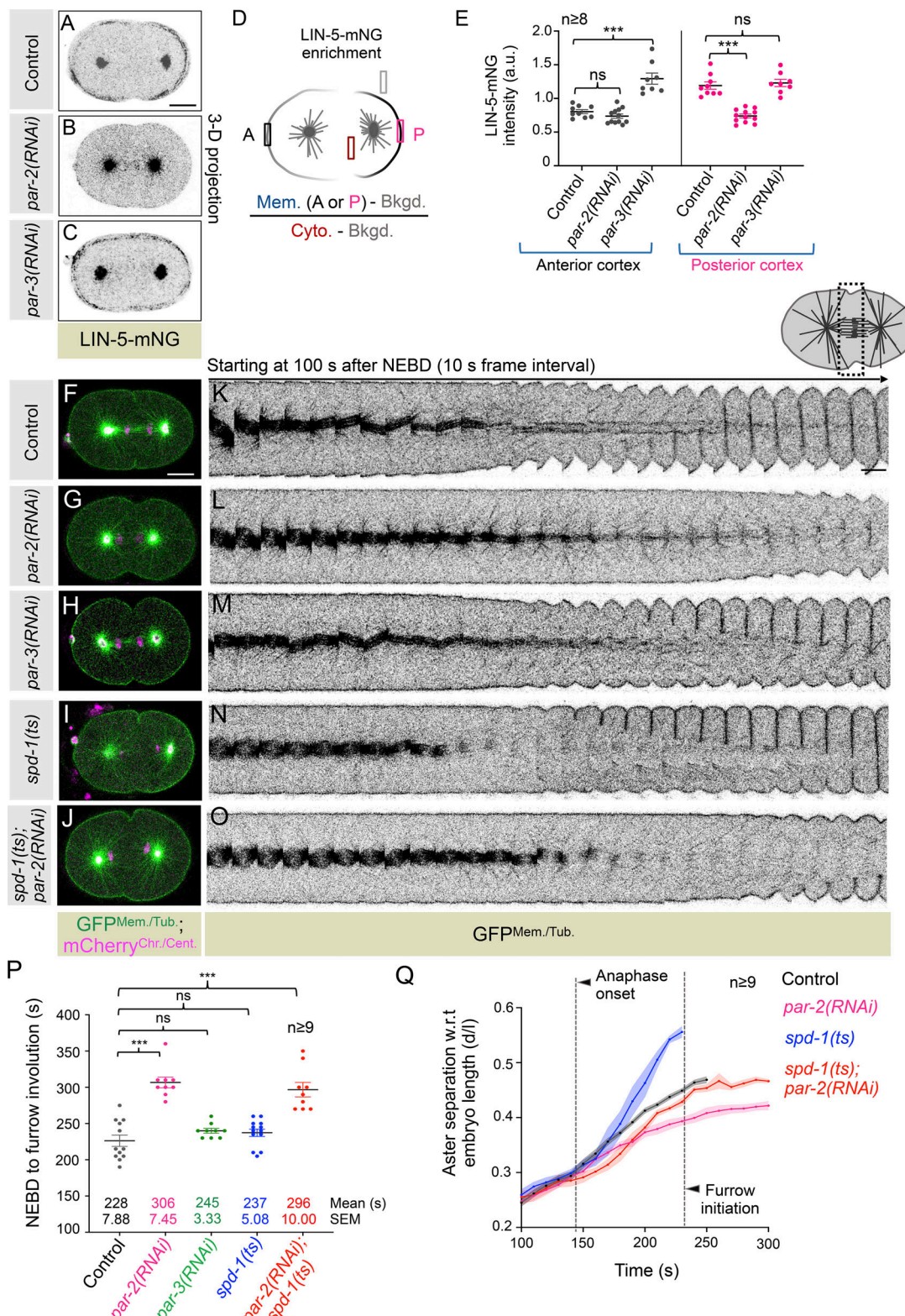

Figure 2. **Cortical accumulation of LIN-5, but not its asymmetric cortical distribution, is vital for timely furrow induction. (A–C)** 3D projected confocal images of a one-cell embryo expressing endogenous LIN-5 tagged with mNeonGreen (mNG) (LIN-5-mNG) at the time of furrow formation in control (A), *par-2(RNAi)* (B), and *par-3(RNAi)* (C). Here, and in subsequent embryo images, the A is to the left, and the P is to the right. The scale bar is 10 µm. **(D)** Schematic of the method to quantify the enrichment of LIN-5-mNG at the furrow initiation stage. Here and in subsequent Fig. panels, Mem. represents membrane intensity; A: anterior; P: posterior; Bkgd. represents background intensity, and Cyto. represents cytoplasmic intensity. **(E)** Quantification of LIN-5-mNG level at the anterior polar cortex (gray circle) and posterior polar cortex (pink circle) in control (*n* = 9), *par-2(RNAi)* (*n* = 12), and *par-3 (RNAi)* (*n* = 8) embryos at furrow onset. *n* is the number of embryos analyzed. Error bars are the SEM; ns, P > 0.05; ***P < 0.001, as determined by two-tailed Student's *t* test. **(F–J)** Representative

images from the time-lapse confocal microscopy of embryo coexpressing GFP$^{Mem./Tub.}$;mCherry$^{Chr./Cent.}$ in control (F) and various RNAi/mutant conditions, as indicated (G–J): control ($n = 12$); par-2(RNAi) ($n = 9$); par-3(RNAi) ($n = 9$); spd-1(ts) ($n = 13$); par-2(RNAi) in spd-1(ts) ($n = 9$). $n$ is the number of embryos analyzed. The scale bar is 10 µm. **(K–O)** Representative pseudokymographs of the division plane (shown on the top right) acquired by time-lapse confocal microscopy of control (K) and various RNAi/mutant embryos coexpressing GFP$^{Mem./Tub.}$;mCherry$^{Chr./Cent.}$, as indicated (L–O). Please note that for the easy comparative analysis of furrow involution timing, only the GFP$^{Mem./Tub.}$ signal is shown (in gray). Pseudokymograph begins at 100 s w.r.t NEBD. The scale bar is 5 µm. **(P)** Quantification of the time interval between NEBD and furrow involution in embryos coexpressing GFP$^{Mem./Tub.}$;mCherry$^{Chr./Cent.}$, which were either left untreated (control), or treated with various RNAi, in spd-1(ts) background, or GFP-$^{Mem./Tub.}$;mCherry$^{Chr./Cent.}$ embryos in spd-1(ts) background that is treated with par-2(RNAi), as indicated. Each dot represents one embryo, and $n$ is the number of embryos analyzed. ns, $P > 0.05$; ***$P < 0.001$, as determined by two-tailed Student's $t$ test. **(Q)** Aster separation w.r.t embryo length (d/l) in embryos GFP$^{Mem./Tub.}$;mCherry-$^{Chr./Cent.}$, as shown for Fig. 1, A and B in control and different RNAi/mutant conditions. Time is w.r.t. NEBD, which was monitored by the appearance of the tubulin signal in the pronuclei. Dotted straight lines mark the average time for the anaphase onset and furrow initiation in control embryos. The respective shaded color shows the error bars in SEM. $n$ is the number of embryos analyzed.

5–inhibited embryos in Fig. 3, A and B; for quantification, see Fig. 3, C and D; and Videos 1 and 2). Similar observations were made by analyzing GFP-NMY-2 localization at the central plane (Fig. S3, A–D; and Videos 3 and 4). Therefore, although this appears to be a reasonable explanation of why LIN-5–depleted embryos delay furrow formation, we noted that in a significant number of LIN-5–inhibited embryos (100%; $n = 9$), the GFP-NMY-2 levels are much more reduced at the anterior cortical region than on the posterior cortex during anaphase (compare 200 s after NEBD between control and LIN-5–inhibited embryos in Fig. 3, A–F; also see Fig. S3, A–F). Coincidentally, we noted that in all such embryos, the spindle is placed relatively closer to the anterior cortex than the posterior cortex. To assess the relationship between spindle-to-cortical distance and its impact on NMY-2 cortical distribution, we measured NMY-2 cortical intensity and pole-to-cortex distance in embryos coexpressing GFP-NMY-2; GFP$^{Tub.}$ (Fig. S3, G–P). The outcome of this analysis indicated that the bulk of NMY-2 clearing from the posterior cortex possibly depends on the position of the spindle. If this hypothesis is correct, then decreasing the distance between the posterior pole and posterior cell cortex should decrease the cortical NMY-2 levels in LIN-5–inhibited embryos. Indeed, increasing the aster separation, and therefore reducing the distance of the posterior pole to the posterior cortex by hcp-4(RNAi), decreases the cortical levels of NMY-2 at the posterior cell cortex in LIN-5–inhibited embryos (Fig. S3, Q–T). Overall, these results indicate that the position of the mitotic spindle is responsible for NMY-2 clearance from the polar cortical region.

To further investigate the role of LIN-5 in furrow formation, independent of its suggested role in NMY-2 removal, we examined embryos depleted of ZYG-9 (zygote defective protein 9; XMAP215 ortholog; Matthews et al., 1998). ZYG-9 is a microtubule-associated protein that promotes the nucleation of long microtubules (Bellanger and Gönczy, 2003; Srayko et al., 2003). In ZYG-9–depleted embryos, two cleavage furrows form: one at the anterior and one at the posterior (Fig. 3, G and H; Werner et al., 2007). Interestingly, in ZYG-9–depleted embryos at the time of furrow formation, cortical GFP-NMY-2 is significantly restricted to the anterior and is largely absent at the posterior, except at the posterior cortical region near the spindle midzone, where the furrow forms (Fig. 3, G and H; Werner et al., 2007). This characteristic localization of GFP-NMY-2 remained unchanged in ZYG-9– and LIN-5–inhibited embryos (Fig. 3, I and J; and Fig. S4, A and B). Since NMY-2 localization at the posterior

cortex is undistinguishable between zyg-9(RNAi) and ZYG-9– and LIN-5–inhibited embryos, this experimental setting offered us an excellent opportunity to examine the function of LIN-5 in controlling posterior cleavage furrow formation, independent of its role in NMY-2 removal. Interestingly, while the posterior furrow formation is stable in zyg-9(RNAi) embryos, we observed periodic oscillations in furrow initiation and regression in ZYG-9– and LIN-5–inhibited embryos (Fig. 3. K–M; and Fig. S4, C and D). In 65% ($n = 17$) of LIN-5– and ZYG-9–inhibited embryos, the cytokinetic furrow underwent cycles of ingression and regression. Similarly, GFP-NMY-2 accumulation at the posterior cortical furrow fluctuated in embryos inhibited for both LIN-5 and ZYG-9 compared with those inhibited for ZYG-9 alone (Fig. 3, N–R). Altogether, these results challenge the notion that LIN-5 role in timely cleavage furrow formation is through NMY-2 removal from the polar cortical region.

Since LIN-5 inhibition in zyg-9(RNAi) embryos reduces spindle length (Fig. S4, E and F), we hypothesized that LIN-5 localizes to the cell cortex nearer to the spindle poles in ZYG-9–depleted embryos. Indeed, we observed mNG-LIN-5 localization at the posterior cortical region, and its levels were higher at the ingressing furrow, likely due to a double membrane (Fig. S4 G). Furthermore, these findings suggest that cortical LIN-5 complexes near the furrow region may facilitate the accumulation of contractile ring proteins, promoting timely and stable furrow formation (see the next section).

### Cortical LIN-5 ensures timely accumulation of contractile ring proteins NMY-2 and Anillin at the equatorial membrane

How do LIN-5–based complexes enrich the polar and lateral cortical regions to ensure accurate furrow timings? Recently, using human cells, we showed that LIN-5 ortholog NuMA, together with dynein/dynactin, is confined to the polar and lateral region of the cell cortex during anaphase (Sana et al., 2022). This distribution of NuMA/dynein/dynactin is critical for the equatorial cortical enrichment of contractile ring regulators RhoA and myosin II (Sana et al., 2022; data not shown). Similarly, we postulated that the enriched pool of LIN-5–based complexes at the polar and lateral cortical regions in C. elegans embryos would be crucial for the timely enrichment of contractile ring–localized proteins, for instance, NMY-2 and Anillin. Thus, in the absence of LIN-5–based complexes, we should observe reduced accumulation of contractile ring proteins at the equatorial cortical region during furrow formation. Indeed, while analyzing the

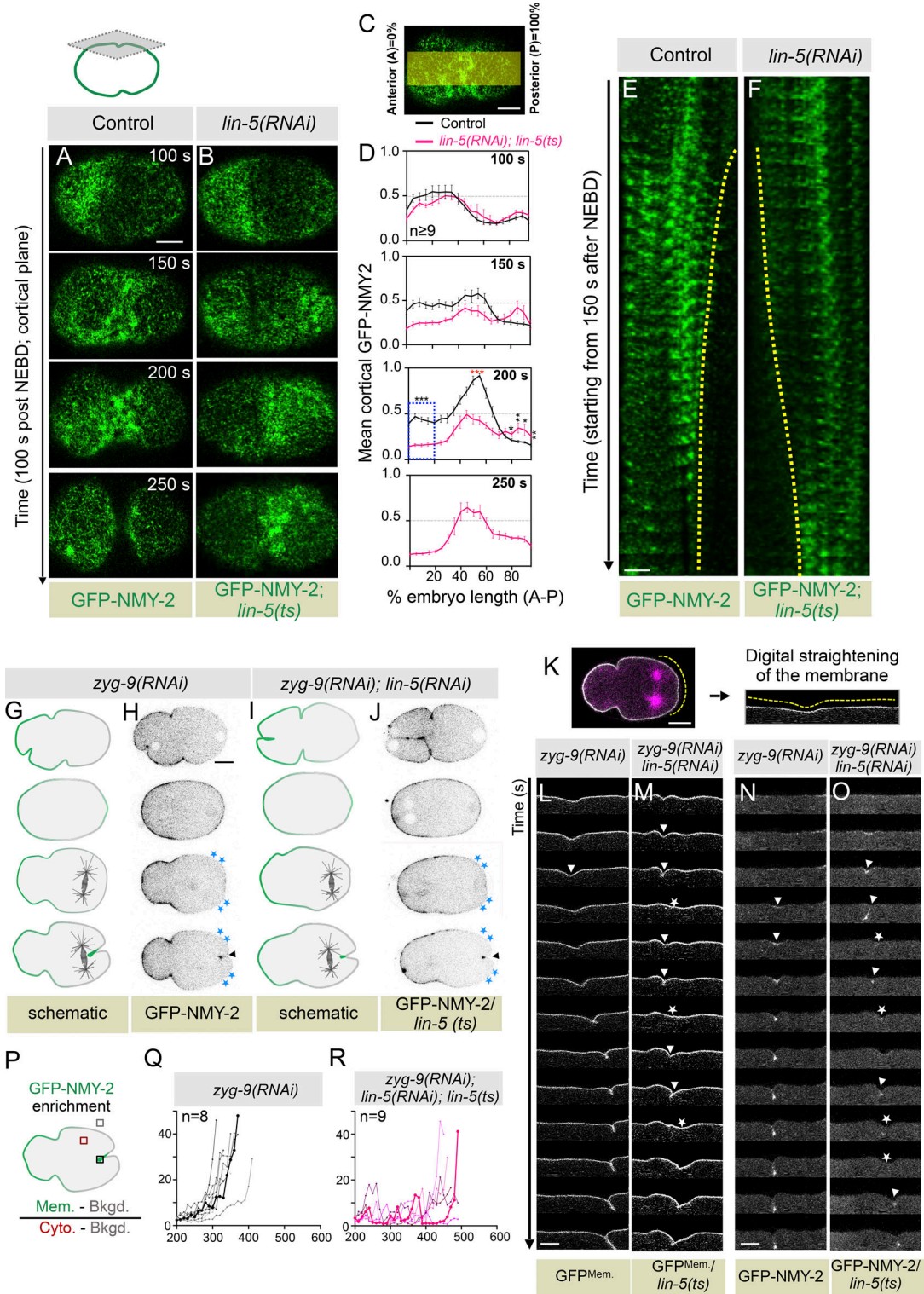

Figure 3. **NMY-2 cortical stripping by LIN-5 is not a key determinant for the cleavage furrow induction. (A and B)** Representative images of the cortical plane from the time-lapse confocal microscopy of the one-cell embryo expressing endogenously tagged NMY-2 with GFP (GFP-NMY-2; in green) in control (A; n = 13) and LIN-5–inhibited [lin-5(RNAi); lin-5(ts)] (B; n = 9) embryos starting from 100 s after NEBD, as indicated. See the corresponding Videos 1 and 2. Please note that LIN-5–inhibited embryos show significant clearance of NMY-2 from the anterior cortical surface at 200 s after NEBD, which is the time when control embryos show furrow induction. Also, note a delay and a significant weak accumulation of GFP-NMY-2 at 200 and 250 s after NEBD at the equatorial cortex in the LIN-5–inhibited embryo. Time is w.r.t NEBD, deduced by the entry of the GFP-NMY-2 signal in the nucleus (not shown). n is the number of embryos analyzed. The scale bar is 10 µm. **(C)** Schematic illustrating the method used to analyze cortical GFP-NMY-2 distribution, as performed by Lewellyn et al. (2010). In brief, a 50-pixel-wide line (~1/2 the embryo width) was drawn, and embryos were divided into 20 equal-length segments from A (0%) to P (100%) to calculate the mean cortical accumulation of GFP-NMY-2 along the A-P axis (see Materials and methods for details). A: anterior; P: posterior. **(D)** Graphs show

the mean cortical intensity of GFP-NMY-2 along the A-P axis (% embryo length A-P) at various time points (in seconds) of control and LIN-5–inhibited embryos after NEBD. Values were normalized by dividing by the average maximum values for controls. GFP-NMY-2 intensity is significantly decreased in LIN-5–inhibited embryos at 200 s after NEBD on the anterior cortical region, compared with control embryos. Also, note a delay and a significant weak accumulation of GFP-NMY-2 at 200 and 250 s after NEBD at the equatorial cortex in LIN-5–inhibited embryos compared with the control embryos. See Fig. S3, A–D for GFP-NMY-2 intensity at the central plane. *n* is the number of embryos analyzed. Error bars are the SEM; ns, P > 0.05; *P < 0.05; **P < 0.01; ***P < 0.001, as determined by two-tailed Student's *t* test. **(E and F)** Kymograph analysis (E, F; see Materials and methods) of the cortical GFP-NMY-2 signal from the control embryo (E) or the LIN-5–inhibited embryo (F). The scale bar is 10 μm. **(G–J)** Schematics of cell cycle stages (G and I) and the corresponding representative images from the time-lapse confocal microscopy (H and J) of embryos expressing GFP-NMY-2 (shown in green in schematic, and gray in confocal images) in *zyg-9(RNAi)* (*n* = 12) (G and H) and *zyg-9 (RNAi)*; *lin-5 (RNAi)* in *lin-5 (ts)* background (*n* = 17) (I and J), as indicated. The blue asterisks on the images indicate the NMY-2 clearance zone before and at the time of furrow involution. Black arrowheads indicate NMY-2 accumulation at furrow involution. *n* is the number of embryos analyzed. The scale bar is 10 μm. **(K–M)** Digitally straightened posterior membrane (K) of an embryo coexpressing GFP$^{Mem.}$ (in gray) and mCherry$^{Tub.}$ (in pink) to analyze the dynamics of furrow formation in the form of pseudokymographs in *zyg-9 (RNAi)* (L), and *zyg-9 (RNAi)*; *lin-5 (RNAi)* in *lin-5 (ts)* background (M), as indicated. For easy comparative analysis, only GFP$^{Mem.}$ fluorescence signal (in gray) is shown for L and M. White arrowheads indicate furrow appearance, whereas white asterisks indicate furrow dissolution. Note that 100% of the ZYG-9–depleted embryos establish a stable posterior furrow, whereas 65% (*n* = 17) of *zyg-9(RNAi)*; *lin-5(RNAi)* in *lin-5(ts)* background fail to form stable posterior furrow and show oscillations in furrow ingression followed by regression. Note the re-use of the *zyg-9(RNAi)* embryo image shown in Fig. 3 K for Fig. S4 C. Also, see Fig. S4, C and D. *n* is the number of embryos analyzed. The scale bar is 10 μm. **(N and O)** Digitally straightened posterior membrane of an embryo expressing GFP-NMY-2 (in gray) to analyze the dynamics of GFP-NMY-2 localization in the form of pseudokymographs in *zyg-9 (RNAi)* (N) and *zyg-9(RNAi)*; *lin-5(RNAi)* in *lin-5 (ts)* background (O), as indicated. White arrowheads indicate GFP-NMY-2 localization at the furrow, whereas white asterisks indicate its dissolution. The scale bar is 10 μm. **(P–R)** Schematic of the method used to quantify GFP-NMY-2 fluorescence intensity at involuted posterior furrow over time w.r.t. NEBD of the male pronucleus (P), in *zyg-9(RNAi)* (Q) and *zyg-9(RNAi)*; *lin-5(RNAi)* in *lin-5(ts)* background (R). NEBD timing was deduced by visualizing the entry of GFP-NMY-2 in the male pronucleus. Each line represents the NMY-2 accumulation pattern of a single embryo. The dark black line in Q and the pink line in R represent the quantification of GFP-NMY-2 of the images shown in N and O, respectively. *n* is the number of embryos analyzed.

dynamics of GFP-NMY-2 in control and LIN-5–inhibited embryos, we noticed significantly weak localization of GFP-NMY-2 at the equatorial cortical region (compare 200 s after NEBD in control and LIN-5–inhibited embryos) in the cortical (Fig. 3, A–D) and central (Fig. S3, A–D) plane. Similarly, an independent time-resolved analysis of GFP-NMY-2 intensity at the equatorial cortical region revealed significantly reduced GFP-NMY-2 levels in LIN-5–inhibited embryos, and that remained significantly low, even at the time of furrow initiation (Fig. 4, A–C, H, and I). Analogous to GFP-NMY-2 intensity, mNG-tagged Anillin (mNG-ANI-1; Lebedev et al., 2023) was localized significantly weakly at the equatorial cortical region in LIN-5–depleted embryos, compared with control (compare 200 s after NEBD in control and *lin-5(RNAi)* embryos in the cortical and the central plane; Fig. S5; and Videos 5, 6, 7, and 8). Next, we assessed the equatorial enrichment of GFP-NMY-2 in *gpr-1/2(RNAi)* and *par-2(RNAi)* embryos, as in both conditions, cortical levels of LIN-5 are diminished. As expected, in GPR-1/2– or PAR-2–depleted embryos, we observed significantly reduced accumulation of GFP-NMY-2 at the equatorial membrane compared with control embryos (Fig. 4, D, E, H, and I).

If the reduced accumulation of contractile ring components at the equator is responsible for the delay in furrow formation seen in LIN-5–inhibited embryos, then ectopically enriching NMY-2 at the equator should rescue this delay. Centralspindlin is the critical component of the contractile ring required for RhoA activation (Fig. S6 A; reviewed in Basant and Glotzer [2018]; Eggert et al. [2006]; Green et al. [2012]). In *C. elegans* embryos, centralspindlin is restricted to the spindle midzone at the time of furrowing, and its avidity to the membrane is compromised by the presence of its negative regulator PAR-5/14-3-3, which prevents the oligomerization of the centralspindlin complex (Fig. S6A; Basant et al., 2015; Douglas et al., 2010). In PAR-5 or PAR-5– and LIN-5–depleted embryos, mNG-tagged centralspindlin component CYK-4-mNG is readily visible

at the equatorial membrane at furrow onset, when compared to control (Fig. S6, B–D; Basant et al., 2015). Concomitantly, this leads to robust accumulation of GFP-NMY-2 at the equatorial cortex in PAR-5 and in PAR-5– and LIN-5–inhibited embryos (Fig. 4, F–I). Notably, the delayed furrow involution timing seen upon LIN-5 inhibition is rescued upon depletion of PAR-5 in LIN-5–inhibited embryos (Fig. 4, J–N and Fig. S6, E–H). These observations suggest that (1) enrichment of LIN-5–based complexes at the polar and lateral surface of the cell cortex is critical to ensure timely localization of contractile ring components at the equatorial cortical region, and (2) because Anillin localization to the equatorial cortex requires RhoA activation, but not myosin II (Hickson and O'Farrell, 2008; Straight et al., 2005), these data indirectly support the idea that cortical LIN-5 positively regulates the equatorial accumulation of contractile ring components, independent of its role of NMY-2 stripping from the polar cortical region to warrant spatiotemporal assembly of the cleavage furrow.

## The cortical pool of the LIN-5/GPR-1/2 complex ensures that the contractile ring components are restricted to the midbody after furrow ingression

Interestingly, LIN-5 and GPR-1/2 are present at the cell cortex even after the first mitosis when the cleavage furrow is fully ingressed (Fig. 5, B and C; and Fig. S7 B). During this time, the spindle midzone and contractile ring are remodeled to form midbody (reviewed in Fededa and Gerlich [2012]; Glotzer [2009]; Green et al. [2012]). The midbody is a densely packed antiparallel microtubule array surrounded by a membrane, and it is composed of spindle midzone–localized proteins, contractile ring components, and their regulators (reviewed in Fededa and Gerlich [2012]; Glotzer [2009]; Green et al. [2012]). As reported previously, we also noted that the components of the contractile ring, including NMY-2 and Anillin, and their regulators, ECT-2, ZEN-4, and CYK-4, are confined to the midbody after furrow

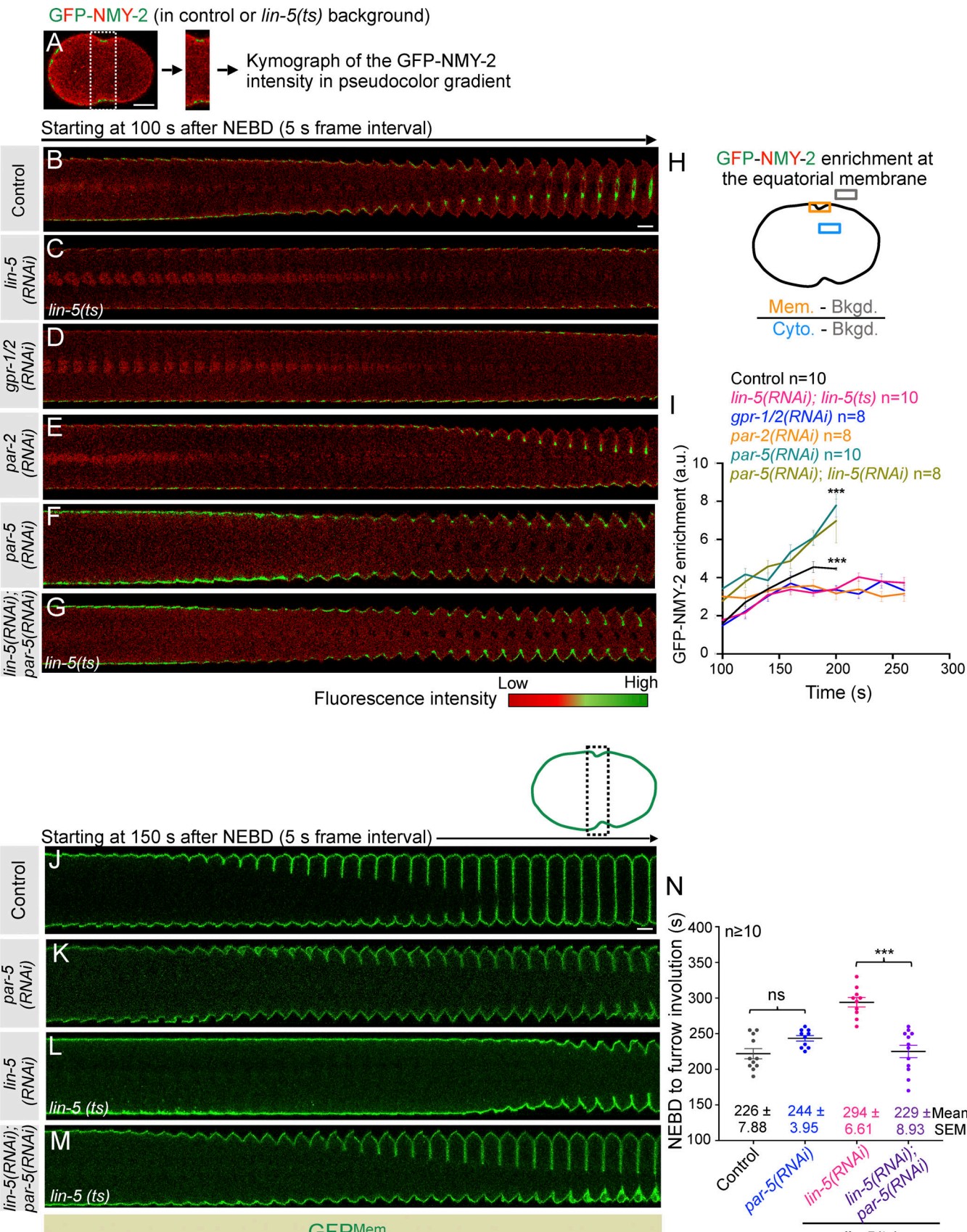

Figure 4. **Cortical levels of LIN-5/GPR-1/2 are essential for the timely enrichment of NMY-2 at the equatorial cortex. (A–G)** Confocal image of a control embryo expressing endogenously tagged NMY-2 with GFP (GFP-NMY-2) in pseudocolor intensity gradient (A). The selected furrow region of GFP-NMY-2–

expressing embryos in control or GFP-NMY-2–expressing embryos in *lin-5(ts)* background is utilized to build time-lapse pseudokymographs under various RNAi conditions, as indicated in B–G: control (B); *lin-5(RNAi)* in *lin-5 (ts)* background (C); *gpr-1/2(RNAi)* (D); *par-2(RNAi)* (E); *par-5(RNAi)* (F); and *lin-5(RNAi)* and *par-5(RNAi)* in *lin-5(ts)* background (G). Pseudokymographs begin at 100 s after NEBD. The scale bar is 10 and 5 μm for the embryo and the pseudokymograph, respectively. A heatmap of fluorescence intensity is given (bottom right) for reference. **(H and I)** Schematic of the method used to quantify GFP-NMY-2 fluorescence intensity at furrow region over time. Quantification of GFP-NMY-2 intensity at the furrow region in control embryos (black line), as well as in *lin-5(RNAi)* in *lin-5 (ts)* background (pink line), *gpr-1/2(RNAi)* (blue line), *par-2(RNAi)* (orange line), *par-5(RNAi)* (cyan line), and *par-5(RNAi)* and *lin-5(RNAi)* in *lin-5(ts)* (asparagus line) embryos. *n* is the number of embryos analyzed. Time is w.r.t. NEBD. Error bars, SEM. ns, P > 0.05; ***P < 0.001, as determined by two-tailed Student's *t* test. Please also see Fig. 3 and Fig. S3 for cortical and central plane localization of GFP-NMY-2 in *LIN-5*–inhibited embryos. **(J–M)** The representative pseudokymographs of the division plane (depleted on the top right) were acquired by time-lapse confocal microscopy of control embryos expressing GFP$^{Mem.}$ and mCherry$^{Tub.}$, which were either left untreated (J), or depleted for PAR-5 (K), *lin-5(ts)* embryos expressing GFP$^{Mem.}$ and mCherry$^{Tub.}$ that were treated with *lin-5(RNAi)* (L), or *lin-5(ts)* embryos expressing GFP$^{Mem.}$ and mCherry$^{Tub.}$ that were treated with *lin-5(RNAi)* and *par-5(RNAi)* (M), as indicated. Please note that only the GFP$^{Mem.}$ signal is shown (in green) for easy comparative analysis of furrow involution timing. Control (*n* = 17); *par-5(RNAi)* (*n* = 10); *lin-5(RNAi)* in *lin-5(ts)* background (*n* = 10); and *lin-5(RNAi)* and *par-5(RNAi)* in *lin-5(ts)* background (*n* = 12). *n* is the number of embryos analyzed. The pseudokymograph begins at 150 s after NEBD. The scale bar is 5 μm. **(N)** Quantification of the time interval between NEBD and furrow involution in control, *par-5(RNAi)* embryos co-expressing GFP$^{Mem.}$ and mCherry$^{Tub.}$, or *lin-5(ts)* embryos coexpressing GFP$^{Mem.}$ and mCherry$^{Tub.}$ that are treated with *lin-5(RNAi)* or *lin-5(RNAi)*, and *par-5(RNAi)*, as indicated. Each dot represents one embryo, and *n* is the number of embryos analyzed. ns, P > 0.05; ***P < 0.001, as determined by two-tailed Student's *t* test.

---

ingression (Fig. 5, B and C; and Fig. S7, C–F; Green et al., 2013; Hirsch et al., 2022). In human cells, we reported that LIN-5 ortholog NuMA-based complexes localize at the polar and lateral region of the cell cortex, and confine centralspindlin and Ect2 at the equatorial cortical region (Sana et al., 2022). Based on this, and on our results related to the involvement of cortical LIN-5/GPR-1/2 in regulating NMY-2 and Anillin levels at the equatorial cortical surface, we hypothesized that the cortical LIN-5–based complexes near the midbody membrane could potentially restrain the accumulation of midbody/midbody membrane–localized proteins. This mechanism may aid in the completion of cytokinesis (Fig. 5 D). Centralspindlin enrichment at the midbody is redundantly controlled (1) by the kinesin-6 motor ZEN-4, which is a part of the centralspindlin complex, and (2) by the physical interaction between CYK-4 and microtubule-bundling protein SPD-1 (Ban et al., 2004; Lee et al., 2015; Hirsch et al., 2022). Our hypothesis predicts that embryos devoid of cortical LIN-5–based complexes might fail to maintain centralspindlin at the midbody cortical membrane surface without a redundant centralspindlin binding partner SPD-1 that tethers centralspindlin to the midzone microtubule bundle. To test this assumption, we first monitored the localization of CYK-4-mNG during and after furrow ingression in embryos disrupted for LIN-5 and SPD-1. In control embryos, CYK-4-mNG pools are enriched at the spindle midzone during furrow involution and are maintained after furrow ingression in the midbody (Fig. 5, E, I, and J). Consistent with the previous observations, in the *spd-1(ts)*, CYK-4-mNG was initially undetectable during furrow involution because of the absence of the spindle midzone (Fig. 5 F). However, in these embryos, CYK-4-mNG is accumulated later at the time of furrow closure because of the bundling of astral microtubules (Fig. 5, F, I, and K; Hirsch et al., 2022). In *lin-5(RNAi)* embryos, we noticed a more robust accumulation of CYK-4-mNG at the spindle midzone during furrow involution, which is maintained after furrow ingression (Fig. 5, G, I, and L). This enriched pool of CYK-4-mNG at the spindle midzone in LIN-5–depleted embryos is most likely due to the absence of dynein-dependent cortical pulling forces that counteract the assembly of spindle midzone in control embryos (Lee et al., 2015). Notably, in embryos that are disrupted for LIN-5 and

SPD-1, the CYK-4-mNG pools at the spindle midzone are significantly enriched compared with *spd-1(ts)* mutant during furrow involution; however, after furrow ingression, these pools were not maintained in the midbody (Fig. 5, H, I, and M). We further confirmed that the loss of CYK-4-mNG pools that we observed in *lin-5(RNAi); spd-1(ts)* embryos was not because of the absence of astral microtubules in *lin-5(RNAi); spd-1(ts)* embryos (Fig. S7, G–J).

Next, we analyzed the localization of Anillin, which is part of the midbody membrane (Kechad et al., 2012). Anillin is localized to the midbody in *spd-1(ts)* mutant embryos, but similar to CYK-4, its localization to the midbody was not maintained in *lin-5(RNAi); spd-1(ts)* embryos. Additionally, we noted that cortical levels of Anillin spread along the juxtaposed membranes in *lin-5(RNAi); spd-1(ts)* embryos (Fig. S7, K–N), indicating that LIN-5 helps in confining contractile ring components to the midbody.

Because midbody-localized contractile ring components are essential for cytokinesis completion (Kechad et al., 2012; Lekomtsev et al., 2012; Simon et al., 2008; Zhao et al., 2006), we next analyzed the relevance of LIN-5/GPR-1/2 cortical accumulation for cytokinesis completion in control, *spd-1(ts)*, *lin-5(RNAi)*, *gpr-1/2(RNAi)*, *spd-1(ts)* and *lin-5(RNAi)*, and *spd-1(ts)* and *gpr-1/2(RNAi)* in embryos expressing GFP$^{Mem./Tub.}$; mCherry$^{Chr./Cent.}$. Notably, in contrast to control, *spd-1(ts)*, and *lin-5(RNAi)*, where we found no instances of furrow regression upon furrow ingression, simultaneous disruption of LIN-5 and SPD-1 resulted in furrow regression in 41% (*n* = 17) of the embryos (Fig. 5, N–U; and Videos 9, 10, 11, and 12). This phenotype was much more pronounced (60%; *n* = 15) in GPR-1/2– and SPD-1-disrupted embryos (Fig. S8, A, C, E, and G; and Video 13). Because GPR-1/2 is necessary for the cortical accumulation of LIN-5 and not for its spindle pole localization, our work indicates that cortically anchored LIN-5–based complexes, and not the spindle pole–enriched pool of LIN-5, help in midbody stabilization and cytokinesis completion. In accordance, ASPM-1 (abnormal spindle-like microcephaly associated)–depleted embryos, which fail to accumulate LIN-5 at the spindle poles (data not shown; Van de Voet et al., 2008), do not show cytokinesis failure after furrow ingression in the *spd-1(ts)* mutant background (Fig. S8, B, D, F, and H).

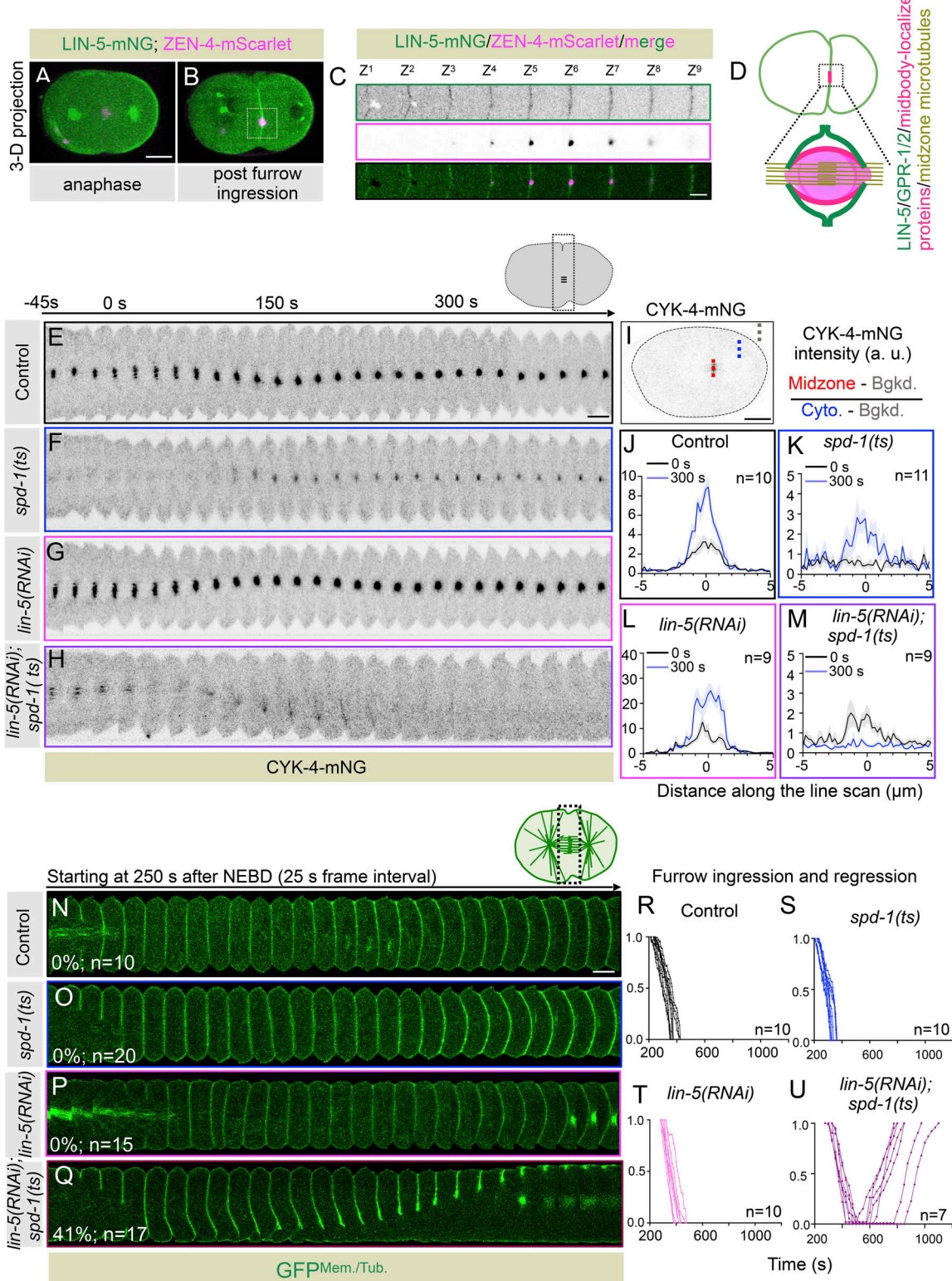

Figure 5. **Postmitotic localization of cortical LIN-5/GPR-1/2 is vital for midbody stability and accurate abscission. (A and B)** 3D projected confocal images of one-cell embryo coexpressing LIN-5-mNG and ZEN-4-mScarlet during anaphase (A) and after furrow ingression (B). 25 images were acquired, and representative images are shown here; the scale bar is 10 μm. **(C)** Different z-sections (1 μm apart) of the selected region of LIN-5-mNG– and ZEN-4-mScarlet–coexpressing embryo, shown in B. Please note that LIN-5 localizes at the membrane surrounding the midbody (labeled with ZEN-4-mScarlet). A similar observation has been made for GPR-1/2 localization; see Fig. S7 B. The scale bar is 5 μm. **(D)** Schematic of a one-cell embryo highlighting the selected midbody region and the surrounding membrane. The midbody is composed of an antiparallel microtubule bundle (in asparagus)–, several midbody (in pink)–, and midbody membrane (in strawberry)–localized proteins, and its surrounding membrane is composed of LIN-5/GPR-1/2 complexes (in green). Also see Fig.

S9 B. **(E–H)** Representative pseudokymographs of the division plane (depicted on the top right) acquired by time-lapse confocal microscopy of control embryos expressing CYK-4-mNG (E), CYK-4-mNG in *spd-1(ts)* background (F), CYK-4-mNG embryos treated with *lin-5(RNAi)* (G), or CYK-4-mNG in *spd-1(ts)* background that are depleted for LIN-5 (H). The pseudokymograph begins at −45 s w.r.t furrow involution, which is marked as 0 s. The furrow involution timing was determined using DIC microscopy (not shown). Please note that the CYK-4-mNG is absent at the spindle midzone in *spd-1(ts)* mutant embryos at furrow involution; however, it is regained later as the cell cycle progresses, as recently shown by Hirsch et al. (2022). In contrast, the *spd-1(ts)* mutant embryos, when treated with *lin-5(RNAi)*, show significantly enriched localization of CYK-4-mNG at the spindle midzone during furrow involution, but CYK-4-mNG midzone localization is not maintained. The scale bar is 5 µm. **(I–M)** Schematic of the method used to quantify CYK-4-mNG fluorescence intensity at the spindle midzone during furrow involution (0 s) and 300 s after furrow involution (I). CYK-4-mNG fluorescence intensity at 0 and 300 s in control embryos (J), *spd-1(ts)* mutant embryos (K), *lin-5(RNAi)* embryos (L), and *spd-1(ts)* mutant embryos that are treated with *lin-5(RNAi)* (M). Please note that the midzone CYK-4-mNG fluorescence intensity is negligible in *spd-1(ts)* mutant embryos at 0 s but not at 300 s. In contrast, *spd-1(ts)* mutant embryos depleted for LIN-5 show robust CYK-4-mNG intensity at 0 s, but no enrichment is seen at 300 s. *n* is the number of embryos analyzed. The respective shaded color shows error bars in SEM. **(N–Q)** Representative pseudokymographs of the division plane (depleted on the top right) acquired by time-lapse confocal microscopy of GFP^Mem./Tub.;mCherry^Chr./Cent.-expressing untreated control embryos (N), in *spd-1(ts)* mutant background (O), treated with *lin-5(RNAi)* (P), or *spd-1(ts)* embryos that are also treated with *lin-5(RNAi)* (Q), as indicated. Only the GFP^Mem./Tub. signal is shown (in green) for easy comparative analysis of abscission failure. See the corresponding Videos 9, 10, 11, and 12. Control (*n* = 10); *spd-1(ts)* (*n* = 20); *lin-5(RNAi)* (*n* = 15); and *lin-5(RNAi)* in *spd-1(ts)* background (*n* = 17). Note that control, *spd-1(ts)*, or *lin-5(RNAi)* does not show any abscission failure; however, *spd-1(ts)* embryos that are treated with *lin-5(RNAi)* show 41% (*n* = 7) abscission failure. *n* is the number of embryos analyzed. The pseudokymographs begin at 250 s after NEBD, as indicated. The scale bar is 5 µm. **(R–U)** Graphs represent furrow ingression kinetics in embryos coexpressing GFP^Mem./Tub.;mCherry^Chr./Cent. in control (R), *spd-1(ts)* mutant embryos (S), *lin-5(RNAi)* embryos (T), and *spd-1(ts)* mutant embryos that are treated with *lin-5(RNAi)* (U). Please note furrow ingression followed by regression in *spd-1(ts)* mutant embryos depleted for LIN-5. A similar observation was made for *spd-1(ts)* mutant embryos depleted for GPR-1/2 (see Fig. S8). *n* is the number of embryos analyzed. Time is w.r.t. NEBD.

Recent work has reported that SPD-1–disrupted embryos show ~40% abscission failure in P0 in embryos expressing NMY-2-GFP (Santos et al., 2023), which was not reported earlier when *spd-1(ts)* was tested at the restrictive temperature of 25–26°C or in *spd-1(RNAi)* conditions (Green et al., 2013; Hirsch et al., 2022; Lee et al., 2015; Verbrugghe and White, 2004). To ensure that the absence of abscission defects in *spd-1(ts)* mutant strain expressing GFP^Mem./Tub.;mCherry^Chr./Cent. is not likely because of residual SPD-1 activity in these transgenic worms, we analyzed and recorded the division of ~50 *spd-1(ts)* mutant embryos at 25°C/26°C by differential interference contrast (DIC) microscopy. 100% of these embryos showed broken spindle phenotype during anaphase because of the absence of spindle midzone; however, none revealed cytokinesis failure/abscission defects (data not shown). Overall, these results indicate that the postmitotic membrane enrichment of LIN-5–based complexes in the vicinity of the midbody confines contractile ring components at the midbody and the surrounding midbody membrane to ensure proper cytokinesis completion.

The spatiotemporal accumulation of RhoA/NMY-2 at the equatorial membrane plays a pivotal role in contractile ring assembly (Bement et al., 2005; reviewed in Basant and Glotzer [2018]; Green et al. [2012]). However, the precise mechanisms that govern the timing and the confinement of RhoA and NMY-2 pools within the narrow equatorial cortical surface to facilitate cytokinesis remain poorly understood. It is hypothesized that the communication between a stimulating cue at the equatorial membrane and an inhibitory signal at the polar region of the cell cortex regulates RhoA and NMY-2 to a restricted membrane zone (reviewed in Basant and Glotzer [2018]; Green et al. [2012]). In support of this hypothesis, previous studies have demonstrated that Aurora A kinase and its coactivator TPXL-1 enriched at the spindle poles and astral microtubules can act as an inhibitory signal that helps in clearing contractile ring components, including Rho GEF ECT-2, Anillin, and filamentous actin from the polar cortical region, in proximity to the spindle (Kapoor and Kotak, 2019; Longhini and Glotzer, 2022; Mangal et al., 2018). Additionally, here, we discovered the role of cortically accumulated LIN-5–based complexes in cleavage furrow formation. In the absence of LIN-5, contractile ring components NMY-2 and Anillin are weakly localized to the equatorial cortex at the time of cleavage furrow formation. This outcome could be linked to failure in proper aster separation in LIN-5–inhibited embryos. However, we show that cortical LIN-5–based complexes regulate the timing of cleavage furrow formation independent of their role in aster separation. Therefore, we think that the enrichment of LIN-5/GPR-1/2 complexes at the polar and lateral region of the cell cortex regulates the timely and robust accumulation of contractile ring components, including NMY-2 and Anillin at the equator (Fig. S9 A). We further show that LIN-5 controls furrow formation independent of its role in spindle positioning and NMY-2 expulsion. This function of cortical LIN-5 or its interacting partners in promoting contractile ring assembly is not unique to *C. elegans* embryos. Earlier observations in *Drosophila* neuroblasts have shown the spindle-independent function of the apically localized partner of inscrutable (Pins; *C. elegans* GPR-1/2 ortholog) in restricting myosin II to the basal cortical region (Cabernard et al., 2010). Similarly, using human cells, we recently showed the relevance of the polarized distribution of NuMA/dynein/dynactin at the polar region of the cell cortex restricts the equatorial enrichment of RhoA at the equatorial membrane (Sana et al., 2022). Therefore, we envisage these evolutionarily conserved proteins have a crucial moonlighting function other than their well-characterized role as cortical force generators in ensuring robustness in cleavage furrow formation. The mechanistic insights of how LIN-5–based complexes enriched at the cell membrane ensure timely accumulation of NMY-2 and Anillin warrant further investigation. However, we envisage two possibilities that are not mutually exclusive: (1) the cortical-localized LIN-5–based complexes at the polar region of the cell cortex sterically confined the localization of contractile ring components to the equator, and/or (2) cortical LIN-5/GPR-1/2–based complexes interact with an enzyme, most likely a kinase/phosphatase that assist in confining the contractile ring components to the equator for proper cleavage furrow assembly.

Interestingly, we also discovered the postmitotic function of cortical LIN-5–based complexes in midbody stabilization (Fig. S9 B). When depleted of LIN-5 or GPR-1/2, many embryos with compromised spindle midzone open their ingressed furrows. In these embryos, the contractile ring components CYK-4 and Anillin initially localized to the midbody; however, these proteins did not remain confined to the midbody, leading to furrow regression and, ultimately, cytokinesis failure. We are of the opinion that the cortically accumulated LIN-5/GPR-1/2 complexes in the vicinity of the midbody confine the midbody/midbody membrane–localized components to promote midbody stabilization for error-free abscission (Fig. S9 B). In summary, this work has characterized an essential but hitherto not well-studied spindle-independent function of cortical force generators in controlling the timing of cleavage furrow formation and the stability of the midbody for proper cytokinesis.

## Materials and methods

### *C. elegans* strains and their maintenance

*C. elegans* wild-type (N2) and transgenic lines were maintained at either 20°C or 25°C using nematode growth medium (NGM) plates seeded with *Escherichia coli* (OP50). While performing RNAi, *C. elegans* strains were grown on *E. coli* (HT115) containing RNAi expressing vector as a food source. Temperature-sensitive strains were maintained at 16°C. A list of all strains used in the study has been provided in Table S1.

### RNAi-mediated interference

Bacterial RNAi feeding strains were obtained from the *C. elegans* ORFeome RNAi library (Rual et al., 2004) or Source BioScience (Kamath et al., 2003) or cloned to the multiple cloning site of the L4440 vector, followed by transforming into HT115 *E. coli* (Timmons et al., 2001).

A detailed description of each bacterial feeding strain used for RNAi experiments, the duration of RNAi, and primers used to design RNAi clones is provided in Table S2. For making NGM agar plates having to feed RNAi strain, a secondary culture of RNAi feeding bacteria was grown in Luria broth with ampicillin (100 µg/ml) until OD 0.6–0.8 at 37°C, followed by induction with 2 mM IPTG for 1–2 h at the same temperature before plating 1 ml of this culture on the NGM agar plates supplemented with 1 mM IPTG. These plates were kept at room temperature for 24 h before starting RNAi experiments. L1–L4 stage larval worms (depending on the experiment) were plated on RNAi plates and incubated at 20° and/or 25°C for 24–72 h before dissection of adult hermaphrodites to obtain embryos. While performing double depletion using RNAi, the single depletion was performed side by side to observe the efficiency of single RNAi by monitoring RNAi-dependent phenotypes.

For each experiment, RNAi-mediated knockdown was confirmed by phenotypic analysis. For instance, strong LIN-5 depletion by RNAi results in meiotic defects and the loss of cortical pulling forces, which leads to the division of P0 embryos either equally or unequally with smaller AB and larger P1 cells, as previously reported in Srinivasan et al. (2003). Also, for a subset of experiments, a robust LIN-5 inhibition was achieved by

performing *lin-5(RNAi)* in the background of *lin-5(ts)*. Only embryos that fulfill the criteria of the strong published phenotype were considered for the quantitative analysis. In a few instances where we had fluorescence-tagged lines for the genes against which we were conducting RNAi, we also assessed the loss of fluorescently tagged protein signal after RNAi to check the efficacy of our RNAi conditions. For temperature-sensitive (ts) lines [*lin-5(ts)*; Lorson et al., 2000, or *spd-1 (ts)*; Verbrugghe and White, 2004], RNAi was done as follows: worms were upshifted to a restrictive temperature (25°C) (exception for *spd-1(ts)* line where we also performed some experiments at 26°C) for the last 24 h of total experimental duration. Since PAR-5 or LIN-5 depletion causes a similar pattern of P0 division (nearly equal division of P0), to ensure robust LIN-5 inhibition, PAR-5 depletion by RNAi was combined with LIN-5 in *lin-5(ts)* background along with the control of *lin-5(RNAi)* in *lin-5(ts)* to be confident for the LIN-5 inhibition. In such instances, single RNAi was performed side by side in N2 to check the efficiency of RNAi. Similarly, for the codepletion analysis of ZYG-9 and LIN-5 by RNAi, we have performed the experiments in *lin-5(ts)* background since the RNAi experiment duration was only 24 h, and in our experience, a strong LIN-5 depletion phenotype with no posterior spindle pole oscillations and nearly equal division of P0 is only visible after >40 h. However, for this experiment, the control L4 stage larval worms (transgenic strains or transgenic strains in *lin-5(ts)* background) were seeded on the feeding plates containing either the RNAi feeding strain for ZYG-9 or for both ZYG-9 and LIN-5 for 24 h at 25°C.

### Live-cell imaging

For recording embryos, gravid worms were dissected in M9 and transferred onto a 2% agarose pad containing slides using a mouth pipette. These were then covered with an 18 × 18 mm coverslip. Time-lapse DIC microscopy or confocal microscopy in combination with DIC was performed on such embryos either on IX53 (Olympus Corporation) with QImaging MicroPublisher 5.0 Color CCD Camera (QImaging) with 100× 1.4 NA objective, or on FV3000 Confocal system with high-sensitivity cooled GaAsP detection unit (Olympus Corporation) using 60× 1.4 NA objective. Images were acquired on inbuilt FV3000 software. Depending on the experiments, images were collected at 5- to 25-s intervals per frame. Movies were subsequently processed using Fiji/ImageJ, maintaining relative image intensities within a series. Z stack series (9 × 1.5 µm z-sections) were projected as maximum intensity projections for embryos expressing CYK-4-mNG in control and the mentioned perturbations (Fig. 5, E–H). Confocal images (1-µm z-sections apart) were projected as 3D projections for embryos expressing LIN-5-mNG (Fig. 2, A–C), LIN-5-mNG; ZEN-4-mScarlet (Fig. 5, A and B), and YFP-GPR-1 (Fig. S7, A and B) from the time-lapse movies.

The kymograph analysis of midplane GFP-NMY-2 and GFP-PH in *zyg-9(RNAi)* conditions (Fig 3, K–O) was done by digitally straightening the membrane using Fiji/ImageJ software ("straighten" is an inbuilt plugin in the software). Images of overlapping microtubules are the maximum intensity projection of three z-stacks (Fig. S7).

## Quantification and statistical analysis

Fiji/ImageJ (https://fiji.sc/) and GraphPad Prism were used for quantitative analysis. Mean values are shown with error bars representing either SEM or SD, as indicated. Statistical significance was calculated using unpaired two-tailed Student's *t* test after checking if the data show normal or nearly normal distribution using the Shapiro–Wilk and Kolmogorov–Smirnov tests. The P value was significant if P <0.05 using GraphPad Prism 9. The P values are mentioned either in the text or on the graph, and the significance values are mentioned as *n.s.*, >0.05; *P < 0.05; **P < 0.01; ***P < 0.001.

## Assigning time "0"

For most datasets and graphs, t = 0 corresponds to NEBD. For CYK-4-mNG–based images (Fig. 5, E–H), t = 0 is the approximate furrow involution time, which was judged by DIC images. For images from different experiments performed in time-lapse confocal microscopy, NEBD was evaluated by a visible inclusion of GFP-TBB-2, mCherry-TBB-2, GFP-NMY-2, or mNG-ANI-1 signal inside the pronuclei.

## Image analysis

All image analysis, including fluorescence intensity measurements and line scan analysis, was performed using Fiji/ImageJ (https://fiji.sc/). As mentioned in the figure or figure legend, a particular plane (cortical or midzone) or 3D projection was used for representation. Images were pseudocolored, and brightness and contrast were equally adjusted using Fiji/ImageJ for representation.

## Quantification of aster separation kinetics

Aster separation kinetics in Fig. 1 B and Fig. 2 Q was performed by drawing a line between two spindle poles to measure the distance (in microns) and dividing the value with the embryo length (in microns) in every 10 s starting from 100 s w.r.t. NEBD. The embryo length is considered because of the slightly varied length from embryo to embryo for normalization. The ratio between pole-to-pole distance and embryo length was plotted over time.

## Analyzing spindle pole–cortex distance

Distance from the anterior spindle pole to the anterior cortex and similarly posterior spindle pole to the posterior cortex (in microns) was measured every 15 s starting from 100 s until furrow involution w.r.t. NEBD. As shown in the figure panel, either individual pole–cortex distance in microns (Fig. S3, K–N) or the ratio between the anterior spindle pole-to-anterior cortex distance and the posterior spindle pole-to-posterior cortex distance (Fig. S3, O and P) was utilized to plot the graph.

## Analysis of furrow ingression kinetics

Furrow ingression and regression were measured by taking the distance (in microns) between two opposing ingressing membranes at the furrow region just before and after the ingression at every 5-s interval. The value, plotted for the graph over time, is a ratio of the distance between the opposing ingressing membranes to the distance between opposing membranes just before ingression (Fig. 5, R–U and Fig. S8, E–H) with time taken w.r.t. NEBD.

## Quantification of fluorescence intensity

### Quantification of contractile ring proteins NMY-2 and Anillin at the cortical plane

For the analysis of cortical GFP-NMY-2 and mNG-ANI-1 as a function of embryo length (Fig. 3, C and D; and Fig. S5, C and D), a 50-pixel-wide (~1/2 of the width of the embryo) line bisecting the embryo was drawn from the anterior to posterior tips of the embryo. Using Fiji/ImageJ software, an average intensity line scan was generated for each time point. Embryos were divided into 20 equal-length segments from anterior (0% embryo length) to posterior (100% embryo length), and the mean GFP-NMY-2/mNG-ANI-1 in each segment was calculated for each time point. For data normalization, all intensity values were divided by the average maximum intensity of control embryos (measured at 55–65% embryo width, 200 s after NEBD at the time of cleavage furrow formation).

For the analysis of GFP-NMY-2 and mNG-ANI-1 intensity at the central plane as a function of embryo length (Fig. S3, C and D; and Fig. S5, G and H), a segmented line of thickness of 1 μm was drawn over the membrane from anterior (0% embryo length) to posterior (100% embryo length). Using Fiji/ImageJ software, an average intensity line scan was generated for each time point. Embryos were divided into 20 equal-length segments from anterior (0% embryo length) to posterior (100% embryo length), and the mean GFP-NMY-2/mNG-ANI-1 in each segment was calculated for each time point. For data normalization, all intensity values were divided by the average maximum intensity of control embryos (measured at 55–65% embryo width, 200 s after NEBD at the time of cleavage furrow formation) imaged in parallel.

To quantify midplane GFP-NMY-2 intensity in *zyg-9(RNAi)* and *zyg-9(RNAi)* and *lin-5(RNAi)* in *lin-5(ts)* embryos, a segmented line of thickness of one pixel wide was drawn over the membrane from anterior to posterior. Using Fiji/ImageJ software, an average intensity line scan was generated for the A-P-A (from the anterior–posterior–anterior membrane) axis at the time of posterior furrow formation (Fig. S4 B).

To quantify the intensity of NMY-2 accumulation at ingressing posterior furrow region in *zyg-9(RNAi)* and *zyg-9(RNAi)* and *lin-5(RNAi)* in *lin-5(ts)* embryos (Fig. 3, Q and R), a fixed square ROI (1.5 μm²) was placed on the NMY-2 accumulation site and taken as the intensity in every 10 s during furrow ingression. The integrated fluorescence intensity in this ROI and a similarly sized ROI in the cytoplasm and background were determined using Fiji/ImageJ. Fluorescence intensity at the ingressing tip and cytoplasm was subtracted from the background, and the ratio of the ingressing region to the similar cytoplasmic region was plotted to indicate NMY-2 enrichment at the ingressing furrow.

To quantify midplane GFP-NMY-2 intensity at furrow-forming membrane at the equatorial plane (Fig. 4 H), a fixed rectangular ROI (ROI of 5.1 μm²) was placed on the furrow-forming site and taken in 20-s intervals starting from 100 s w.r.t. NEBD. The integrated fluorescence intensity in this ROI

and a similarly sized ROI in the cytoplasm and background were determined using Fiji/ImageJ. Fluorescence intensity at the furrow-forming site and cytoplasm was subtracted from the background, and the ratio of the furrow-forming site to cytoplasm to indicate equatorial NMY-2 enrichment was used for plotting the graph. Similar quantification was done in Fig. S3, I, J, S, and T to measure the NMY-2 fluorescence intensity at the central plane in the polar cortical region with time. A fixed ROI (ROI of 2.0 μm²) was taken and measured every 15 s starting from 100 s w.r.t. NEBD until the cleavage furrow formed. The integrated fluorescence intensity in this ROI and a similarly sized ROI in the background were determined using Fiji/ImageJ. Fluorescence intensity at the anterior and posterior polar cell cortex was subtracted from the background. After subtraction, the ratio of NMY-2 intensity of the anterior polar cortex to the posterior polar cortex was used for plotting the graph.

### CYK-4-mNG and mNG-ANI-1 intensity at the spindle midzone/ midbody after furrow ingression

For the line scan analysis of CYK-4-mNG fluorescence intensity at the spindle midzone/midbody (Fig. 5, I–M), a vertical line of 10 μm was drawn on the spindle midzone. The integrated fluorescence intensity in this line and an identical-sized line in the cytoplasm and background were determined using Fiji/ImageJ. Fluorescence intensity of either spindle midzone during furrow involution (0 s) or midbody (300 s after involution) and cytoplasm was subtracted from the background, and the ratio of spindle midzone or midbody intensity to the cytoplasm was deduced to indicate CYK-4-mNG enrichment. Note that a sum projection of the three z-stacks in which the midbody was most visible (or three central slices, if no midbody was visible) was utilized. A similar quantification was performed for midbody accumulation analysis of mNG-ANI-1 intensity at 300 s after involution at the midbody without normalization (Fig. S7, M and N).

### LIN-5-mNG fluorescence intensity

Quantification of LIN-5-mNG was performed by taking a rectangular fixed ROI (ROI of 2.0 μm²) on the midplane of anterior and posterior polar cortices at the time of furrow formation (Fig. 2, A–E). Though 3D projected images had been used for representation purposes, the intensity was calculated from a single midplane z-section where the cortical intensity of the LIN-5 is usually highest. The integrated fluorescence intensity in this ROI and a similarly sized ROI in the cytoplasm and background were determined using Fiji/ImageJ. Fluorescence intensity at the anterior polar cortex and cytoplasm was subtracted from the background, and the ratio of the anterior polar cortex to the cytoplasm to indicate anterior polar cortical LIN-5 enrichment was used for plotting the graph. A similar strategy was applied to plot the graph for the posterior polar cortex.

### Online supplemental material

Fig. S1 shows cortical LIN-5 is essential for timely cleavage furrow formation, independent of its role in spindle elongation. Fig. S2 shows LIN-5 regulates the timing of cleavage furrow initiation independently of spindle positioning. Fig. S3 shows LIN-5 controls furrow initiation timing independently of NMY-2 removal from the polar cortex and promotes timely NMY-2 accumulation at the equatorial cortex. Fig. S4 shows the analysis of NMY-2 localization in *zyg-9(RNAi)* and *zyg-9(RNAi)* in LIN-5–inhibited embryos. This figure further shows LIN-5 localization in *zyg-9(RNAi)* condition. Fig. S5 shows LIN-5 promotes timely Anillin accumulation at the equatorial cortex. Fig. S6 shows the effects of PAR-5 depletion in the absence of LIN-5 on cleavage furrow formation. Fig. S7 shows the localization of midbody proteins in control and LIN-5– and SPD-1–inhibited embryos. Fig. S8 shows the relevance of GPR-1/2 in controlling proper abscission in the absence of SPD-1. Fig. S9 shows the working model summarizing the role of cortical LIN-5/GPR-1/2 in cleavage furrow formation and abscission. Videos 1 and 2 show GFP-NMY-2 localization without (Video 1) and with LIN-5 (Video 2) disruption at the cortical plane. Videos 3 and 4 show GFP-NMY-2 localization without (Video 3) and with LIN-5 (Video 4) disruption at the central plane. Videos 5 and 6 show mNG-ANI-1 localization without (Video 5) and with LIN-5 (Video 6) disruption at the cortical plane. Videos 7 and 8 show mNG-ANI-1 localization without (Video 7) and with LIN-5 (Video 8) disruption at the central plane. Videos 9, 10, 11, 12, and 13 show GFP^Mem./Tub.;mCherry^Chr./Cent. localization in control (Video 9), in *spd-1(ts)* background (Video 10), in *lin-5(RNAi)* (Video 11), in *spd-1(ts)* and *lin-5(RNAi)* (Video 12), and in *spd-1(ts)* and *gpr-1/2(RNAi)* (Video 13). Table S1 lists *C. elegans* strains. Table S2 lists details of RNAi clones, RNAi conditions, and primers used in the study.

### Data availability

All the data are available in the main text or the supplementary materials. Other materials generated in this manuscript are available from the corresponding author upon request and, when applicable, fulfillment of the appropriate Material Transfer Agreements.

## Acknowledgments

We thank Arshad Desai, Karen Oegema, Michael Glotzer, Esther Zanin, Amy Maddox, Masanori Mishima, Anthony Hyman, Marie Delattre, and Caenorhabditis Genetics Center (CGC) for sharing reagents and worm strains. We are grateful to Arshad Desai, Lionel Pintard, Martin Lowe, Andrew Goryachev, and K. Subramaniam for their suggestions on the manuscript. We thank DST-FIST, UGC Center for the Advanced Study, Department of Biotechnology-Indian Institute of Science (DBT-IISc) Partnership Program, and IISc for the infrastructure support.

This work is supported by the DBT grant (BT/PR36084/BRB/10/1857/2020) and the DST-SERB grant (CRG/2022/005151) to S. Kotak. Open Access funding provided by the Indian Institute of Science.

Author contributions: K. Adhikary: conceptualization, data curation, formal analysis, investigation, methodology, project administration, software, validation, visualization, and writing—review and editing. S. Kapoor: methodology and

writing—review and editing. S. Kotak: conceptualization, funding acquisition, project administration, resources, supervision, visualization, and writing—original draft, review, and editing.

Disclosures: The authors declare no competing interests exist.

Submitted: 12 June 2024

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

# Supplemental material

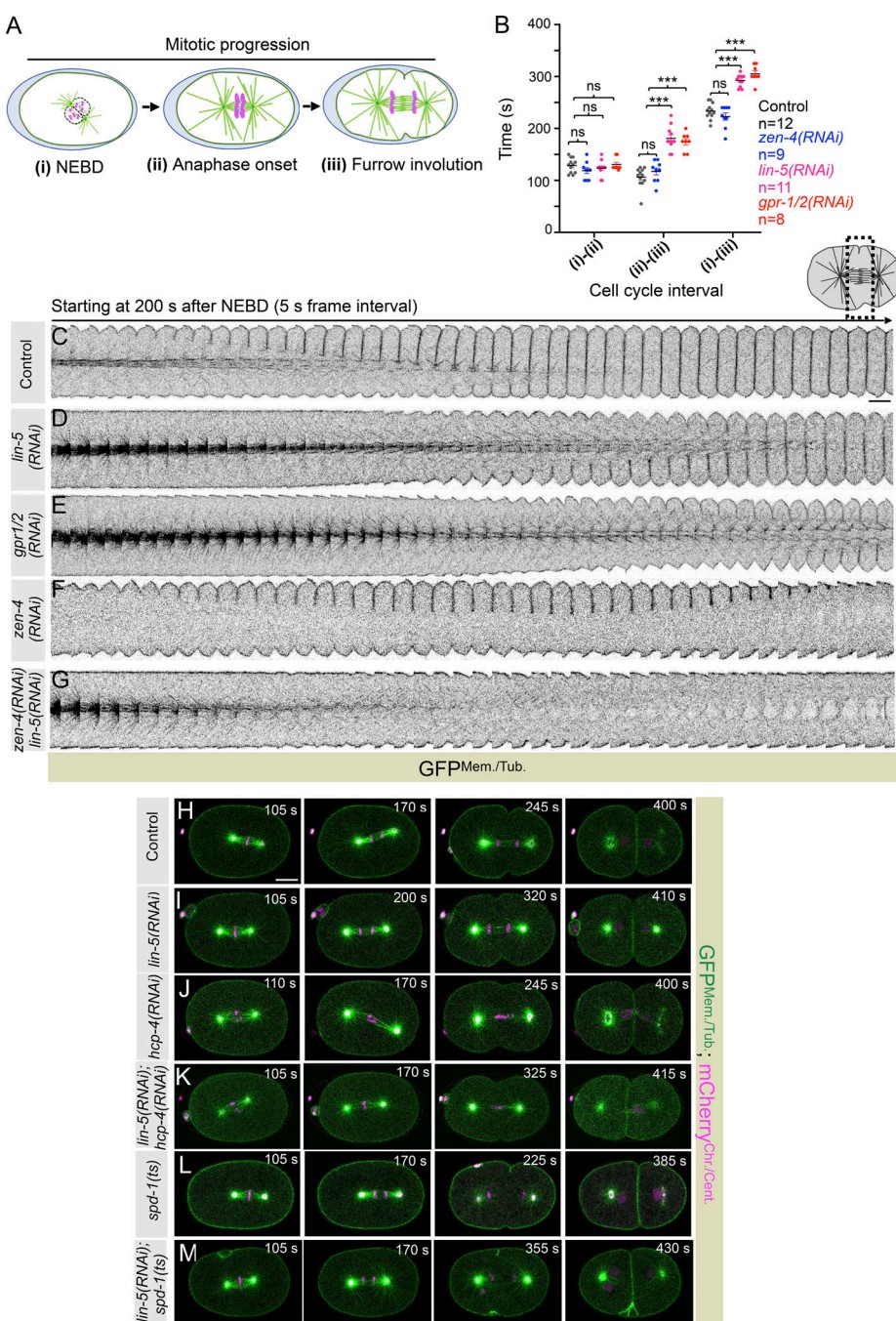

Figure S1. **LIN-5 regulates timely furrow induction independent of its role in spindle elongation. (A)** Schematics of the cell cycle progression, indicating various stages: NEBD (i), anaphase onset (ii), and furrow involution (iii). **(B)** Quantification of the cell cycle interval between the above-indicated stages (i), (ii), and (iii) from the confocal time-lapse movies of embryos coexpressing GFP-tagged PLCδ1-PH (membrane marker), GFP-TBB-2 (microtubule marker), mCherry-H2B (chromatin marker), and mCherry-γ-tubulin (centrosomal marker), referred to as GFP[Mem./Tub.];mCherry[Chr./Cent.] in control, *zen-4(RNAi)*, *lin-5(RNAi)*, *gpr-1/2(RNAi)* embryos, as indicated. The time interval between stages (i) and (ii) in control, *zen-4(RNAi)*, *lin-5(RNAi)*, and *gpr-1/2(RNAi)* embryos is nonsignificant (ns). However, there is a significant increase in a time interval between stages (ii) and (iii) of ~60 s in *lin-5(RNAi)* and ~70 s in *gpr-1/2(RNAi)* when compared to control and *zen-4(RNAi)*. Consequently, a similar delay persists from the stage (i) to (iii). *n* is the number of embryos analyzed. Error bars are the SEM; ns, P > 0.05; ***P < 0.001, as determined by two-tailed Student's *t* test. **(C–F)** Representative pseudokymographs of the division plane (shown on the top right) acquired by time-lapse confocal microscopy of control embryos coexpressing GFP[Mem./Tub.];mCherry[Chr./Cent.]. **(G)** These embryos were either left untreated (C), depleted for LIN-5 (D), depleted for GPR-1/2 (E), depleted for ZEN-4 (F), or codepleted for LIN-5 and ZEN-4 (G), as indicated. Please note re-use of the *lin-5(RNAi)* pseudokymograph in Fig. 1 D. Control, *n* = 17; *lin-5(RNAi)*, *n* = 11, *gpr-1/2(RNAi)*, *n* = 8, *zen-4(RNAi)*, *n* = 10, *lin-5(RNAi)*; *zen-4(RNAi)*, *n* = 10. For comparative analysis, only GFP[Mem./Tub.] localization is shown. Pseudokymographs start 200 s after NEBD, which was deduced by the entry of tubulin in the pronuclei. *n* is the number of embryos analyzed. The scale bar is 5 μm. **(H–M)** Representative images from the time-lapse confocal microscopy of embryo coexpressing GFP[Mem./Tub.];mCherry[Chr./Cent.] in control (H) and in various RNAi/mutant conditions, as indicated (I–M): control (*n* = 12); *lin-5(RNAi)* (*n* = 19); *hcp-4(RNAi)* (*n* = 13); *lin-5(RNAi)*; *hcp-4(RNAi)* (*n* = 11); *spd-1(ts)* (*n* = 13); *lin-5(RNAi)* in *spd-1(ts)* (*n* = 12). *n* is the number of embryos analyzed. In this and subsequent one-cell embryo images, the posterior is to the right. Time is in seconds (s) w.r.t. NEBD. The scale bar is 10 μm.

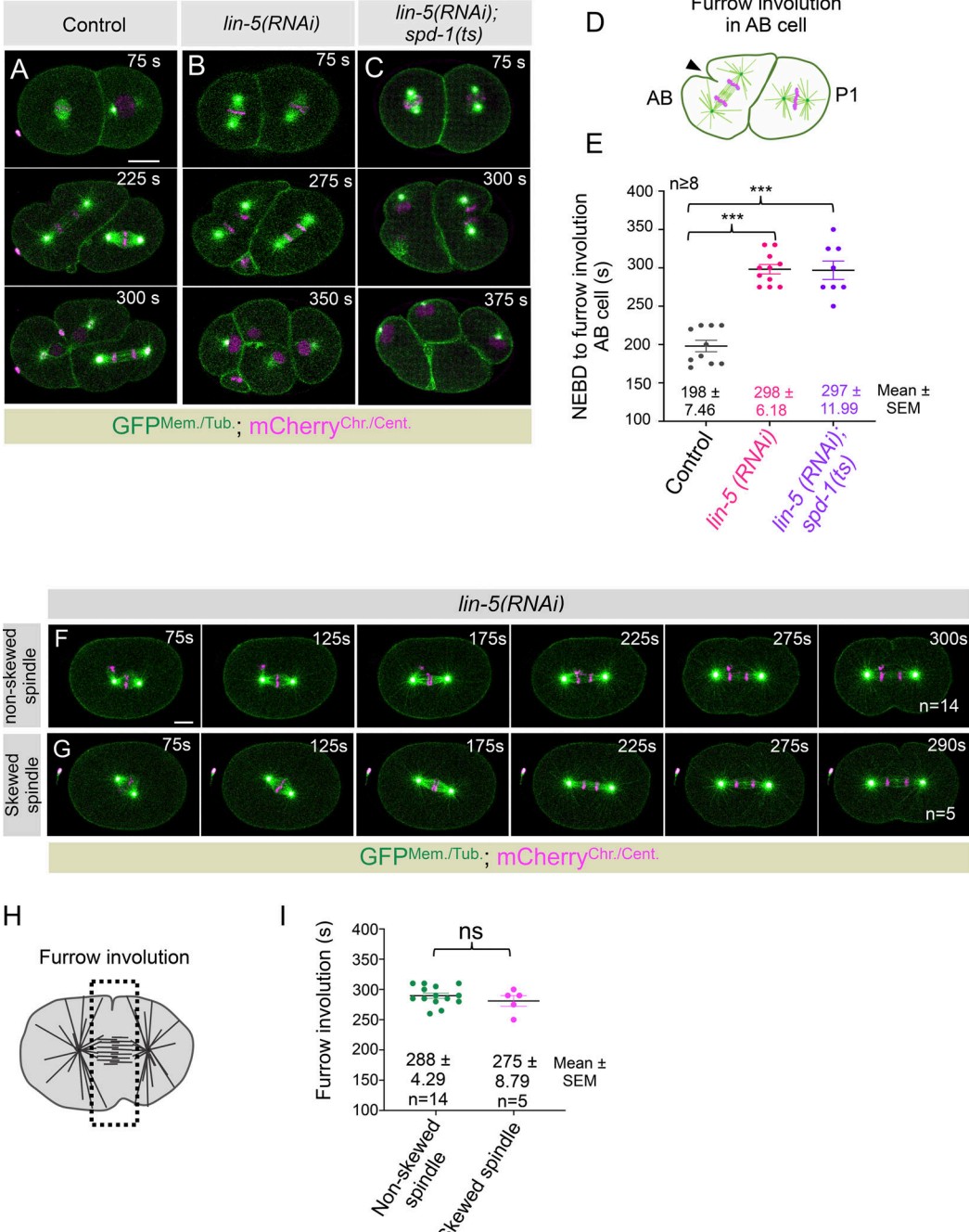

**Figure S2. LIN-5 controls furrow initiation timing independent of its function in spindle positioning. (A–C)** Representative images from the time-lapse confocal microscopy of two-cell stage embryo coexpressing GFP$^{Mem./Tub.}$;mCherry$^{Chr./Cent.}$ in control (A) and in *lin-5(RNAi)* (B), and *lin-5(RNAi)* in *spd-1(ts)* background (C). 75 s on the top panel represents the time w.r.t. NEBD. Note a delay in furrow involution in the anterior AB cell at the two-cell stage in *lin-5(RNAi)* and *lin-5(RNAi)* in *spd-1(ts)* embryos compared with control embryos. The scale bar is 10 µm. Since a significant percentage (~40%) of LIN-5–depleted embryos in *spd-1(ts)* show cytokinesis failure at the late stage after furrow ingression (see Fig. 5), this analysis was conducted in the rest of the embryos that did not show cytokinesis failure and progress to the two-cell-stage embryo. **(D and E)** Schematic represents the furrow involution in AB cell (D) and the quantification of the time interval between NEBD and furrow involution in embryos coexpressing GFP$^{Mem./Tub.}$;mCherry$^{Chr./Cent.}$ that are either left untreated or depleted for LIN-5, or LIN-5 depletion was performed in *spd-1(ts)* background (E), as indicated. Each dot represents one embryo, and *n* is the number of embryos analyzed. The solid black line on the graph represents the mean, whose values and SEM values are also mentioned at the bottom. ns, P > 0.05; ***P < 0.001, as determined by two-tailed Student's *t* test. **(F and G)** Representative images from the time-lapse confocal microscopy of the one-cell embryo co-expressing GFP$^{Mem./Tub.}$;mCherry$^{Chr./Cent.}$ in *lin-5(RNAi)*. The spindle was characterized into two categories based on orientation at anaphase: nonskewed spindle (F) and skewed spindle (G). 75 s on the left panel represents the time w.r.t. NEBD. The scale bar is 10 µm. **(H and I)** Schematic represents the furrow involution in P0 cell (H) and the quantification of the time interval between NEBD and furrow involution in embryos coexpressing GFP$^{Mem./Tub.}$;mCherry$^{Chr./Cent.}$ with nonskewed spindle and skewed spindle (I), as indicated. Each dot represents one embryo, and *n* is the number of embryos analyzed. The solid black line on the graph represents the mean, whose values and SEM values are also mentioned at the bottom. ns, P > 0.05; ***P < 0.001, as determined by two-tailed Student's *t* test.

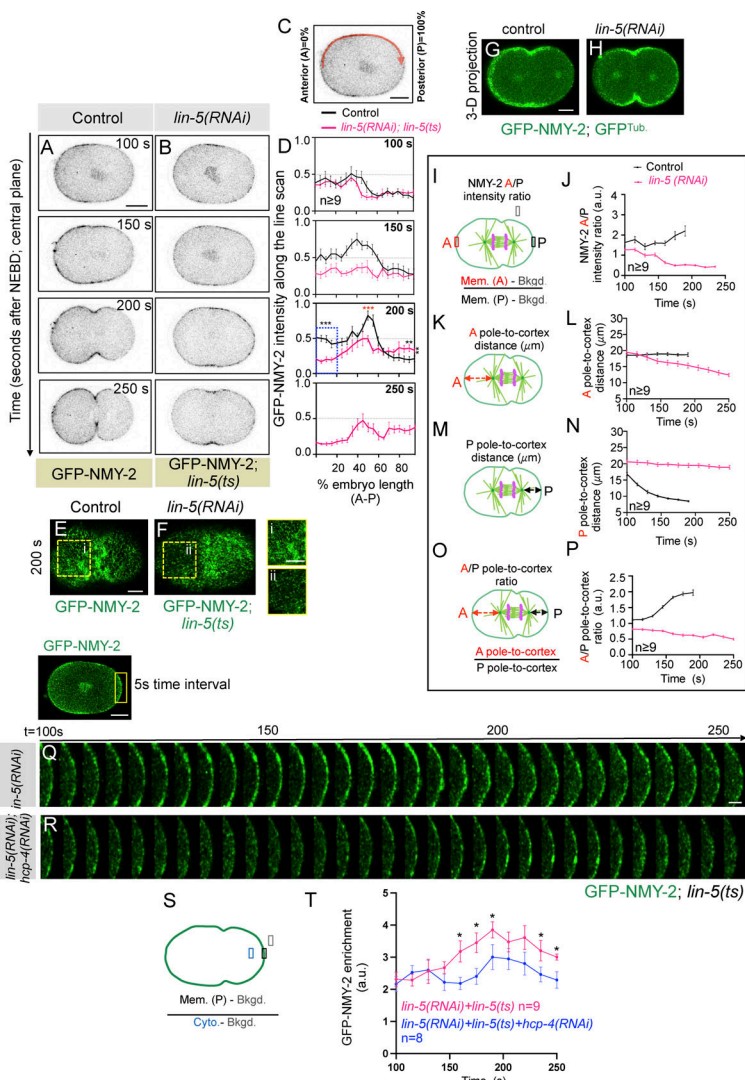

Figure S3. **LIN-5 controls furrow initiation timing independent of its function in removing NMY-2 from the polar cortical region. (A and B)** Representative images of the central plane from the time-lapse confocal microscopy of the one-cell embryo expressing endogenously tagged NMY-2 with GFP (GFP-NMY-2; in gray) in control (A) and LIN-5–inhibited [*lin-5(RNAi); lin-5(ts)*] (B) embryos starting from 100 s after NEBD, as indicated. Also, please see the corresponding Videos 3 and 4. Note that ~100% of *LIN-5–inhibited embryos* (*n* = 11) reveal much stronger NMY-2 clearance from the anterior cortical surface during anaphase at ~200 s in comparison with control embryos. Also, note a significantly weak accumulation of GFP-NMY-2 at the equatorial membrane at the time of furrow initiation in *LIN-5–inhibited embryos*. Time is w.r.t NEBD, deduced by the entry of the GFP-NMY-2 signal in the nucleus. The scale bar is 10 µm. **(C)** Schematic illustrating the method used to analyze cortical GFP-NMY-2 distribution. In brief, a one-pixel-wide line scan along the cell cortex was drawn from A to P. Embryos were divided into 20 equal-length segments from A (0%) to P (100%), which were used to calculate the mean cortical intensity (see Materials and methods for details). A: anterior; P: posterior. **(D)** Quantification indicates the mean cortical intensity of GFP-NMY-2 along the line scan (% embryo length A-P) at various time points (in seconds) after NEBD. Values were normalized by dividing by the average maximum values for controls. GFP-NMY-2 intensity is observed to be significantly decreased in *LIN-5–inhibited embryos* at 200 s on the anterior polar cortex relative to control embryos. Also, note that there is a significantly weak localization of GFP-NMY-2 at the equatorial membrane at 200 s and at 250 s after NEBD in *LIN-5–inhibited embryos* compared with the control embryos; also see Fig. 3 for cortical GFP-NMY-2 intensity measurements. *n* is the number of embryos analyzed. Error bars are SEM; ns, P > 0.05; *P < 0.05; **P < 0.01; ***P < 0.001, as determined by two-tailed Student's *t* test. **(E and F)** Examples of cortical GFP-NMY-2 accumulation in control (E) and *LIN-5–inhibited* (F) embryos at 200 s after NEBD at the anterior cell cortex. The scale bar is 10 µm. Boxed regions (i and ii) are the magnified view of the corresponding insets. Here, the scale bar is 5 µm. **(G and H)** 3D projected confocal images of one-cell embryo coexpressing GFP-NMY-2 and GFP[Tub.] (in green) in control (A) and *lin-5(RNAi)* (B) embryos. The scale bar is 10 µm. **(I–P)** Schematic of the method (I, K, M, and O) and the quantification (J, L, N, and P) for the A/P ratio of GFP-NMY-2 fluorescence intensity at the polar cortices (I and J); A pole-to-cortex distance (K and L); P pole-to-cortex distance (M and N), and the A-to-P pole-to-cortex distance ratio (O and P), as indicated. In this and other Fig. panels, Mem. represents membrane intensity, and Bkgd. represents background intensity. A: anterior; P: posterior. *n* is the number of embryos analyzed. Error bars are the SEM. **(Q and R)** Representative images of the posterior cortical region (shown on the top left) used for the pseudokymograph analysis from the time-lapse confocal microscopy of a one-cell embryo expressing endogenously tagged NMY-2 with GFP (GFP-NMY-2; in green) in *lin-5(ts)* mutant embryos and depleted for LIN-5 by RNAi (Q), or codepleted for LIN-5 and HCP-4. Time is w.r.t. NEBD. The scale bar is 10 and 5 µm for the embryo and pseudokymograph, respectively. **(S and T)** Schematic of the method used to quantify GFP-NMY-2 fluorescence intensity at the posterior cortex over time (S). Quantification of GFP-NMY-2 enrichment at the posterior cortex in embryos depleted for LIN-5 in *lin-5(ts)* background (pink line), as well as depleted for LIN-5 and HCP-4 in *lin-5(ts)* background (blue line). Time is w.r.t. NEBD. *n* is the number of embryos analyzed. Error bars, SEM. ns, P > 0.05; *P < 0.05, as determined by two-tailed Student's *t* test.

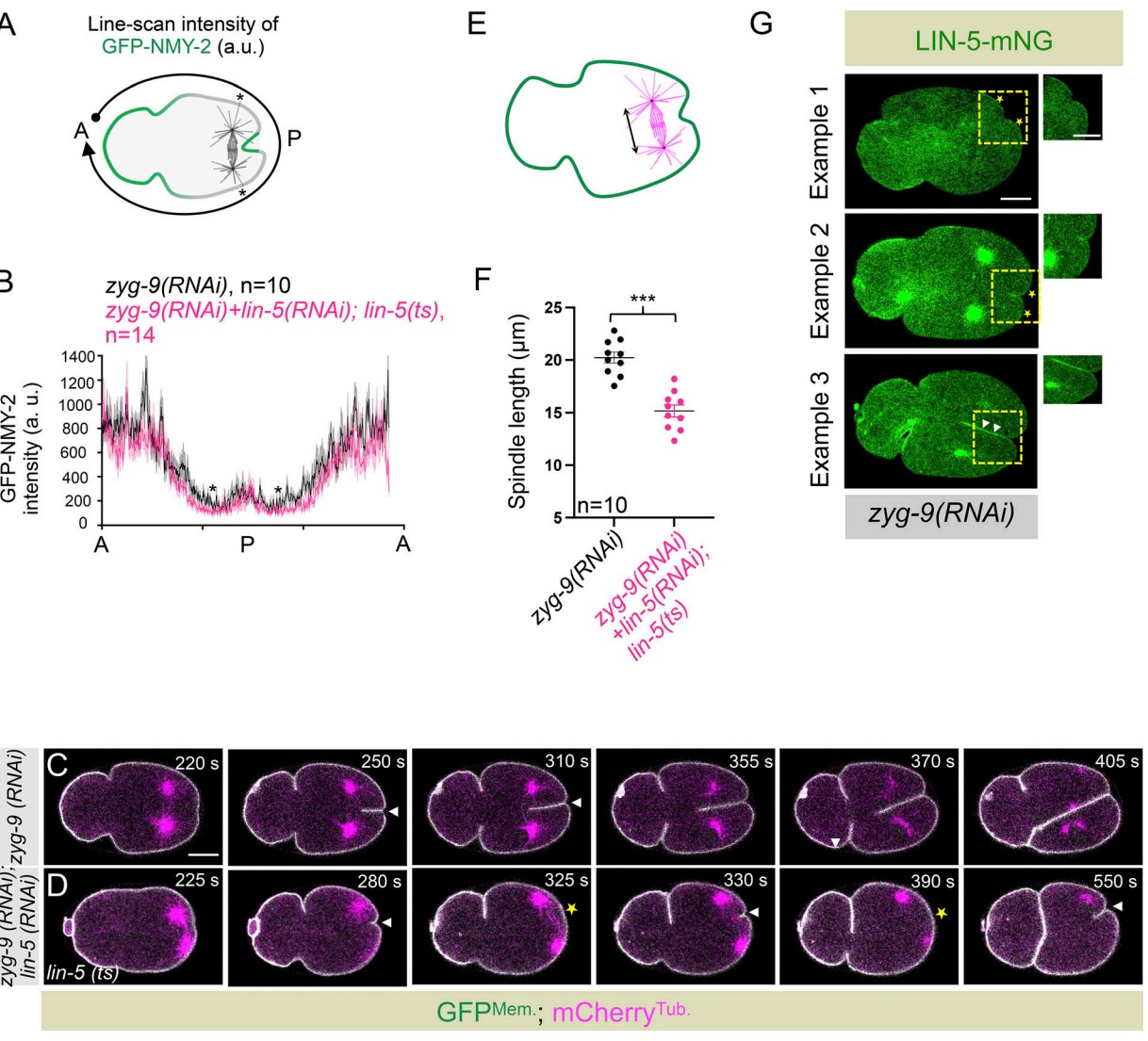

Figure S4. **NMY-2 membrane stripping by LIN-5 is not a critical determinant for cleavage furrow induction. (A and B)** Schematic of the method to quantify the line scan intensity of GFP-NMY-2 (A) and its quantification along the A-P-A axis (B), as indicated. Black asterisks in A and B indicate NMY-2 clearance at the posterior cortical membrane near the posterior furrow involution region. **(C and D)** Representative images from the time-lapse confocal microscopy of embryos coexpressing GFP$^{Mem.}$ and mCherry$^{Tub.}$ in *zyg-9(RNAi)* (C), and *zyg-9(RNAi)* and *lin-5(RNAi)* in the *lin-5(ts)* background (D). White arrowheads indicate furrow, and yellow asterisks indicate the dissolution of the furrow in *zyg-9(RNAi)* embryos inhibited for LIN-5. Note the re-use of the *zyg-9(RNAi)* embryo image shown in Fig. S4C for Fig. 3 K. The scale bar is 10 µm. **(E and F)** Schematic of the method used to measure the pole-to-pole distance at the time of furrow initiation in embryos coexpressing GFP$^{Mem.}$ and mCherry$^{Tub.}$ (E). Quantification of pole–pole distance at the time of furrow onset/involution (F) in *zyg-9(RNAi)* and in *zyg-9(RNAi)* and *lin-5(RNAi)* in the *lin-5(ts)* background, as indicated. Each dot represents one embryo, and *n* is the number of embryos analyzed. Mean is represented in the black line. Error bars, SEM. ns, P > 0.05; ***P < 0.001, as determined by two-tailed Student's *t* test. **(G)** Representative images from the time-lapse confocal microscopy of embryos endogenously expressing LIN-5-mNG in *zyg-9(RNAi)*; asterisks indicate LIN-mNG signal at the posterior cortex, and white arrowheads indicate the presence of cortical LIN-5-mNG at the invaginating furrow. The scale bar is 10 µm. Boxed regions are the magnified view of the corresponding inset. The scale bar for the inset is 5 µm.

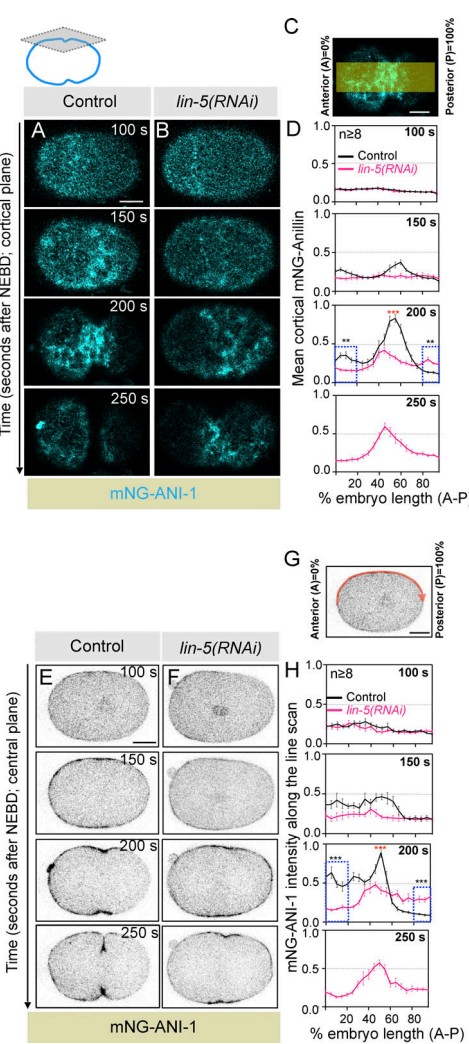

Figure S5.   **LIN-5 controls the timely enrichment of Anillin at the equatorial cortex. (A and B)** Representative images of the cortical plane from the time-lapse confocal microscopy of one-cell embryo coexpressing endogenously tagged Anillin with mNG (mNG-ANI-1; in cyan) and NMY-2 with mKate. Note that only the mNG-ANI-1 signal was imaged using confocal microscope as it was the brightest for live recording. Images show control (A) and *LIN-5–depleted* embryos (B) starting from 100 s after NEBD, as indicated. Also, see corresponding Videos 5 and 6. Note the significantly better clearance of the mNG-ANI-1 signal at the anterior cortex in LIN-5–depleted embryos at the time of furrow initiation w.r.t. control embryos. Also note a weak accumulation of mNG-ANI-1 signal at the equatorial membrane in the *lin-5(RNAi)* embryo compared with control. Time is w.r.t NEBD, deduced by studying the entry of the mNG-ANI-1 signal in the nucleus (not shown). The scale bar is 10 μm. **(C)** Schematic illustrating the method used to analyze cortical mNG-ANI-1 distribution, as performed by Lewellyn et al. (2010). In brief, a 50-pixel-wide line (~1/2 the embryo width) was drawn, and embryos were divided into 20 equal-length segments from A (0%) to P (100%) to calculate the mean cortical accumulation of mNG-ANI-1 along the A-P axis (see Materials and methods for details). A: anterior; P: posterior. **(D)** Quantification indicates the mean cortical accumulation of mNG-ANI-1 intensity along the A-P axis (% embryo length A-P) at various time points (in seconds) after NEBD. Values were normalized by dividing by the average maximum values for controls. Control embryos are shown in a black line, and LIN-5–depleted embryos are indicated by a pink line. mNG-ANI-1 intensity is observed to be significantly decreased at 200 s on the anterior polar cortex in LIN-5–depleted embryos relative to control embryos. Also note that there is a significantly weak localization of mNG-ANI-1 at the equatorial cortical region at 200 s and at 250 s after NEBD in *lin-5(RNAi)* embryos in comparison with the control embryos. *n* is the number of embryos analyzed. Error bars are SEM; ns, P > 0.05; *P < 0.05; **P < 0.01; ***P < 0.001, as determined by two-tailed Student's *t* test. **(E and F)** Representative images of the central plane from the time-lapse confocal microscopy of the one-cell embryo coexpressing endogenous tagged mNG-ANI-1 (in gray) and NMY-2 with mKate. Note that only the mNG-ANI-1 signal was imaged using confocal microscope as it was the brightest for live recording. Images show control (E) and *LIN-5–depleted* embryos (F) starting from 100 s after NEBD, as indicated. Also, please see the corresponding Videos 7 and 8. Note the significantly better clearance of the mNG-ANI-1 signal at the anterior cortex in LIN-5–depleted embryos at the time of furrow initiation w.r.t. control embryos. Also, note a significant accumulation of mNG-ANI-1 signal at the equatorial membrane during furrow initiation in control embryo w.r.t. LIN-5–depleted embryos. Time is w.r.t NEBD, deduced by the entry of the mNG-ANI-1 signal in the nucleus. The scale bar is 10 μm. **(G)** Schematic illustrating the method used to analyze cortical mNG-ANI-1 distribution. In brief, a one-pixel-wide line scan along the cell cortex was drawn from A to P. Embryos were divided into 20 equal-length segments from A (0%) to P (100%), which were used to calculate the mean cortical intensity (see Materials and methods for details). A: anterior; P: posterior. **(H)** Quantification indicates the mean cortical intensity of mNG-ANI-1 along the line scan (% embryo length A-P) at various time points (in seconds) after NEBD. Values were normalized by dividing by the average maximum values for controls. Control embryos are shown by the black line, and LIN-5–depleted embryos are shown by the pink line. mNG-ANI-1 intensity is significantly decreased at 200 s on the anterior polar cortex relative to control embryos. Also note that there is a significantly weak localization of mNG-ANI-1 at the equatorial cortical region at 200 s and at 250 s after NEBD in *lin-5(RNAi)* embryos in comparison with the control embryos. *n* is the number of embryos analyzed. Error bars are the SEM; ns, P > 0.05; *P < 0.05; **P < 0.01; ***P < 0.001, as determined by two-tailed Student's *t* test.

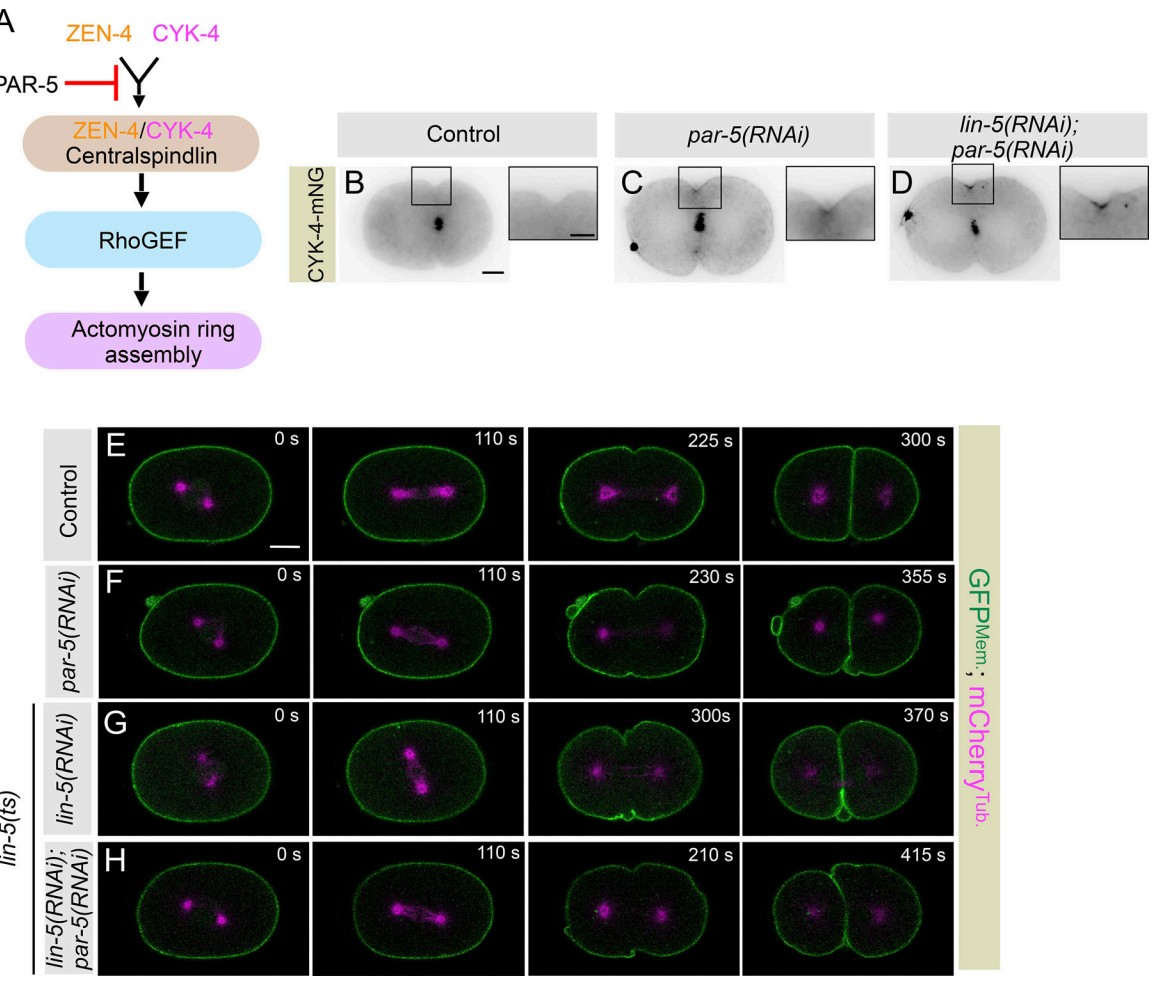

Figure S6. **PAR-5 depletion rescues the furrow initiation delay seen in *lin-5(RNAi)* embryos. (A)** Schematic representation of an evolutionarily conserved pathway required for actomyosin ring assembly and the role of PAR-5 in negatively regulating ZEN-4 and CYK-4 oligomerization. **(B–D)** Representative images from the time-lapse confocal microscopy of embryo coexpressing CYK-4-mNG in control (*n* = 5) (B) and *par-5(RNAi)* (*n* = 5) (C), and *par-5(RNAi)*; *lin-5(RNAi)* (*n* = 11) (D). *n* is the number of embryos analyzed. The scale bar is 10 μm. Insets with the corresponding images show the magnified view of the furrow region. Here, the scale bar is 5 μm. **(E–H)** Representative images acquired by time-lapse confocal microscopy of control embryos coexpressing GFP^Mem. and mCherry^Tub., which are either left untreated (E), or depleted for PAR-5 (F), *lin-5(ts)* embryos expressing GFP^Mem. and mCherry^Tub. that are treated with *lin-5(RNAi)* (G), or *lin-5(ts)* embryos expressing GFP^Mem. and mCherry^Tub. that are treated with *lin-5(RNAi)* and *par-5(RNAi)* (H), as indicated. Control (*n* = 17); *par-5(RNAi)* (*n* = 10); *lin-5(ts)+lin-5(RNAi)* (*n* = 10); *lin-5 (ts)+lin-5(RNAi)+par-5 (RNAi)* (*n* = 12). 0 s represents NEBD. *n* is the number of embryos analyzed. The scale bar is 10 μm.

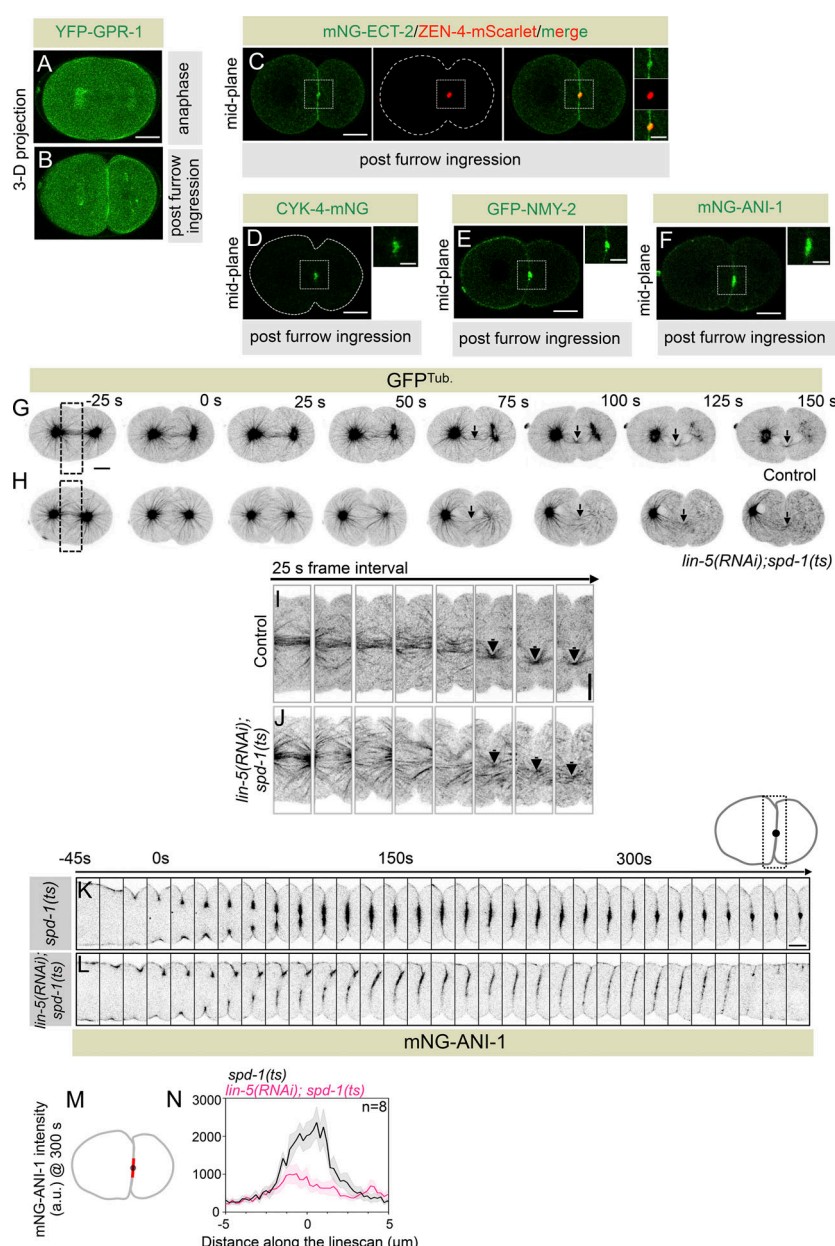

Figure S7. **Cortical accumulation of LIN-5 after furrow ingression is essential for the maintenance of midbody-localized proteins. (A and B)** 3D projected confocal images of a one-cell embryo expressing GPR-1 tagged with YFP (YFP-GPR-1) during anaphase (A) and after furrow ingression (B). More than 13 recordings were made, and the representative images are shown here. Please note the robust localization of the YFP-GPR-1 signal at the juxtaposed membrane after furrow ingression. The scale bar is 10 μm. **(C–F)** Representative midplane confocal images of embryos coexpressing mNG-ECT-2 (in green) and ZEN-4-mScarlet (in red) after furrow ingression (C). Similarly, confocal microscopy images of embryos expressing CYK-4-mNG (D), GFP-NMY-2 (E), and mNG-ANI-1 (F) after furrow ingression. Insets of the corresponding images at the midbody region are also shown. More than eight embryos were imaged, and the representative embryos are shown in the figure panel. The scale bar is 10 and 5 μm for the representative embryo images and corresponding insets, respectively. **(G and H)** Representative time-lapse images of GFP^Tub.-expressing (inverted grayscale) embryos revealing bundling of astral microtubules (black arrow) in control (G) and in *spd-1(ts)* mutant embryos that are depleted for LIN-5 with failed abscission (H). The representative images are from −25 s w.r.t furrow involution, which is 0 s. Black dotted insets represent regions tracked for pseudokymograph analysis in I and J (see below). The scale bar is 10 μm. **(I and J)** Pseudokymograph analysis of astral microtubule bundling in control (I) and in *spd-1(ts)* mutant embryos that are treated with *lin-5(RNAi)* with failed abscission (J). Black arrows indicate astral microtubule bundling. Maximum projection of three z-sections is shown for *lin-5(RNAi); spd-1(ts)*. Scale bar, 10 μm. **(K and L)** Representative pseudokymographs of the division plane (depicted on the top right) acquired by time-lapse confocal microscopy of control embryos expressing mNG-ANI-1 in *spd-1(ts)* embryos (K), or mNG-ANI-1 in *spd-1(ts)* embryos treated with *lin-5(RNAi)* (L). The pseudokymograph begins at −45 s w.r.t furrow involution, which is 0 s. The time interval in between each frame is 15 s. Please note that the mNG-ANI-1 is significantly enriched at the midbody after furrow ingression; in contrast, mNG-ANI-1 intensity spreads along the membrane in *spd-1(ts)* embryos treated with *lin-5(RNAi)* embryos. The scale bar is 5 μm. **(M and N)** Schematic of the method used to quantify mNG-ANI-1 fluorescence intensity at the midbody during 300 s after furrow involution. The red line drawn on the midbody represents the one-pixel-wide line scan of 10 μm (M). mNG-ANI-1 fluorescence intensity at 300 s in *spd-1(ts)* mutant embryos (black line), in *spd-1(ts)* mutant embryos that are treated with *lin-5(RNAi)* (pink line) (N). *n* is the number of embryos analyzed. Error bars shown by the shaded area are the SEM.

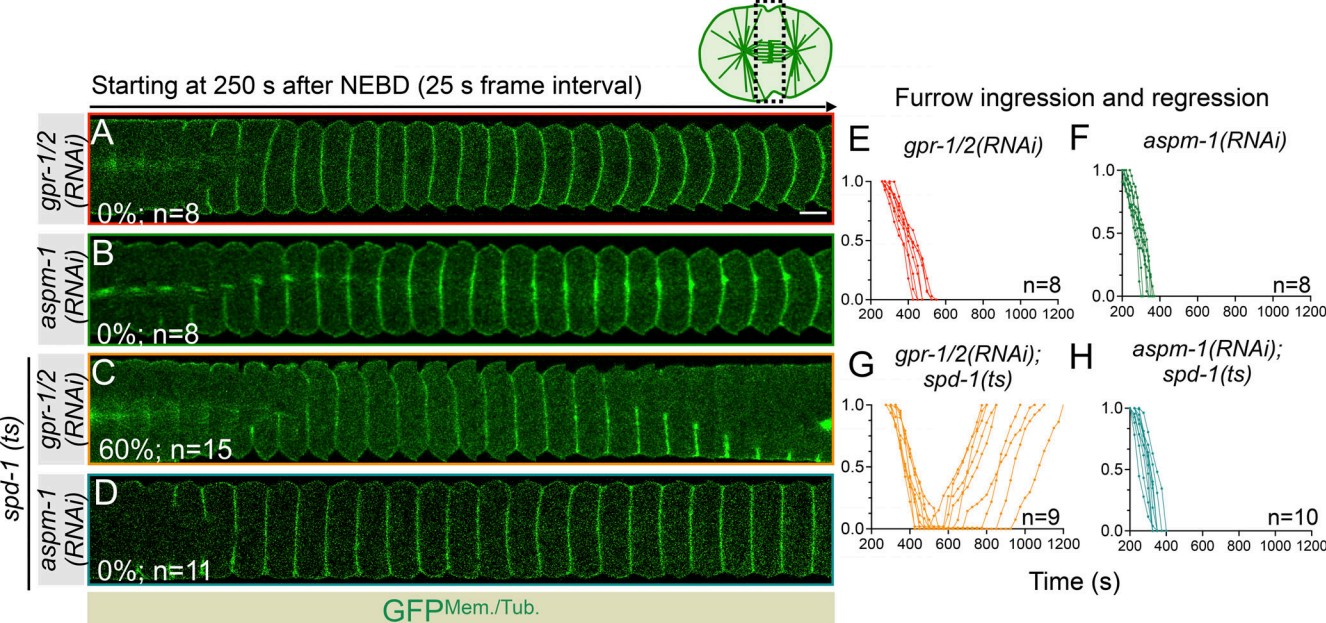

Figure S8. **GPR-1/2 is critical for regulating proper abscission after furrow ingression. (A–D)** Representative pseudokymographs of the division plane (depicted on the top right) acquired by time-lapse confocal microscopy of embryos coexpressing GFP[Mem./Tub.];mCherry[Chr./Cent.] that are treated with *gpr-1/2(RNAi)* (A), *aspm-1(RNAi)* (B), *spd-1(ts)* embryos that are treated with *gpr-1/2(RNAi)* (C), or *spd-1(ts)* embryos that are treated with *aspm-1(RNAi)* (D), as indicated. For comparative analysis, only GFP[Mem./Tub.] localization is shown. *gpr-1/2(RNAi)* (n = 8); *aspm-1 (RNAi)* (n = 8); *gpr-1/2(RNAi)* in *spd-1(ts)* background (n = 15) (also see corresponding Video 13); *aspm-1(RNAi)* in *spd-1(ts)* background (n = 11). *n* is the number of embryos analyzed. Note that *aspm-1(RNAi)* embryos in *spd-1(ts)* background do not show any abscission failure; however, *spd-1(ts)* embryos that are treated with *gpr-1/2(RNAi)* show 60% abscission failure. The pseudo-kymograph starts at 250 s w.r.t NEBD. The scale bar is 5 μm. **(E–H)** Graphs represent furrow ingression kinetics of embryo coexpressing GFP[Mem./Tub.];mCherry[Chr./Cent.] in *gpr-1/2(RNAi)* embryos (E), *aspm-1(RNAi)* embryos (F), *gpr-1/2(RNAi)* embryos in *spd-1(ts)* mutant background (G), and *aspm-1(RNAi)* embryos in *spd-1(ts)* mutant background (H). Please note furrow ingression followed by regression in *spd-1(ts)* mutant embryos that are depleted for GPR-1/2. A similar observation was made for *spd-1(ts)* mutant embryos depleted for LIN-5 (see Fig. 5). *n* is the number of embryos analyzed. Time is w.r.t. NEBD.

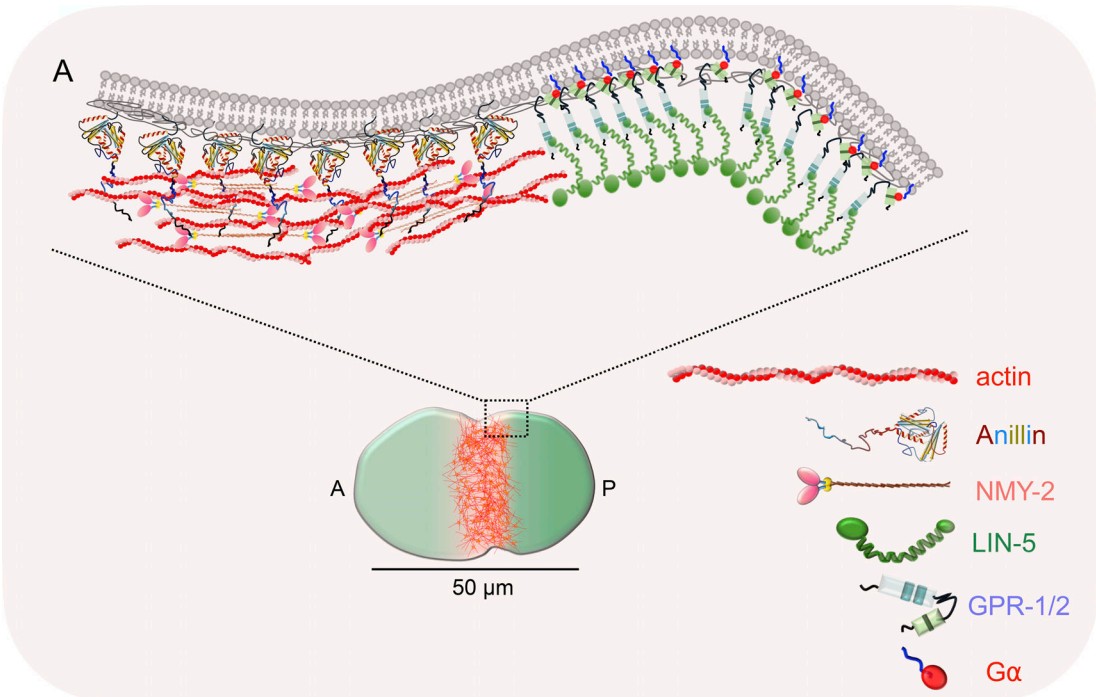

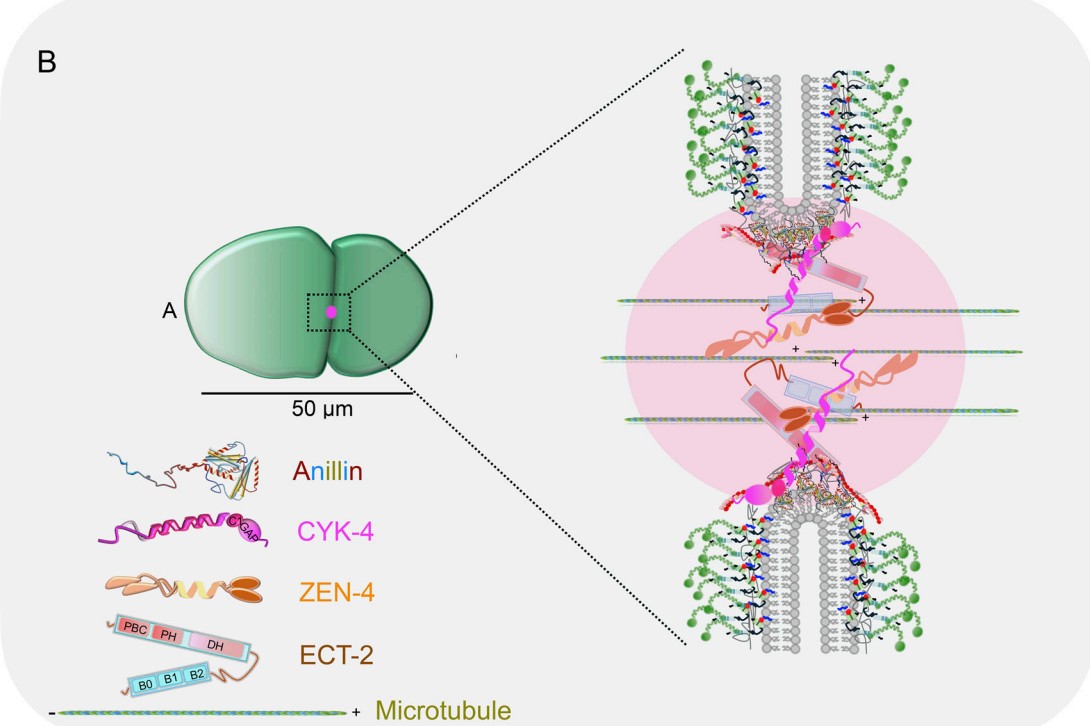

Figure S9.  **Membrane pools of LIN-5/GPR-1/2 complexes regulate cleavage furrow formation and abscission. (A)** Model demonstrating that enriched pools of LIN-5/GPR-1/2 via membrane binding property of Gα at the polar region and lateral region of the cell cortex confine the timely accumulation of actomyosin and Anillin at the equatorial cortical region for timely cytokinetic furrow formation. Inset highlights the equatorial membrane region where actin, NMY-2, and Anillin accumulate and regulate furrow formation. In contrast, LIN-5/GPR-1/2/Gα complexes accumulate at the polar and lateral membrane surface. **(B)** Model showing that membrane-enriched pools of LIN-5 and GPR-1/2-based complexes after furrow ingression in the vicinity of the midbody and its membrane regulate the confinement of midbody/midbody membrane–localized proteins, including centralspindlin components—CYK4 and Anillin. Inset highlighting the two juxtaposed equatorial membrane after furrow ingression enriches robust levels of cortical LIN-5/GPR-1/2 pools at the membrane near the midbody and surrounding midbody membrane. The midbody and the surrounding midbody membrane are occupied by microtubules, ECT-2/CYK-4/ZEN-4, Anillin, and other proteins (SPD-1, Aurora B, etc., not shown). Our work indicates that cortical LIN-5/GPR-1/2 after furrow ingression redundantly works microtubule-bundling protein SPD-1 in ensuring the stability of the midbody and thus abscission zone. In the absence of SPD-1, contractile ring proteins are restricted to the midbody and midbody surrounding membrane because of the presence of LIN-5/GPR-1/2 in the proximal region.

Video 1.   **Time-lapse confocal microscopy of the cortical plane of control one-cell *C. elegans* embryos expressing endogenously tagged GFP-NMY-2 (in green).** The movie was acquired at 1 frame every 5 s and played at 7 frames/s. Time is in seconds w.r.t. NEBD. Scale bar, 10 μm. This video corresponds to the images shown in Fig. 3 A.

Video 2.   **Time-lapse confocal microscopy of the cortical plane of LIN-5–depleted one-cell *C. elegans* embryos expressing endogenously tagged GFP-NMY-2 (in green) in *lin-5(ts)* background.** The movie was acquired at 1 frame every 5 s and played at 7 frames/s. Please note that in contrast to the control (Video 1), robust NMY-2 clearance was observed during anaphase at the anterior. Also note that in LIN-5–inhibited embryos, the equatorial cortical intensity of GFP-NMY-2 is significantly reduced, as described in the text. Time is in seconds w.r.t. NEBD. Scale bar, 10 μm. This video corresponds to the images shown in Fig. 3 B.

Video 3.   **Time-lapse confocal microscopy of the central plane of control one-cell *C. elegans* embryos expressing endogenously tagged GFP-NMY-2 (inverted grayscale).** The movie was acquired at 1 frame every 5 s and played at 7 frames/s. Time is in seconds w.r.t. NEBD. Scale bar, 10 μm. This video corresponds to the images shown in Fig. S3 A.

Video 4.   **Time-lapse confocal microscopy of the central plane of LIN-5–depleted one-cell *C. elegans* embryos expressing endogenously tagged GFP-NMY-2 (inverted grayscale) in *lin-5(ts)* background.** The movie was acquired at 1 frame every 5 s and played at 7 frames/s. Please note that in contrast to the control (Video 3), robust NMY-2 clearance was observed during anaphase at the anterior. Also note that in LIN-5–inhibited embryos, the equatorial cortical intensity of GFP-NMY-2 is significantly reduced, as described in the text. Time is in seconds w.r.t. NEBD. Scale bar, 10 μm. This video corresponds to the images shown in Fig. S3 B.

Video 5.   **Time-lapse confocal microscopy of the cortical plane of control one-cell *C. elegans* embryos expressing endogenously tagged ANI-1 with mNG (in cyan).** The movie was acquired at 1 frame every 5 s and played at 7 frames/s. Time is in seconds w.r.t. NEBD. Scale bar, 10 μm. This video corresponds to the images shown in Fig. S5 A.

Video 6.   **Time-lapse confocal microscopy of the cortical plane of control one-cell *C. elegans* embryos expressing endogenously tagged mNG-ANI-1 (in cyan) and depleted for LIN-5.** The movie was acquired at 1 frame every 5 s and played at 7 frames/s. Please note that in contrast to the control (Video 5), the equatorial cortical intensity of mNG-Anillin is significantly reduced, as described in the text. Time is in seconds w.r.t. NEBD. Scale bar, 10 μm. This video corresponds to the images shown in Fig. S5 B.

Video 7.   **Time-lapse confocal microscopy of the central plane of control one-cell *C. elegans* embryos expressing endogenously tagged mNG-ANI-1 (inverted grayscale).** The movie was acquired at 1 frame every 5 s and played at 7 frames/s. Time is in seconds w.r.t. NEBD. Scale bar, 10 μm. This video corresponds to the images shown in Fig. S5 E.

Video 8.   **Time-lapse confocal microscopy of the central plane of control one-cell *C. elegans* embryos expressing endogenously tagged mNG-ANI-1 (inverted grayscale) and depleted for LIN-5.** The movie was acquired at 1 frame every 5 s and played at 7 frames/s. Please note that in contrast to the control (Video 7), the equatorial intensity of mNG-Anillin is significantly reduced, as described in the text. Time is in seconds w.r.t. NEBD. Scale bar, 10 μm. This video corresponds to the images shown in Fig. S5 F.

Video 9.   **Time-lapse confocal microscopy of control one-cell *C. elegans* embryos expressing GFP$^{Mem./Tub.}$;mCherry$^{Chr./Cent.}$.** The movie was acquired at 1 frame every 25 s, played at 9 frames/s. Time is in seconds w.r.t. anaphase onset. Scale bar, 10 μm. This video corresponds to the images shown in Fig. 5 N.

Video 10.   **Time-lapse confocal microscopy of one-cell *C. elegans* embryos expressing GFP$^{Mem./Tub.}$;mCherry$^{Chr./Cent.}$ in *spd-1(ts)* mutant background at a restrictive temperature.** The movie was acquired at 1 frame every 25 s, played at 9 frames/s. Time is in seconds w.r.t. anaphase onset. Scale bar, 10 μm. This video corresponds to the images shown in Fig. 5 O.

Video 11.   **Time-lapse confocal microscopy of one-cell *C. elegans* embryos expressing GFP$^{Mem./Tub.}$;mCherry$^{Chr./Cent.}$ and depleted for LIN-5.** The movie was acquired at 1 frame every 25 s, played at 9 frames/s. Time is in seconds w.r.t. anaphase onset. Scale bar, 10 μm. This video corresponds to the images shown in Fig. 5 P.

Video 12.   **Time-lapse confocal microscopy of one-cell *C. elegans* embryos expressing GFP$^{Mem./Tub.}$;mCherry$^{Chr./Cent.}$ in *spd-1(ts)* background and depleted for LIN-5 at a restrictive temperature.** The movie was acquired at 1 frame every 25 s, played at 9 frames/s. Time is in seconds w.r.t. anaphase onset. Please note the abscission failure as described in the text. Time is in seconds w.r.t. anaphase onset. Scale bar, 10 μm. This video corresponds to the images shown in Fig. 5 Q.

Video 13.   **Time-lapse confocal microscopy of one-cell *C. elegans* embryos expressing GFP$^{Mem./Tub.}$;mCherry$^{Chr./Cent.}$ in *spd-1(ts)* background and depleted for GPR-1/2 at restrictive temperature.** The movie was acquired at 1 frame every 25 s, played at 9 frames/s. Time is in seconds w.r.t. anaphase onset. Please note the abscission failure as described in the text. Time is in seconds w.r.t. anaphase onset. Scale bar, 10 μm. This video corresponds to the images shown in Fig. S8 C.

**Provided online are Table S1 and Table S2. Table S1 lists *C. elegans* strains. Table S2 shows bacterial feeding strains and primers used to design RNAi clone.**

