## [Peer Review File · The Journal of Cell Biology]

A cortical pool of LIN-5 (NuMA) controls cytokinetic furrow formation and cytokinesis completion

Kuheli Adhikary, Sukriti Kapoor, and Sachin Kotak

Corresponding Author(s): Sachin Kotak, Indian Institute of Science Bangalore

Review Timeline:

Submission Date:	2024-06-12
Editorial Decision:	2024-08-18
Revision Received:	2025-02-26
Editorial Decision:	2025-04-04
Revision Received:	2025-04-10

Monitoring Editor: Karen Oegema

Scientific Editor: Dan Simon

Transaction Report:

DOI: <https://doi.org/10.1083/jcb.202406059>

August 18, 2024

Re: JCB manuscript #202406059

Prof. Sachin Kotak
Indian Institute of Science Bangalore
Microbiology and Cell Biology (MCB)
CV Raman Avenue
Bangalore, Karnataka 560012
Iceland

Dear Prof. Kotak,

Thank you for submitting your manuscript entitled "A direct role for LIN-5 (NuMA) in control of cytokinetic furrow formation and cytokinesis completion." The manuscript has been evaluated by expert reviewers, whose reports are appended below. Unfortunately, after an assessment of the reviewer feedback, our editorial decision is against publication in JCB.

Although both reviewers were interested in the relationship between LIN-5 and furrow ingression they both felt that the work fell short of explaining why LIN-5 depletion delays furrow ingression. Both reviewers felt that the idea that LIN-5 limits the spreading of ring components away from the equator was not sufficiently supported. Reviewer 2 additionally asked whether a mechanical explanation (initial set up of the spindle in a skewed orientation followed by subsequent rotation onto the long axis) might be the cause of the delay, since it would take additional time for cortical patterning and furrow formation to reinitiate after the spindle moves into place.

While your manuscript is intriguing, we feel that addressing the reviewer comments and arriving at a convincing mechanistic explanation would require a more substantial amount of new experiments than can be done in a typical revision period. If you wish to expedite publication of the current data, it may be best to pursue publication at another journal.

However, given interest in the topic, we would be open to resubmission to JCB of a significantly revised and extended manuscript that fully addresses the reviewers' concerns and is subject to further peer-review. If you would like to resubmit this work to JCB, please contact the journal office to discuss an appeal of this decision or you may submit an appeal directly through our manuscript submission system. Please note that priority and novelty would be reassessed at resubmission.

Regardless of how you choose to proceed, we hope that the comments below will prove constructive as your work progresses. We would be happy to discuss the reviewer comments further once you've had a chance to consider the points raised in this letter. You can contact the journal office with any questions at cellbio@rockefeller.edu.

Thank you for thinking of JCB as an appropriate place to publish your work.

Sincerely,

Karen Oegema, PhD
Monitoring Editor
Journal of Cell Biology

Dan Simon, PhD
Scientific Editor
Journal of Cell Biology

Reviewer #1 (Comments to the Authors (Required)):

During cytokinesis an equatorial zone of active RhoA induces cleavage furrow formation. LIN-5/NUMA is enriched at the poles of the cells and is an anchor for dynein with a well-established role in spindle positioning. Due to its polar enrichment, it seems plausible that LIN-5/NUMA not only controls pulling forces on the spindle but also regulates contractility during cytokinesis. In a previous publication the lab had shown in human cells that LIN-5/NUMA limits the accumulation of RhoA at the cell poles and promotes the enrichment of RhoA at the cell equator, possibly by restricting the Ect2/Centralspindlin complex to the equator (Sana JCB 2022). In the presented manuscript the authors investigate whether LIN-5 has a similar function in furrow formation independent of its role in spindle positioning in *C. elegans*.

The authors show that LIN-5 depleted embryos exhibit a delay in furrow formation even when the aster separation defect is rescued. Furthermore, reduction of LIN-5 levels at the posterior pole by PAR-2 depletion also causes a similar delay in furrow formation. Together those observations suggests that LIN-5 has a direct role in forming the cytokinetic furrow. Next the authors show that cortical NMY-2 levels are increased at the posterior and decreased at the anterior and equator at the time point of furrow involution. To analyze the effect of LIN-5 depletion on NMY-2 accumulation independently of aster positioning a *zyg-9(RNAi)* background is used. In *zyg-9(RNAi)* embryos co-depleted of LIN-5 accumulation of NMY-2 at the furrow site is compromised, but clearing above the asters is normal. Together the authors propose that polar LIN-5 confines NMY-2 at the equatorial region and thereby promotes proper accumulation at the equator. Lastly, in *lin-5(RNAi)* embryos CYK-4 strongly accumulates at the spindle midzone probably due to reduced aster separation. However, in the absence of SPD-1 and LIN-5 late accumulation (300s) of CYK-4 fails suggesting a role of LIN-5 in stabilizing CYK-4 at the midzone region. Importantly, most LIN-5 phenotypes are also exhibited after *grp-1/2(RNAi)* which causes a loss of LIN-5 from the membrane but not the asters. Suggesting that the membrane localized pool of LIN-5 promotes timely furrowing.

Based on the data the authors propose that polar enriched LIN-5 limits the spreading of ring components and CYK-4 away from the cell equator and thereby contributes to their equatorial enrichment. Overall, the paper is well written and the data is clearly presented. The data largely supports the conclusion that membrane localized LIN-5 has a direct role in furrow formation. However, the data remains incomplete in supporting the notion that LIN-5 limits the spreading of ring components away from the equator. For acceptance in JCB in particular this point needs further support. Additionally, no molecular insight in how LIN-5 confines NMY-2 and CYK-4 to the furrow site is presented.

Main Points:

1. The authors propose that polar enriched LIN-5 'confines' NMY-2 to the equator and CYK-4 to the midbody. If cortical NMY-2 or membrane-bound CYK-4 are no longer confined, I would expect a broadening of the equatorial NMY-2/CYK-4 zone after *lin-5(RNAi)*. However, based on the current data for NMY-2 (Fig. 3, 4) a spreading away from the equator is not observed and has not been checked for CYK-4. Therefore, it should be analyzed whether membrane localized CYK-4 spreads away from the equator in *lin-5(RNAi)* at furrow formation.
2. A previous report demonstrated that the LIN-5/GRP-1/2 complex removes NMY-2 from the polar cortex via dynein (Chapa-y-Lazo 2020, JCB). Inconsistent with that model the authors only observe an increase of NMY-2 at the posterior but not the anterior pole after *lin-5* depletion (Fig.3G). To more thoroughly analyze this: measure polar NMY-2 intensity on cortical (as e.g. in Lewellyn 2010 MBoC Fig.3) and central plane images over time. Could it be that there is a delay in the removal of NMY-2 from the poles but at the time of furrow indentation (Fig. 3G) NMY-2 finally is removed?
3. The authors conclude that the NMY-2 decrease at the anterior and increase at the posterior is due to altered aster positioning in *lin-5(RNAi)*. To support this claim, the authors should check whether increased aster separation in *lin-5(RNAi)* again lowers NMY-2 levels in the posterior after *lin-5(RNAi)*.
4. In Fig. 5A-G the authors show that in *lin-5(RNAi)* *lin-5ts* embryos equatorial NMY-2 levels are lower than in controls and conclude the LIN-5 confines NMY-2 to the equator. Why is the reduction in equatorial NMY-2 levels not seen in a similar type of analysis in Fig. 3E-G? Is it because the experiment was not performed in the *lin-5ts* background? If the *lin-5(RNAi)* *lin-5ts* gives a stronger phenotype this genotype should also be used to analyze polar NMY-2 levels. A reduction in cortical NMY-2 should be visible in both type of measurements (Fig. 3G/4G). Additionally, to place a box on the membrane (Fig. 4F) is not the ideal tool for measuring intensity on the equator, since depending on the position of the box different sized cytoplasmic/outside regions are included which will strongly change the sum intensity. For such type of measurements a linescan is more suitable. The authors need to clarify all those issues before drawing conclusions on the NMY-2 levels.
5. The *zyg-9* background is used to analyze the *lin-5* phenotype independently of aster positioning (Fig. 3H-P). How does LIN-5 localize in *zyg-9(RNAi)* embryos? Is LIN-5 enriched over the posterior asters and excluded from the membrane closest to the central spindle? Knowing where LIN-5 localizes in *zyg-9(RNAi)* is important for the conclusion of this experiment.
6. To exclude that the loss of CYK-4 localization on the spindle midzone region in *lin-5(RNAi)* *spd-1ts* (Fig. 5M) is not caused by the absence of the overlapping astral MTs (Hirsch JCB 2022), it must be demonstrated that astral MTs still overlap at the same extend in *spd-1ts* *lin-5(RNAi)* embryos as in *spd-1ts* alone.
7. The number of embryos quantified is often too low for statistical analysis and meaningful conclusions (e.g. Fig. 3 n=5 (Fig. 3R) or n=6 (Fig. 3S)). Please increase for all quantifications the number of embryos to at least 8 (better 10) embryos.

Minor points:

1. A previous study reported a premature in furrow formation in *spd-1ts* (Chapa-y-Lazo JCB, 2020, Fig. 2C). Why do the author not observe a delay?
2. The authors state: "Notably, the simultaneous depletion of LIN-5 along with ZEN-4 leads to the complete loss of furrow formation, as reported earlier (Figure S1F; Dechant and Glotzer, 2003; Verbrugghe and White, 2004; Chapa-y-Lazo et al., 2020)" In none of the references I could find the *lin-5,zen-4* double mutant condition, so change the statement or references.

4. In Fig. 3G the authors conclude that in *lin-5(RNAi)* NMY-2 levels at the anterior are reduced and increased at the posterior. Can the authors provide statistics to support this conclusion.
5. In Fig. 3A/B cortical LifeAct images are shown. Are the same changes on cortical LifeAct localization seen as for NMY-2? Also, LifeAct intensity is very bright and therefore it is really hard to see any differences. Please reduce intensity scaling for LifeAct.
6. I find some labellings confusing and difficult to follow: for example in Fig. 4G it is labeled *lin-5 (RNAi) lin-5 (ts)* but in Fig. 3S it is only labeled *lin-5(RNAi)*. This makes it hard for the reader to follow which exact genotype was used. Please always put the full genotype in the figure panel.
7. In Fig.3 L-P the posterior line is digitally straightened? How?
8. In Fig. 4 some samples do not represent the quantification of the graph. In *par-2(RNAi)* the levels on graph at 100s are higher than in controls, but this is not seen in the images. Similarly, in the images cortical NMY-2 levels in *lin-5(RNAi)* seem already less before onset of furrowing (100s) but this is not seen in the graph.
8. Include equatorial NMY-2 quantification in *par-5(RNAi)* and *par-5 (RNAi) lin-5 (RNAi)* in Fig. 4G.
9. In Fig. 4N the *lin-5(RNAi)lin-5ts* data looks to me the same as in Fig 1J: same distribution of dots and mean, only slightly different SEM. Is it really a different data set as indicated by the label (*lin-5(RNAi)lin-5ts* and *lin-5(RNAi)*)?
10. Do not only plot the ratio of A/P pole-cortex distance (Fig. S3E/F) but also the A-cortex and P-cortex distance since this distance is important.
11. In this sentence a word seems missing. "Our work indicates that cortical LIN-5/GPR-1/2 post furrow ingression redundantly works microtubule bundler protein SPD-1 in ensuring the stability of the midbody and thus abscission zone. "
12. Throughout the paper student's t-test was used. For this test the data must follow a normal distribution. Please state how you checked normality of your data.
13. The Movie legend of Movie 1 and 2 are identical. Specify the genotype of each movie.
13. Define the abbreviation: w.r.t.
14. Specify in the strain list the exact genotype of each strain and not only which strains were crossed with each other. (e.g. CYK-4-mNG;WH12: Cross: OD3619 x WH12) Cite the original publication the strain was first used.
15. Do GRP-1/2 depleted embryos also exhibit also a delay in furrow formation?

Reviewer #2 (Comments to the Authors (Required)):

This study explores possible new roles for LIN-5 in the regulation of cleavage furrow initiation and stabilization of the spindle midzone during cell division in the *C. elegans* zygote. LIN-5 and its partners GPR-1/2 have well-documented roles in controlling forces on astral microtubules that regulate spindle position during cell division. Here, the authors study two additional defects in cytokinesis produced by depleting LIN-5 or GPR-1/2: A delay of ~ 70 seconds in the initiation of visible furrow ingression and a failure to maintain the normal levels of contractile ring components at the midbody after furrow ingression, which can lead to furrow regression in certain mutants (*spd-1*).

Through a series of experiments designed mainly to rule out alternative models, the authors conclude that LIN-5 and GPR-1/2 act in a way that is independent of their roles in regulating pulling force on microtubules to promote timely furrow ingression and sustained midbody enrichment of contractile ring components. They propose (without testing) several possible models for how polar enrichment of LIN-5 and GPR-1/2 could act at a distance to promote accumulation of contractile ring components at the furrow during initiation and at the midbody after ingression.

The paper contains a number of interesting observations, but unfortunately, for this reviewer at least, they do not make a compelling case for a novel role for LIN-5 in controlling equatorial/midbody accumulation of contractile ring components. As described above, many of the experiments reported here are designed to rule out alternative hypotheses based on regulation of cortical pulling forces in microtubules by LIN-5. Some of these experiments are reasonably compelling (e.g. the rescue of spindle pole separation defects does not rescue delays in furrow initiation), but many are subject to multiple interpretations that make it difficult to draw strong conclusions. In general, it is difficult to make a strong case for a novel mechanism by ruling out possible alternative explanations because the list of possible alternatives is quite long, and in its current form at least, the paper lacks any direct evidence for the proposed alternative models.

Major Comments

(1) The authors nicely show that "rescuing" the aster separation defects observed in *lin-5(RNAi)* by co-depleting SPD-1 or HCP-4 do not prevent the delays in furrow initiation. However, another defect observed in *lin-5(RNAi)* embryos is delayed rotation of the mitotic spindle, such that the spindle is skewed at anaphase onset, causing the actomyosin contractile ring to form in a skewed orientation. This skew is clearly present in Movie S2 showing accumulation of myosin at the cortex in *lin-5(RNAi)* embryos. The movie shows that the contractile ring initiates in a skewed orientation and before rotating into alignment with the long axis. This could cause significant delays in the initiation of a visible furrow. Can the authors rule out this simple physical explanation?

(2) The zyg-9; lin-5 experiment is hard to interpret. The stated goal is to test more firmly a previously proposed model that posteriorly enriched LIN-5 promotes the removal of myosin II from the posterior pole to facilitate rapid furrow initiation, but it is unclear how this experiment tests that model directly in a controlled way. It is true that the large-scale distribution of myosin (high at anterior and low near posterior) is similar in zyg-9(RNAi) and zyg-9(RNAi);lin-5(RNAi) embryos, but myosin does accumulate, albeit at much lower levels, on the posterior cortex where the posterior furrow forms. Isn't it the spatial pattern of this local accumulation, and how it is affected by LIN-5 depletion, that matters? Where is LIN-5 localized in zyg-9(RNAi)? How do they rule out a more local role for LIN-5/Dynein in removing NMY-2 from regions proximal to the posterior spindle? The spindle poles are much closer to the cortex in LIN-5/ZYG-9 double depletions which could affect the accumulation of cortical myosin through other pathways.

(3) The measurements of equatorial myosin accumulation in Figure 4 are hard to interpret for reasons related to Major Comment 1 above. If the contractile ring forms in an initially skewed position, then measurements of myosin at the equatorial cortex would not accurately report on the accumulation of myosin within the nascent contractile ring. Since it is presumably the very earliest pattern of myosin II accumulation that is relevant to the initial delay in furrow ingression, it would be more informative to perform a more time-resolved analysis of the temporal dynamics and spatial pattern of myosin (and F-actin) accumulation and organization across the cortical surface during the period immediately after anaphase onset, rather than relying on observations of intensity in individual equatorial sections.

(4) The PAR-5 rescue experiment is also difficult to interpret. The observation that PAR-5 depletion compensates for the effects of LIN-5 does not imply anything about how LIN-5 is acting. PAR-5 depletion does not just affect the cortical localization of centralspindlin, it also affects the cortical localization and distribution of the PAR proteins.

(5) Jordan and Canman (2016) also considered how PAR proteins PAR-2 and PAR-6 affect contractile ring assembly and furrow ingression and propose a role for the PARs in preventing accumulation of anillin and septin at the cleavage furrow. It would be useful to consider these results in interpreting the PAR depletion experiments shown in Figure 2.

(6) The observation that furrow initiation is delayed in AB cells where LIN-5 does not have a major role in axial spindle position is interesting, but in these cells there are significant delays in spindle pole separation that could affect the patterning of myosin and other contractile ring components. Further analysis of the LIN-5 phenotype in these cells could be very informative.

Minor Comments

(1) NEBD is used as a time reference to compute delays. Why not use anaphase onset - a more proximal and equally accurate marker?

(2) Fig S3 Quantitative measurements of polar myosin levels are made after the furrow has already initiated. Why not before/during initiation?

(3) Time axis labels for kymographs are confusing. What does 100 sec onwards from NEBD mean exactly? Labelled vertical tick marks would be more clear.

The point-by-point response to reviewers

Reviewer #1

We sincerely thank this reviewer for recognizing that “*the data is clearly presented, and the data largely support the conclusion.*” We also greatly appreciate the reviewer's insightful remarks, which have helped us substantially improve the manuscript by addressing several outstanding issues.

Major points:

1. The authors propose that polar enriched LIN-5 ‘confines’ NMY-2 to the equator and CYK-4 to the midbody. If cortical NMY-2 or membrane-bound CYK-4 are no longer confined, I would expect a broadening of the equatorial NMY-2/CYK-4 zone after lin-5(RNAi). However, based on the current data for NMY-2 (Fig. 3, 4) a spreading away from the equator is not observed and has not been checked for CYK-4. Therefore, it should be analyzed whether membrane localized CYK-4 spreads away from the equator in lin-5(RNAi) at furrow formation.

Response: We thank the reviewer for raising this important point. Contractile ring components, such as NMY-2 and Anillin, typically accumulate not only at the equatorial cortex but also at the anterior cortex, while being generally excluded from the posterior cortex due to the proximity of the mitotic spindle to the posterior cortex during furrow onset (Lewellyn et al., 2010, PMID: 19889842; Mangal et al., 2018, PMID: 29311228). In contrast, in LIN-5 depleted embryos, however, we observe the opposite pattern—contractile ring component GFP-NMY-2, is notably excluded from the anterior cortex (Fig. 3). Since NMY-2 localization is not tightly confined to the furrow-forming site, visualizing its spreading upon LIN-5 depletion has proven challenging. Nevertheless, we have shown that the overall intensity of GFP-NMY-2 at the equatorial cortex during furrow formation is significantly reduced in LIN-5-depleted embryos (Fig. 4).

As suggested by the reviewer (major point 2), we have now strengthened these findings by:

1. Analyzing the dynamics of GFP-NMY-2 in both control and LIN-5 inhibited embryos at the central and cortical planes using linescan analysis.
2. Characterizing the localization of another contractile ring component, Anillin, using endogenously tagged mNeonGreen-Anillin (mNG-ANI-1; Lebedev et al., 2023, PMID: 37665665) in both control and LIN-5 depleted embryos. We focused on Anillin because its localization to the equatorial membrane requires RhoA activation, but not myosin II (Straight et al., 2005, PMID: 15496454; Hickson and O'Farrell, 2008, PMID: 18209105).

As shown in the revised manuscript, we found that LIN-5 is essential for the proper accumulation of both NMY-2 and Anillin at the equatorial cortical region at the time of furrow onset. Specifically, in LIN-5-depleted embryos, the equatorial cortical accumulation of NMY-2 and Anillin is significantly reduced. These new data are included in Fig. 3A-3D, Fig. S3A-S3D, and Fig. S5 and are discussed in the revised manuscript on pages 14 and 15.

As the reviewer is aware, the centralspindlin complex (CYK-4/ZEN-4) localizes to the spindle midzone and becomes part of the midbody as the furrow fully ingresses (Hirsch et al., 2021, PMID: 34994802). Notably, unlike LIN-5, centralspindlin is not essential for timely furrow formation (Jantsch-Plunger et al., 2000, PMID: 10871280; Dechant and Glotzer, 2003, PMID: 12636915; Canman et al., 2008, PMID: 19056985; Lewellyn et al., 2010, PMID: 19889842; Lewellyn et al., 2011, PMID: 21464231; Schlientz et al., 2024, bioRxiv-<https://doi.org/10.1101/2024.10.29.620943>). Since the centralspindlin component CYK-4

hardly localizes to the equatorial cortex in control embryos during furrow onset (Basant et al., 2015, PMID: 25898168; Schlientz et al., 2024, bioRxiv-<https://doi.org/10.1101/2024.10.29.620943>), we did not attempt to visualize CYK-4 at the equatorial cortical zone. However, as CYK-4, along with ZEN-4, is crucial for contractile ring ingression and is typically confined to the midbody and its membrane (Fig. 5; Green et al., 2013, PMID: 24217623; Hirsch et al., 2021, PMID: 34994802), we tracked CYK-4-mNG localization as a proxy for contractile ring components after furrow ingression in *spd-1(ts)* embryos upon LIN-5 depletion (Fig. 5H). Our analysis revealed that CYK-4-mNG fails to remain confined to the midbody and its surrounding membrane in these embryos. Instead, it looks like it spreads across the membrane, leading to cytokinesis failure. Prompted by the reviewer's comment, we have now also visualized Anillin localization at the midbody by examining mNG-ANI-1. Similar to GFP-CYK-4, we found that mNG-ANI-1 fails to remain confined to the midbody in *spd-1(ts)* embryos upon LIN-5 depletion, and it appears that it spreads away. This new data is shown below with a few examples and has been added as one representative image with the quantification in Fig. S7G-S7J. These findings regarding the localization of CYK-4 and Anillin collectively strengthen our previous conclusion that cortical LIN-5 regulates the proper confinement of midbody components post-furrow ingression.

Cortical accumulation of LIN-5 post furrow ingression is essential for the confinement of Anillin at the midbody

(A-F) Multiple examples of pseudokymographs of the division plane (depicted on the top left) acquired by time-lapse confocal microscopy of *spd-1(ts)* embryos expressing mNG-ANI-1 (A-C), or *spd-1(ts)* embryos expressing mNG-ANI-1 that are treated with *lin-5(RNAi)* (D-F). The pseudokymograph begins at -45 s w.r.t furrow involution, which is 0 s. The time interval between each frame is 15 s. Please note that the mNG-ANI-1 is significantly enriched at the midbody post-furrow ingression in *spd-1(ts)* mutant embryos. In contrast, mNG-ANI-1 intensity is diffused in *spd-1(ts)* embryos that are depleted for LIN-5. The scale bar is 5 μ m.

2. A previous report demonstrated that the LIN-5/GRP-1/2 complex removes NMY-2 from the polar cortex via dynein (Chapa-y-Lazo 2020, JCB). Inconsistent with that model the authors only observe an increase of NMY-2 at the posterior but not the anterior pole after *lin-5* depletion (Fig.3G). To more thoroughly analyze this: measure polar NMY-2 intensity on cortical (as e.g. in Lewellyn 2010 MBoC Fig.3) and central plane images over time. Could it

be that there is a delay in the removal of NMY-2 from the poles but at the time of furrow indentation (Fig. 3G) NMY-2 finally is removed?

Response: As suggested, we have now analyzed the localization dynamics of NMY-2 at both the cortical and central planes over time. Consistent with our earlier findings, we observed that in LIN-5 depleted embryos, GFP-NMY-2 is significantly more cleared from the anterior cortex during anaphase compared to controls (Fig. 3D). These results, along with our ZYG-9 and LIN-5 double depletion experiments in Fig. 3G, indicate that contrary to the model proposed by Chapa-y-Lazo et al., 2020 (PMID: 32497213), where NMY-2 is actively stripped by LIN-5/dynein, the clearance of NMY-2 seems to be influenced by the proximity of the mitotic spindle to the cell cortex. This new data has been incorporated into Fig. 3 and Fig. S3 and is discussed on pages 11 and 12 of the revised manuscript.

We believe that the complete removal of NMY-2 from both polar cortices is not essential for furrow formation. However, as the reviewer pointed out, there is a gradual reduction of NMY-2 from the polar cortical regions, a process similarly observed for another contractile ring protein, Anillin (Mangal et al., 2018, PMID: 29311228).

3. The authors conclude that the NMY-2 decrease at the anterior and increase at the posterior is due to altered aster positioning in *lin-5(RNAi)*. To support this claim, the authors should check whether increased aster separation in *lin-5(RNAi)* again lowers NMY-2 levels in the posterior after *lin-5(RNAi)*.

Response: We thank the reviewer for their insightful suggestion. We have now monitored GFP-NMY-2 localization at the posterior cortex from 100 s post-NEBD to furrow formation in *lin-5(RNAi); hcp-4(RNAi)* embryos, which exhibit increased aster separation. As shown below, the enhanced aster separation in these embryos decreases the distance between the posterior pole and the posterior cell cortex. However, this decrease is not as pronounced as in control embryos, as the spindle in *lin-5(RNAi)* embryos tends to position itself at the embryo center or towards the anterior.

Nevertheless, as demonstrated in two distinct examples, we observed that NMY-2 cortical levels at the posterior cortex are significantly lower in *lin-5(RNAi); hcp-4(RNAi)* embryos compared to *lin-5(RNAi)* embryos. This finding supports the idea that a spindle-guided signal, rather than LIN-5-mediated NMY-2 removal, controls NMY-2 clearance. This new data has been included with one representative image in Fig. S3M-S3Q and is discussed on page 12 of the revised manuscript. We decided not to include the posterior pole-to-posterior cortex distance in these conditions, as we believe it will be apparent to readers.

Position of the spindle dictates NMY-2 clearance

(A-C) Images of the posterior cortical region (shown on the left) used for the kymograph analysis from the time-lapse confocal microscopy of a one-cell embryo expressing endogenous tagged NMY-2 with GFP (GFP-NMY-2; in green) in *lin-5(ts)* mutant embryos that are depleted for LIN-5 by RNAi (A), or *lin-5(ts)* mutant embryos that are codepleted for LIN-5 and HCP-4, shown by two independent examples (B, C). Time is w.r.t. NEBD. The scale bar is 5 μ m.

(D, E) Schematic of the method used to measure the distance between posterior spindle pole and posterior cortex (D), and its outcome (E) in embryos depleted for LIN-5 in *lin-5(ts)* background (pink line), as well as in embryos codepleted for LIN-5 and HCP-4 in *lin-5(ts)* background (blue line). Error bars, SEM. ns, $p > 0.05$; *, $p < 0.05$, as determined by two-tailed Student's t-test.

(F, G) Schematic of the method used to quantify GFP-NMY-2 fluorescence intensity at the posterior cortex over time (F). Quantification of GFP-NMY-2 enrichment at the posterior cortex in embryos depleted for LIN-5 in *lin-5(ts)* background (pink line), as well as in embryos codepleted for LIN-5 and HCP-4 in *lin-5(ts)* background (blue line). Time is w.r.t. NEBD. Error bars, SEM. ns, $p > 0.05$; *, $p < 0.05$, as determined by two-tailed Student's t-test.

4A. In Fig. 5A-G the authors show that in lin-5(RNAi) lin-5ts embryos equatorial NMY-2 levels are lower than in controls and conclude the LIN-5 confines NMY-2 to the equator. Why is the reduction in equatorial NMY-2 levels not seen in a similar type of analysis in Fig. 3E-G? Is it because the experiment was not performed in the lin-5ts background? If the lin-5(RNAi) lin-5ts gives a stronger phenotype this genotype should also be used to analyze polar NMY-2 levels. A reduction in cortical NMY-2 should be visible in both type of measurements (Fig. 3G/4G).

Response: We thank the reviewer for raising this important point and for highlighting the discrepancy. The analysis in Fig. 4A-4G was carried out using an endogenously tagged GFP-NMY-2 strain generated by the Goldstein lab [nmy-2(cp13[nmy-2::GFP + LoxP]) I]. In contrast, the analysis in Fig. 3E-3G were performed using embryos ectopically expressing GFP-NMY-2. As the reviewer correctly noted, *lin-5(RNAi)*; *lin-5(ts)* provide a more robust and consistent phenotype with less variability compared to *lin-5(RNAi)* alone. In response, we have now quantified NMY-2 intensity (both cortical and central plane) in the CRISPR line as well as in the *lin-5(RNAi)*; *lin-5(ts)* background. Additionally, in instances where *lin-5(RNAi)* was used, we relied heavily on phenotypic indicators such as the strong mispositioning of the mitotic spindle and defects in polar body exclusion to ensure that LIN-5 is effectively depleted in these embryos. This has been clearly stated in the revised materials and methods section.

4B. Additionally, to place a box on the membrane (Fig. 4F) is not the ideal tool for measuring intensity on the equator, since depending on the position of the box different sized cytoplasmic/outside regions are included which will strongly change the sum intensity. For such type of measurements, a linescan is more suitable. The authors need to clarify all those issues before drawing conclusions on the NMY-2 levels.

Response: We appreciate the reviewer's comment, but we respectfully disagree with the assertion that "placing a box on the membrane is not ideal for measuring cortical intensity." For all conditions where NMY-2 intensity was measured at the equatorial cortical region, we consistently used a rectangular box (4F) that minimally included cytoplasmic areas, and it was carefully positioned at the future furrow formation site. While we acknowledge that using the box will vary the total membrane intensity, this effect should be minimal and uniform across all embryos, given that the box covers only a small cytoplasmic area. Further, we conducted this analysis across multiple time frames in a minimum of 8 embryos and thus, even accounting for human error in placing the boxes on the furrow region, we get a significant difference in intensity, which matches the visual decrease/increase in intensity in various conditions.

Additionally, as per the reviewer's suggestion, we have now included linescan analysis for GFP-NMY-2 and mNG-ANI-1 over time at both the cortical and central planes following

LIN-5 depletion. In line with the previous results in Fig. 4, these data show that the contractile ring components GFP-NMY-2 and mNG-ANI-1 are weakly localized to the equatorial cortical region when LIN-5 is depleted. This new data has been incorporated into Fig. 3, Fig. S3, and Fig. S5, and is discussed on pages 14 and 15 of the revised manuscript.

5. The zyg-9 background is used to analyze the lin-5 phenotype independently of aster positioning (Fig. 3H-P). How does LIN-5 localize in zyg-9(RNAi) embryos? Is LIN-5 enriched over the posterior asters and excluded from the membrane closest to the central spindle? Knowing where LIN-5 localizes in zyg-9(RNAi) is important for the conclusion of this experiment.

Response: We thank the reviewer for raising this important point, which was also emphasized by the second reviewer (major concern 2). Following the suggestion, we have examined LIN-5 localization in *zyg-9(RNAi)* embryos. We observed that in *zyg-9(RNAi)* embryos, the spindle undergoes oscillations, likely because it is not stably anchored to the cortex. This posed some difficulty in visualizing cortical LIN-5 during anaphase in the *zyg-9(RNAi)* background. To address this limitation, we captured multiple snapshots of embryos expressing endogenously tagged mNG-LIN-5 at various cell cycle stages in *zyg-9(RNAi)* condition.

As shown in Fig. S4G, this analysis revealed that LIN-5 is present at the posterior cortex, although it is difficult to conclude if it is excluded from the membrane closest to the spindle midzone. Additionally, we found that LIN-5 beautifully and uniformly decorates the double membrane of the invaginating furrow, displaying a distinct localization pattern compared to NMY-2, which only enriches at the furrow tip in *zyg-9(RNAi)* embryos. To further indirectly assess LIN-5 presence at the posterior cortex, we examined the effect of LIN-5 depletion on posterior aster separation in *zyg-9(RNAi)* embryos. Indeed, LIN-5 depletion resulted in reduced aster separation, suggesting that LIN-5 is likely localized at the cortex opposite the poles and plays a role in promoting aster separation. Based on these findings, we propose that LIN-5 localization at the posterior cortex close to the invaginating membrane is crucial for stabilizing NMY-2 at the furrow formation site. In the presence of LIN-5, NMY-2 accumulates progressively at the furrow tip, promoting efficient furrow formation in ZYG-9-depleted embryos.

6. To exclude that the loss of CYK-4 localization on the spindle midzone region in lin-5(RNAi) spd-1ts (Fig. 5M) is not caused by the absence of the overlapping astral MTs (Hirsch JCB 2022), it must be demonstrated that astral MTs still overlap at the same extend in spd-1ts lin-5(RNAi) embryos as in spd-1ts alone.

Response: We sincerely thank the reviewer for suggesting this possibility. LIN-5 depletion partially suppresses the rupture of the spindle midzone observed in *spd-1(ts)* embryos (compare Fig. 5H with 5F), likely due to the absence of cortical pulling forces (Lee et al., 2015; PMID: 26088160). For this reason, we initially did not consider that the loss of CYK-4 might be linked to the absence of overlapping astral microtubules in *lin-5(RNAi); spd-1(ts)* embryos, as the spindle midzone is not entirely diminished and CYK-4 remains localized at the midzone (compare 0 s in *lin-5(RNAi); spd-1(ts)* with *spd-1(ts)* condition in Fig. 5). However, as per the reviewer's suggestion, we have now analyzed the presence of overlapping astral microtubules during late cytokinesis in *lin-5(RNAi); spd-1(ts)* embryos. As shown in Fig. S7G-S7J, our data clearly demonstrate that overlapping astral microtubules are indeed present in *lin-5(RNAi); spd-1(ts)* embryos. This observation aligns with previous findings regarding astral microtubule localization in *spd-1(ts)* embryos (Hirsch et al., 2022; PMID: 34994802).

7. The number of embryos quantified is often too low for statistical analysis and meaningful conclusions (e.g. Fig. 3 n=5 (Fig. 3R) or n=6 (Fig. 3S)). Please increase for all quantifications the number of embryos to at least 8 (better 10) embryos.

Response: We have increased the number of embryos used for statistical analysis throughout the manuscript; thanks.

Minor points:

*1. A previous study reported a premature in furrow formation in *spd-1ts* (Chapa-y-Lazo JCB, 2020, Fig. 2C). Why do the author not observe a delay?*

-Chapa-y-Lazo et al. defined the “timing of the initial sign of furrow formation” as the onset of furrow appearance in the live recording relative to post-anaphase. In contrast, we have referred to the timing of “furrow involution”—the appearance of two adhered back-to-back plasma membranes in a side view, as described by Lewellyn et al., 2010—as the benchmark for furrow formation. We prefer using furrow involution-based timing measurements as we believe it provides greater precision in assessing the exact timing of furrow formation.

Another possible explanation for the differences in timing observed between our study and that of Chapa-y-Lazo et al. could be the use of different *C. elegans* strains. While Chapa-y-Lazo et al. measured furrow initiation timing in embryos expressing GFP-NMY-2, our analysis was conducted in embryos expressing GFP^{Mem./Tub.} and mCherry^{Chr./Cent.} Importantly, our results align with the findings from the *spd-1(RNAi)* experiments performed by Lewellyn et al., 2010 (Fig. 5), where the authors concluded that “*the equatorial recruitment of contractile ring proteins and furrow formation occurred with normal timing in SPD-1 depleted embryos.*”

*2. The authors state: "Notably, the simultaneous depletion of LIN-5 along with ZEN-4 leads to the complete loss of furrow formation, as reported earlier (Figure S1F; Dechant and Glotzer, 2003; Verbrugghe and White, 2004; Chapa-y-Lazo et al., 2020)" In none of the references I could find the *lin-5,zen-4* double mutant condition, so change the statement or references.*

-Thanks, we have now changed the statement.

*3. In Fig. 3G the authors conclude that in *lin-5(RNAi)* NMY-2 levels at the anterior are reduced and increased at the posterior. Can the authors provide statistics to support this conclusion.*

-Thanks; we have now provided the *statistics in Fig. 3 to support this conclusion.*

4. In Fig. 3A/B cortical LifeAct images are shown. Are the same changes on cortical LifeAct localization seen as for NMY-2? Also, LifeAct intensity is very bright and therefore it is really hard to see any differences. Please reduce intensity scaling for LifeAct.

-Thank you for your comment. The LifeAct intensity appeared bright because we were not showing LifeAct alone but rather merged with NMY-2. For clarity, we have removed LifeAct from Fig. 3 and as discussed above, we have added data on another contractile ring protein, Anillin.

*5. I find some labellings confusing and difficult to follow: for example in Fig. 4G it is labeled *lin-5 (RNAi) lin-5 (ts)* but in Fig. 3S its only labeled *lin-5(RNAi)*. This makes it hard for the reader to follow which exact genotype was used. Please always put the full genotype in the figure panel.*

-Thanks—we have fixed it throughout the manuscript.

6. In Fig.3 L-P the posterior line is digitally straightened? How?

-Digital straightening of a curved line (or the signal at the cell cortex) of an image in ImageJ can be done using the “Straighten” plugin, available through the “Edit” menu. We have used

this plugin to make the images shown in Fig. 3K-3O. We have added this information in the materials and methods section.

7. In Fig. 4 some samples do not represent the quantification of the graph. In par-2(RNAi) the levels on graph at 100s are higher than in controls, but this is not seen in the images. Similarly, in the images cortical NMY-2 levels in lin-5(RNAi) seem already less before onset of furrowing (100s) but this no seen in the graph.

-We thank the reviewer for raising these important points. The increased cortical enrichment of GFP-NMY-2 at 100 s in *par-2(RNAi)* embryos is due to the overall lower cytoplasmic intensity of GFP-NMY-2 at 100 s post-NEBD in these embryos. For control embryos, the mean cortical intensity after background subtraction is 1391.2, while the mean cytoplasmic intensity is 893.6, resulting in a ratio of less than 2. However, in PAR-2-depleted embryos, although the cortical intensity is nearly similar (1268.8), the cytoplasmic intensity is much lower (412.9), leading to a higher ratio compared to controls.

Similarly, in *lin-5(RNAi)* embryos with the *lin-5(ts)* mutation at 100 s, the mean cortical intensity is 1481.4, and the mean cytoplasmic intensity is 871.4, giving a ratio of less than 2. Please note that the representative LIN-5-inhibited embryo images were used to illustrate the reduced GFP-NMY-2 localization during anaphase, ~200 s. At this point, control embryos usually initiate furrow formation.

8. Include equatorial NMY-2 quantification in par-5(RNAi) and par-5 (RNAi) lin-5 (RNAi) in Fig. 4G.

-Added, thanks

9. In Fig. 4N the lin-5(RNAi)lin-5ts data looks to me the same as in Fig 1J: same distribution of dots and mean, only slightly different SEM. Is it really a different data set as indicated by the label (lin-5(RNAi)lin-5ts and lin-5(RNAi)?

-We apologize for overlooking this data, and we are thankful to the reviewer for finding this error. By mistake, during compiling Figures, we included *lin-5(RNAi);lin-5(ts)* data instead of *lin-5(RNAi)* data points. This has been rectified in the revised version.

10. Do not only plot the ratio of A/P pole-cortex distance (Fig. S3E/F) but also the A-cortex and P-cortex distance since this distance is important.

-Thanks for this point, as requested we have now added the anterior-cortex, and posterior-cortex distance data as well in the Fig. S3K-S3N.

11. In this sentence a word seems missing. "Our work indicates that cortical LIN-5/GPR-1/2 post furrow ingression redundantly works microtubule bundler protein SPD-1 in ensuring the stability of the midbody and thus abscission zone. "

-Fixed, thanks

12. Throughout the paper student's t-test was used. For this test the data must follow a normal distribution. Please state how you checked normality of your data.

-We thank the reviewer for raising this point. We have now thoroughly assessed our data's normality using the Kolmogorov-Smirnov tests. We have also looked into the Q-Q plot, which indicates the normality of a data set.

For data that did not follow a normal distribution, we initially applied the Mann-Whitney test, a non-parametric alternative for unpaired samples. However, since the p-value remained

unchanged, we proceeded with a two-tailed Student's t-test, as the data exhibited only slight skewness, lacked outliers, and was approximately normally distributed.

13. The Movie legend of Movie 1 and 2 are identical. Specify the genotype of each movie.

-We have specified the genotype of each movie in the revised manuscript.

14. Define the abbreviation: w.r.t.

-Thanks, defined, 'with respect to'.

15. Specify in the strain list the exact genotype of each strain and not only which strains were crossed with each other. (e.g. CYK-4-mNG;WH12: Cross: OD3619 x WH12) Cite the original publication the strain was first used.

-Thanks, we have now added the information of the original paper where the strain was utilized for the first time. If the strain was received from CGC, we have mentioned this in the table.

16. Do GPR-1/2 depleted embryos also exhibit also a delay in furrow formation?

-Yes indeed, GPR-1/2 depleted embryos exhibit a delay in furrow formation. We have now included the data related to the impact of GPR-1/2 depletion on furrow involution timing in Fig. S1B and S1E, and these results are discussed on p. 7 of the revised manuscript. Since GPR-1/2 is needed for cortical anchoring of LIN-5, this data additionally shows that cortical accumulation of LIN-5 is relevant for timely furrow involution.

Reviewer #2

The reviewer acknowledged that “*the paper contains a number of interesting observations.*” However, they raised the concern that “*it is difficult to make a strong case for the novel mechanism... because the list of possible alternatives is quite long.*”

We would like to emphasize that LIN-5/GPR-1/2 are critical components of the cortical force generation machinery in metazoans, and they have been shown to play key roles in aster separation and spindle positioning by anchoring dynein at the cell cortex (Kotak, 2019; PMID: 30823600). It remains unclear to us which alternatives the reviewer is referring to, aside from their first major concern. In the original version of the manuscript, we explained the relevance of cortical LIN-5 in furrow formation, independent of its role in aster separation and spindle positioning (see Fig. 1, Fig. 2, and Fig. S2). In the revised manuscript, prompted by the reviewer's first and third concerns, we also explored whether the delayed furrow onset could be linked to the “skewed spindle” at the onset of anaphase following LIN-5 depletion. We appreciate the reviewer for raising this point. Additionally, we have made efforts to address other concerns raised by the reviewer, as detailed below.

Major points:

(1) The authors nicely show that “rescuing” the aster separation defects observed in lin-5(RNAi) by co-depleting SPD-1 or HCP-4 do not prevent the delays in furrow initiation. However, another defect observed in lin-5(RNAi) embryos is delayed rotation of the mitotic spindle, such that the spindle is skewed at anaphase onset, causing the actomyosin contractile ring to form in a skewed orientation. This skew is clearly present in Movie S2 showing accumulation of myosin at the cortex in lin-5(RNAi) embryos. The movie shows that the contractile ring initiates in a skewed orientation and before rotating into alignment with the long axis. This could cause significant delays in the initiation of a visible furrow. Can the authors rule out this simple physical explanation?

Response: We appreciate the reviewer for highlighting this point. When we started working on this project, we considered this possibility. Importantly, not all embryos depleted of LIN-5 or GPR-1/2 begin anaphase in a skewed orientation. However, regardless of spindle orientation, all such embryos exhibit a delay in cleavage furrow formation. Since no correlation was observed between skewed spindle orientation during anaphase and the delay in furrow formation, we initially chose not to include this observation in the original manuscript.

In response to the reviewer's comment, we have now addressed this in the revised manuscript. Briefly, we categorized the percentage of skewed and non-skewed spindles in embryos coexpressing GFP^{Mem.};mCherry^{Tub.} upon LIN-5 inhibition [*lin-5(RNAi)*; *lin-5(ts)*]. As shown on the next page, this analysis revealed that 20% of the embryos showed skewed spindles, while 80% displayed non-skewed spindles. Importantly, the average timing of furrow ingression was not significantly different between embryos with skewed and non-skewed spindles. Similar results were obtained for embryos expressing GFP^{Mem./Tub.};mCherry^{Chr./Cent.}, which were also depleted of LIN-5 by RNAi. These findings, now included in a new Fig. S2F-S2I, demonstrate that the delay in furrow formation upon LIN-5 depletion is not caused by the initial skewed orientation of the spindle. Additionally, we emphasize that while embryos with skewed spindles do require extra time to align the mitotic spindle along the anteroposterior axis, this alignment typically occurs before or shortly after the onset of anaphase. Since furrow ingression happens during mid-to-late anaphase, when the spindle is already aligned, we do not believe this spindle orientation affects the accumulation of contractile ring components (NMY-2/Anillin- tested in this study) at the equatorial membrane after LIN-5 depletion.

Additionally, the novel phenotype we identified, related to the role of cortical LIN-5 and GPR-1/2 in midbody stability after furrow ingression (Fig. 5), is independent of any known spindle-related role of LIN-5.

Fig. 3

Skewed orientation of the mitotic spindle in LIN-5 depleted/inhibited embryos is not the cause for the delay in furrow involution timing

(A, B) Representative images from the time-lapse confocal microscopy of the one-cell embryo coexpressing GFP^{Mem.};mCherry^{Tub.} in *lin-5(RNAi)*; *lin-5(ts)* background. The spindle was characterized into two categories based on orientation at anaphase: non-skewed spindle (A) and skewed spindle (B). 75 s on the left panel represents the time w.r.t. NEBD. The scale bar is 10 μm.

(C, D) Representative images from the time-lapse confocal microscopy of the one-cell embryo coexpressing GFP^{Mem./Tub.};mCherry^{Chr./Cent.} in *lin-5(RNAi)*. The spindle was characterized into two categories based on orientation at anaphase: non-skewed spindle (C) and skewed spindle (D). 75 s on the left panel represents the time w.r.t. NEBD. The scale bar is 10 μm.

(E-G) Schematic represents the furrow involution in P0 cell (E) and the quantification of the time interval between NEBD and furrow involution in embryos coexpressing either GFP^{Mem.};mCherry^{Tub.} and are inhibited for LIN-5 [*lin-5(RNAi)*; *lin-5(ts)*] (F), or GFP^{Mem./Tub.};mCherry^{Chr./Cent.} in LIN-5 depleted background (G). The spindle is divided into two categories with non-skewed spindle (green dots) and skewed spindle (pink dots), as indicated. Each dot represents one embryo, and the solid black line on the graph represents the mean, whose values and SEM values are also mentioned at the bottom. ns, $p > 0.05$ as determined by a two-tailed Student's t-test.

(2) The *zyg-9; lin-5* experiment is hard to interpret. The stated goal is to test more firmly a previously proposed model that posteriorly enriched LIN-5 promotes the removal of myosin II from the posterior pole to facilitate rapid furrow initiation, but it is unclear how this experiment tests that model directly in a controlled way. It is true that the large-scale distribution of myosin

(high at anterior and low near posterior) is similar in zyg-9(RNAi) and zyg-9(RNAi);lin-5(RNAi) embryos, but myosin does accumulate, albeit at much lower levels, on the posterior cortex where the posterior furrow forms. Isn't it the spatial pattern of this local accumulation, and how it is affected by LIN-5 depletion, that matters? Where is LIN-5 localized in zyg-9(RNAi)? How do they rule out a more local role for LIN-5/Dynein in removing NMY-2 from regions proximal to the posterior spindle? The spindle poles are much closer to the cortex in LIN-5/ZYG-9 double depletions which could affect the accumulation of cortical myosin through other pathways.

Response: We apologize for not providing sufficient clarity in the initial manuscript, which may have led to misunderstandings. Chapa-y-Lazo et al. proposed that the active removal of NMY-2 from the polar region by the GPR-1/2/LIN-5/dynein pathway helps create anisotropy in NMY-2 localization at the equatorial cortex, thus facilitating timely furrow initiation. In Fig. 3, we demonstrated that upon LIN-5 inhibition, the mitotic spindle is positioned closer to the anterior cortex than the posterior cortex. Under these conditions, cortical NMY-2 is significantly reduced from the anterior region compared to the posterior.

As suggested by Reviewer 1, we have strengthened these findings through imaging and quantification of cortical and central planes in embryos expressing GFP-NMY-2 (please see the updated Fig. 3 and corresponding Fig. S3). These observations contradict the model that the GPR-1/2/LIN-5 pathway, via dynein, removes NMY-2 from the cortex. If this were the case, we would expect higher levels of NMY-2 at the anterior cortex, given that the GPR-1/2/LIN-5/dynein pathway is active at both the anterior and posterior cortex.

To further support this observation, we utilized *zyg-9(RNAi)*. Upon ZYG-9 depletion, the bulk cortical pool of NMY-2 resides at the anterior cortex. Thus, we believe this experimental setup provides an effective way to assess the relevance of LIN-5 in cleavage furrow formation independently of its role in NMY-2 stripping. NMY-2 localization remains the same in ZYG-9-depleted or LIN-5 and ZYG-9 codepleted conditions (see Fig. 3G-3J and Fig. S4A and S4B). Moreover, the accumulation of NMY-2 near the spindle midzone in *zyg-9(RNAi)* background is solely driven by midzone-directed pathways regulated by the centralspindlin. In *cyk-4(ts)* embryos depleted for ZYG-9 lead to complete loss of posterior furrow formation (Werner et al., 2007; PMID: 17669650; Loria et al., 2012; PMID: 22226748).

The instability of NMY-2 accumulation at the furrow-forming site in ZYG-9 and LIN-5 inhibited embryos and its subsequent effect on furrow ingression led us to hypothesize a specific role for cortical LIN-5 in stabilizing accumulated NMY-2 over time, ensuring furrow stability. Our model suggests that LIN-5 plays a role in preventing the diffusion of midzone-directed cortical contractile proteins including NMY-2 from regions near the posterior spindle following ZYG-9 depletion.

We agree with the reviewer that examining the localization pattern of LIN-5 at the posterior cortex is critical, and we appreciate this suggestion. A similar point was raised by Reviewer 1 (see our response to their major point 5). In *zyg-9(RNAi)* embryos, LIN-5 is localized to the posterior cortex. We also observed LIN-5 at the invaginating membrane, further supporting its role in stabilizing NMY-2 during furrow ingression. Additionally, we demonstrate that this pool of LIN-5 is functional, as LIN-5 loss results in a shorter pole-to-pole distance, indicating that cortical LIN-5-based forces are essential for spindle elongation in the *zyg-9(RNAi)* background. Moreover, we emphasize that a shorter spindle cannot be responsible for furrow destabilization in *zyg-9(RNAi);lin-5(RNAi);lin-5(ts)* embryos, as we have established in Fig. 1 that LIN-5 role in aster separation does not affect the timing of cleavage furrow formation.

(3) The measurements of equatorial myosin accumulation in Figure 4 are hard to interpret for reasons related to Major Comment 1 above. If the contractile ring forms in an initially skewed position, then measurements of myosin at the equatorial cortex would not accurately report on the accumulation of myosin within the nascent contractile ring. Since it is presumably the very earliest pattern of myosin II accumulation that is relevant to the initial delay in furrow ingression, it would be more informative to perform a more time-resolved analysis of the

temporal dynamics and spatial pattern of myosin (and F-actin) accumulation and organization across the cortical surface during the period immediately after anaphase onset, rather than relying on observations of intensity in individual equatorial sections.

Response: Please refer to our response to Major Point 1, where we explain that we do not believe the initially skewed orientation of the mitotic spindle (observed in ~20% of the embryos) is responsible for the weak NMY-2 localization at the equatorial cortex. Furthermore, we have now performed a comprehensive analysis of the cortical and midplane distribution of NMY-2, along with another contractile ring component, Anillin. Both analyses confirm that upon LIN-5 depletion, the timely accumulation of contractile ring components at the equatorial cortex is significantly impaired.

In the revised manuscript, we have expanded our analysis on Anillin, given that Anillin localization to the equatorial membrane requires RhoA activation, but not myosin II (Straight et al., 2005, PMID: 15496454; Hickson and O'Farrell, 2008, PMID: 18209105). The decrease in equatorial enrichment of Anillin observed in LIN-5 depleted embryos further supports the idea that cortical LIN-5 directly regulates the timely accumulation of contractile ring components at the equatorial membrane, rather than acting indirectly by stripping NMY-2 from the polar cortex. These new data are included in Fig. 3A-3D and Fig. S5 and are discussed on pages 14 and 15 of the revised manuscript.

Please also refer to our response to a similar major point (Point #2) raised by the first reviewer. As we have now included our analysis of Anillin, another contractile ring protein, we have decided not to analyze the spatial localization of F-actin, as this would be redundant given our current findings.

(4) The PAR-5 rescue experiment is also difficult to interpret. The observation that PAR-5 depletion compensates for the effects of LIN-5 does not imply anything about how LIN-5 is acting. PAR-5 depletion does not just affect the cortical localization of centralspindlin, it also affects the cortical localization and distribution of the PAR proteins.

Response: The goal of this experiment was to investigate whether the reduced levels of equatorial cortical NMY-2 are the primary cause of the delayed furrow involution observed in LIN-5-depleted embryos. If this hypothesis is correct, restoring NMY-2 levels at the equatorial cortex should rescue the delayed furrow involution phenotype. Given that PAR-5 depletion leads to excessive cortical NMY-2 accumulation by enhancing centralspindlin localization at the cortex (Basant et al., 2015, PMID: 25898168), we tested whether increasing NMY-2 levels could compensate for the furrow involution delay in *lin-5(RNAi)* embryos. If the delay was caused by a factor independent of the contractile ring, increasing NMY-2 at the equatorial cortex would not have restored normal furrow involution timing. However, we observed a complete rescue of furrow involution timing, suggesting that the LIN-5 complex plays a key role in maintaining NMY-2 levels at the equatorial cortex. These results emphasize that the NMY-2 levels maintained by LIN-5, rather than its clearance from the polar cortex, are critical for regulating the timing of furrow involution.

(5) Jordan and Canman (2016) also considered how PAR proteins PAR-2 and PAR-6 affect contractile ring assembly and furrow ingression and propose a role for the PARs in preventing accumulation of anillin and septin at the cleavage furrow. It would be useful to consider these results in interpreting the PAR depletion experiments shown in Figure 2.

The reviewer correctly pointed out that Jordan et al., 2016; PMID: 26728855- defined the role of aPAR and pPAR in cytokinesis. However, we believe the rationale behind their experiments and ours is quite different. While we acknowledge their findings on differential NMY-2 levels upon PAR depletion, we find it challenging to compare them with our data due to fundamental differences in experimental design. Their study was conducted in *nmy-2(ts)* mutants under semi-permissive conditions, whereas our research was strictly performed with control embryos as the standard reference. Moreover, while the authors focused on the contractile ring after

furrow ingression, we specifically examined the role of PAR proteins at the time of furrow formation.

Our rationale for depleting aPAR and pPAR was to examine their impact on cortical LIN-5-based complexes. Our data indicate that weak cortical levels of LIN-5 [in *par-2(RNAi)*], but not the equally enriched levels of LIN-5 [in *par-3(RNAi)*], affect the timing of cleavage furrow formation — a parameter that was not assessed in their study.

As suggested by Reviewer 1, we have now included data related to GPR-1/2 depletion, which is crucial for the cortical accumulation of LIN-5. This new data, consistent with our *par-2(RNAi)* results, shows that cortical LIN-5 is critical for controlling the timing of furrow formation.

(6) The observation that furrow initiation is delayed in AB cells where LIN-5 does not have a major role in axial spindle position is interesting, but in these cells there are significant delays in spindle pole separation that could affect the patterning of myosin and other contractile ring components. Further analysis of the LIN-5 phenotype in these cells could be very informative.

Response. We thank the reviewer for raising this concern. As requested, we have now rescued the aster separation defect by inactivating SPD-1 in *lin-5(RNAi)* embryos in the AB cell. In *lin-5(RNAi); spd-1(ts)* embryos, we observe a similar delay in furrow involution. It is important to note that a subset of *lin-5(RNAi); spd-1(ts)* embryos exhibit unstable midbodies and fail to complete cytokinesis (~41%, Fig. 5). Therefore, for this analysis, we focused on the rest of the embryos that do not show abscission defects in the one-cell embryo. These results are presented in the new Fig. S2C-S2D and discussed on p. 9 of the revised manuscript. Please note that these findings are consistent with our previous observations in the P0 cell, where we demonstrated that LIN-5 does not influence cleavage furrow formation via its role in aster separation.

Minor points:

(1) NEBD is used as a time reference to compute delays. Why not use anaphase onset - a more proximal and equally accurate marker?

-Both NEBD (nuclear envelope breakdown) and anaphase onset are commonly used to measure the timing of furrow onset. However, since NEBD marks the onset of mitosis, we used NEBD timing as a reference point. Other studies such as Lewellyn et al., 2010 (PMID: 19889842); Mangal et al., 2018 (PMID: 29311228); and Lebedev et al., 2023 (PMID: 37665665) also rely on NEBD as the starting point for their analyses.

(2) Fig S3 Quantitative measurements of polar myosin levels are made after the furrow has already initiated. Why not before/during initiation?

-Thanks, we have now perform quantitative analysis of NMY-2 at and before furrow onset (see new Fig. 3, Fig. S3). We have also added now the quantitative analysis of Anillin in the revised manuscript.

(3) Time axis labels for kymographs are confusing. What does 100 sec onwards from NEBD mean exactly? Labelled vertical tic marks would be more clear.

-Thanks, the timing is with respect to NEBD and now we have simply mentioned time after NEBD in the figure legends, thanks.

April 4, 2025

RE: JCB Manuscript #202406059R-A

Sachin Kotak
Indian Institute of Science Bangalore

Dear Prof. Kotak,

Thank you for submitting your revised manuscript entitled "A direct role for LIN-5 (NuMA) in control of cytokinetic furrow formation and cytokinesis completion." We would be happy to publish your paper in JCB pending final revisions necessary to meet our formatting guidelines and text/figure changes to address the remaining reviewer comments (see details below).

A. MANUSCRIPT ORGANIZATION AND FORMATTING:

1) Text limits: Character count for Reports is < 20,000, not including spaces. Count includes title page, abstract, introduction, results & discussion, and acknowledgments. Count does not include materials and methods, figure legends, references, tables, or supplemental legends.

2) Figure formatting: Reports may have up to 5 main text figures. Scale bars must be present on all microscopy images, including inset magnifications. Please add scale bars for Figures 3E/F/K-O & 4A and for magnifications in S3E/F, S4G, S6B, & S7C-F.

Also, please avoid pairing red and green for images and graphs to ensure legibility for color-blind readers. If red and green are paired for images, please ensure that the particular red and green hues used in micrographs are distinctive with any of the colorblind types. If not, please modify colors accordingly or provide separate images of the individual channels.

3) Statistical analysis: Error bars on graphic representations of numerical data must be clearly described in the figure legend. The number of independent data points (n) represented in a graph must be indicated in the legend. Please, indicate whether 'n' refers to technical or biological replicates (i.e. number of analyzed cells, samples or animals, number of independent experiments). If independent experiments with multiple biological replicates have been performed, we recommend using distribution-reproducibility SuperPlots (please see Lord et al., JCB 2020) to better display the distribution of the entire dataset, and report statistics (such as means, error bars, and P values) that address the reproducibility of the findings.

Statistical methods should be explained in full in the materials and methods. For figures presenting pooled data the statistical measure should be defined in the figure legends. Please also be sure to indicate the statistical tests used in each of your experiments (both in the figure legend itself and in a separate methods section) as well as the parameters of the test (for example, if you ran a t-test, please indicate if it was one- or two-sided, etc.). Also, if you used parametric tests, please indicate if the data distribution was tested for normality (and if so, how). If not, you must state something to the effect that "Data distribution was assumed to be normal but this was not formally tested."

4) Title: To convey the advance more clearly, we suggest revising the title to the following:
"A cortical pool of LIN-5 (NuMA) controls cytokinetic furrow formation and cytokinesis completion."

5) Materials and methods: Should be comprehensive and not simply reference a previous publication for details on how an experiment was performed. Please provide full descriptions (at least in brief) in the text for readers who may not have access to referenced manuscripts. The text should not refer to methods "...as previously described."

6) For all cell lines, vectors, constructs/cDNAs, etc. - all genetic material: please include database / vendor ID (e.g. Addgene, ATCC, etc.) or if unavailable, please briefly describe their basic genetic features, even if described in other published work or gifted to you by other investigators (and provide references where appropriate). Please be sure to provide the sequences for all of your oligos: primers, si/shRNA, RNAi, gRNAs, etc. in the materials and methods. You must also indicate in the methods the source, species, and catalog numbers/vendor identifiers (where appropriate) for all of your antibodies, including secondary. If antibodies are not commercial, please add a reference citation if possible.

7) Microscope image acquisition: The following information must be provided about the acquisition and processing of images:
a. Make and model of microscope

- b. Type, magnification, and numerical aperture of the objective lenses
- c. Temperature
- d. Imaging medium
- e. Fluorochromes
- f. Camera make and model
- g. Acquisition software
- h. Any software used for image processing subsequent to data acquisition. Please include details and types of operations involved (e.g., type of deconvolution, 3D reconstitutions, surface or volume rendering, gamma adjustments, etc.).

8) References: There is no limit to the number of references cited in a manuscript. References should be cited parenthetically in the text by author and year of publication. Abbreviate the names of journals according to PubMed. JCB formatting does not allow for supplemental references, please remove these and add any non-duplicate references to the main reference list.

9) Supplemental materials: Reports may generally have up to 3 supplemental figures and 10 videos. You currently exceed this limit but, in this case, we will be able to give you the extra space but please try to consolidate these if possible. Please also note that tables, like figures, should be provided as individual, editable files. A summary of all supplemental material should appear at the end of the Materials and methods section. Please include one brief sentence per item.

10) Video legends: Should describe what is being shown, the cell type or tissue being viewed (including relevant cell treatments, concentration and duration, or transfection), the imaging method (e.g., time-lapse epifluorescence microscopy), what each color represents, how often frames were collected, the frames/second display rate, and the number of any figure that has related video stills or images.

11) eTOC summary: A ~40-50 word summary that describes the context and significance of the findings for a general readership should be included on the title page. The statement should be written in the present tense and refer to the work in the third person. It should begin with "First author name(s) et al..." to match our preferred style.

13) A separate author contribution section is required following the Acknowledgments in all research manuscripts. All authors should be mentioned and designated by their first and middle initials and full surnames. We encourage use of the CRediT nomenclature (<https://casrai.org/credit/>).

14) ORCID IDs: ORCID IDs are unique identifiers allowing researchers to create a record of their various scholarly contributions in a single place. Please note that ORCID IDs are required for all authors. At resubmission of your final files, please be sure to provide your ORCID ID and those of all co-authors.

15) Journal of Cell Biology now requires a data availability statement for all research article submissions. These statements will be published in the article directly above the Acknowledgments. The statement should address all data underlying the research presented in the manuscript. Please visit the JCB instructions for authors for guidelines and examples of statements at (<https://rupress.org/jcb/pages/editorial-policies#data-availability-statement>).

B. FINAL FILES:

-- A response to final reviewer comments.

****The license to publish form must be signed before your manuscript can be sent to production. A link to the electronic license to publish form will be sent to the corresponding author only. Please take a moment to check your funder requirements before choosing the appropriate license.****

Thank you for your attention to these final processing requirements. Please revise and format the manuscript and upload materials within 7 days. If you need an extension for whatever reason, please let us know and we can work with you to determine a suitable revision period.

Thank you for this interesting contribution, we look forward to publishing your paper in Journal of Cell Biology.

Sincerely,

Karen Oegema, PhD
Monitoring Editor
Journal of Cell Biology

Dan Simon, PhD
Scientific Editor
Journal of Cell Biology

Reviewer #1 (Comments to the Authors (Required)):

In the revised version the authors included a thorough analysis of NMY-2 and anillin localization in *lin-5* mutant embryos. This shows that spindle mispositioning towards the anterior region in *lin-5* mutants allows NMY-2 enrichment at the posterior and suppresses NMY-2 accumulation at anterior pole. They continue to demonstrate that this phenotype can be rescued by altering the position of the aster: bringing the posterior aster closer to the pole decreases NMY-2 levels again in *lin-5* mutant embryos. Thus, changes in aster position in *lin-5* mutants affect the accumulation of NMY-2 on the poles. This raises the question whether the close opposition of the two asters (Fig. 1B, S4C, D), rather than confinement by LIN-5, is responsible for low NMY-2/anillin levels at the cell equator in *lin-5* mutants. The asters generate an inhibitory signal and if the asters fail to separate sufficiently the inhibitory signal could overwrite the stimulatory signal from the spindle midzone. Since the confinement of contractile ring components at the cell equator is a key conclusion of the manuscript, the authors must address this point by either increasing inter-aster distance by *spd-1* or *hcp-4* depletion in *lin-5* mutants or by discussing this alternative explanation in the text. After addressing my last points, I fully support publication in JCB.

Minor points:

1. Abstract: "We show that the cortical pool of LIN-5, recruited by GPR-1/2 and important for cortical force generation, regulates cleavage furrow formation independently of its roles in aster separation, spindle positioning, and myosin II removal from the cell cortex."
Since the data of authors suggests that LIN-5 is not required for "myosin II removal", I would suggest removing this part from the sentence.
2. In figure S1B the label on axis is missing.
3. Fig. 2A-C
"Here, and in subsequent embryo images, embryos are approximately 50 μm in length"
Why is it necessary to mention this?
4. "We also noted that a significant pool of contractile ring machinery, including ECT-2, ZEN-4, CYK-4, NMY-2, and Anillin are enriched at the midbody after furrow ingression (Fig. 5B and 5C; Fig. S7C-S5F)."
To me it sounds as authors report for the first time the localization of those proteins to the midbody. This, however, is not the case and appropriate references should be cited. Also, probably the authors mean Fig. S7A-F. Another site note, for me: ECT-2, ZEN-4 and CYK-4 are not part of the "contractile ring machinery", they are rather signaling components to build the machine.

5. "In these embryos, the contractile ring components CYK-4 and Anillin initially localized to the midbody and midbody ring; however, these proteins did not remain confined, and spread along the cortical surface, leading to furrow regression and ultimately cytokinesis failure."

For CYK-4 the authors do not show spreading along the cortical surface. They show that CYK-4 is not maintained on the overlapping MTs in *spd-1,lin-5* double. Therefore, sentence must be rewritten.

Reviewer #2 (Comments to the Authors (Required)):

The authors have done a very nice job of addressing my concerns (and clearing up some confusions in my part), as well as the concerns of the other reviewers. I support publication without reservation.

Response to the reviewers

Reviewer #1

We thank the reviewer for supporting our work. However, the reviewer asked for a few minor corrections/changes, that we have made, as explained below.

A few remaining concerns:

This raises the question whether the close opposition of the two asters (Fig. 1B, S4C, D), rather than confinement by LIN-5, is responsible for low NMY-2/anillin levels at the cell equator in lin-5 mutants. The asters generate an inhibitory signal and if the asters fail to separate sufficiently the inhibitory signal could overwrite the stimulatory signal from the spindle midzone. Since the confinement of contractile ring components at the cell equator is a key conclusion of the manuscript, the authors must address this point by either increasing inter-aster distance by spd-1 or hcp-4 depletion in lin-5 mutants or by discussing this alternative explanation in the text.

Response. We thank the reviewer for raising this point. We agree with the reviewer that the weak levels of NMY-2 and Anillin at the equatorial membrane in LIN-5-inhibited embryos could be linked to failure in proper aster separation. However, as shown in Fig. 1, cortical LIN-5-based complexes regulate the timing of cleavage furrow formation independently of their role in aster separation. The delayed furrow formation observed in LIN-5-depleted embryos correlates with reduced NMY-2 and Anillin at the equatorial cortex, and this delay persists even when aster separation is restored (Fig. 1B and 1J). This suggests that the low NMY-2 and Anillin levels are not merely a consequence of defective aster separation.

Additionally, in LIN-5-inhibited embryos, the lack of outward pulling forces results in more significant centralspindlin accumulation at the spindle midzone compared to control embryos (Fig. 5G vs. 5E, and Lee et al., 2015; PMID: 26088160). Therefore, not only aster-based inhibitory signal but also spindle midzone-based stimulatory signal is enhanced in these conditions. Nevertheless, since we have not formally analyzed NMY-2/Anillin localization upon co-depletion of LIN-5 and HCP-4, we have briefly discussed this possibility in the revised discussion on p. 20, as suggested by the reviewer.

Abstract: "We show that the cortical pool of LIN-5, recruited by GPR-1/2 and important for cortical force generation, regulates cleavage furrow formation independently of its roles in aster separation, spindle positioning, and myosin II removal from the cell cortex."

Since the data of authors suggests that LIN-5 is not required for "myosin II removal", I would suggest removing this part from the sentence.

Response. Thank you. As requested, we have incorporated this modification into the abstract.

In figure S1B the label on axis is missing.

Response. Added, thanks.

Fig. 2A-C "Here, and in subsequent embryo images, embryos are approximately 50 μ m in length" Why is it necessary to mention this?

Response. We agree that this information is not necessary and have therefore removed it from the sentence.

"We also noted that a significant pool of contractile ring machinery, including ECT-2, ZEN-4, CYK-4, NMY-2, and Anillin are enriched at the midbody after furrow ingression (Fig. 5B and 5C; Fig. S7C-S5F). "

To me it sounds as authors report for the first time the localization of those proteins to the midbody. This, however, is not the case and appropriate references should be cited. Also, probably the authors mean Fig. S7A-F. Another site note, for me: ECT-2, ZEN-4 and CYK-4 are not part of the "contractile ring machinery", they are rather signaling components to build the machine.

Response. We have slightly revised the sentence and added the appropriate references. Fig. S7A and S7B pertain to the localization of GPR-1/2, but we actually meant Fig. S7B–S7F. We also appreciate the reviewer's remarks that ECT-2, ZEN-4, and CYK-4 are part of the signaling network required for the assembly of the contractile ring machinery, and we have modified this and other sentences.

"In these embryos, the contractile ring components CYK-4 and Anillin initially localized to the midbody and midbody ring; however, these proteins did not remain confined, and spread along the cortical surface, leading to furrow regression and ultimately cytokinesis failure."

*For CYK-4 the authors do not show spreading along the cortical surface. They show that CYK-4 is not maintained on the overlapping MTs in *spd-1,lin-5* double. Therefore, the sentence must be rewritten.*

Response. Thanks, based on the reviewer's suggestion, we have now reworded the sentence.

Reviewer #2

We are deeply thankful to the reviewer for their kind words.